# Towards Performatively Stable Equilibria in Decision-Dependent Games for Arbitrary Data Distribution Maps

## Abstract

In decision-dependent games, multiple players optimize their decisions under a data distribution that shifts with their joint actions, creating complex dynamics in applications like market pricing. A practical consequence of these dynamics is the *performatively stable equilibrium*, where each player's strategy is a best response under the induced distribution. Prior work relies on $\beta$-smoothness, assuming Lipschitz continuity of loss function gradients with respect to the data distribution, which is impractical as the data distribution maps, i.e., the relationship between joint decision and the resulting distribution shifts, are typically unknown, rendering $\beta$ unobtainable. To overcome this limitation, we propose a gradient-based sensitivity measure that directly quantifies the impact of decision-induced distribution shifts. Leveraging this measure, we derive convergence guarantees for performatively stable equilibria under a practically feasible assumption of strong monotonicity. Accordingly, we develop a sensitivity-informed repeated retraining algorithm that adjusts players' loss functions based on the sensitivity measure, guaranteeing convergence to performatively stable equilibria for arbitrary data distribution maps. Experiments on prediction error minimization game, Cournot competition, and revenue maximization game show that our approach outperforms state-of-the-art baselines, achieving lower losses and faster convergence.

## 1 Introduction

In supervised learning, models are typically trained and tested under the assumption that data are sampled from a fixed distribution. However, in real-world applications, model predictions often influence decisions that shift the underlying data distribution, a phenomenon known as performative prediction (Perdomo et al., 2020). For example, a company predicting high demand may raise prices to maximize revenue, inadvertently reducing the actual demand. Such dynamics are ubiquitous in economy (Hardt et al., 2022), education (Zhang, 2024), and recommendation systems (Eilat & Rosenfeld, 2023), where decisions reshape data distributions.

While performative prediction has been extensively studied in single learner settings, where the distribution reacts solely to a single model, real-world scenarios often involve multiple competing agents. In these decision-dependent games (Narang et al., 2023), each player optimizes decisions based on a distribution influenced by not only their own actions but also those of others. For instance, a firm's demand may decrease due to its own price increase or competitors' price reductions, creating complex multi-agent interactions and dynamics that challenge traditional learning models.

Existing research in decision-dependent games focuses on two equilibria: Nash equilibria and performatively stable equilibria. Nash equilibria ensure that each player's decision is a best response across all potential induced distributions given a predefined data distribution map $\mathcal{D}_i(X)$, which describes the relationship between the joint decision $X = (x_1, x_2, \cdots, x_n)$ and the induced distribution $\mathcal{D}_i$ (Narang et al., 2023; Zhu et al., 2023). However, validating Nash equilibria is challenging in practice, as $\mathcal{D}_i(X)$ is typically unknown and may take various forms (Huang et al., 2013), making the inference of resulting distributions challenging or even infeasible during the optimization process. Mis-specifying $\mathcal{D}_i(X)$ can lead to unreliable equilibria, limiting their applicability.

In contrast, performatively stable equilibria require each player's decision to be a best response under the distribution induced by the current joint decision, enabling empirical validation without prior knowledge of $\mathcal{D}_i(X)$ (Cutler et al., 2024). Such a performatively stable equilibrium is highly desirable, since the joint decision optimized on the distribution $\mathcal{D}_i(X^{PS})$ will consistently converge to this equilibrium, eliminating the need for further updates of decisions. These equilibria are practical consequences of rational multi-player decisions, making them highly relevant to real-world applications. However, current studies rely on impractical assumptions, particularly $\beta$-smoothness, which assumes Lipschitz continuity of loss function gradients with respect to the data distribution (Narang et al., 2023; Cutler et al., 2024). Since $\mathcal{D}_i(X)$ is typically unknown, quantifying the extent of distribution shifts to verify $\beta$ is challenging, making it difficult to apply theoretical results in practice (A detailed example is provided in Appendix A). Additionally, prior research (Narang et al., 2023; Cutler et al., 2024) has applied the $\mathcal{W}_1$ distance to quantify distribution shifts; however, its sample complexity grows exponentially with the data dimensions (Fournier & Guillin, 2015), limiting the applicability of existing methods in high-dimensional spaces. To illustrate these challenges, we consider a revenue maximization game:

**Example 1.1.** *(Revenue Maximization Game (Narang et al., 2023)) Suppose $n$ firms, each indexed by $i \in [n]$, set prices $x_i \in \mathbb{R}^{d_i}$ for products across $d_i$ areas to maximize revenue $R_i = z_i^T x_i$. The demand $z_i \in \mathbb{R}^{d_i}$ decreases as $x_i$ increases or as competitors' prices $x_{-i} = (x_1, \cdots, x_{i-1}, x_{i+1}, \cdots, x_n)^T \in \mathbb{R}^{\sum_{j \neq i}^n d_j}$ decrease. Furthermore, price changes in one area $x_{i_j}$ may also affect demand in other areas $z_{i_k}$ ($k \neq j$), impacting $R_i$. The data distribution map $\mathcal{D}_i(X)$, relating the joint decision $X$ to the induced demand distribution, can be modeled as a logistic function (Bowerman et al., 2003; Phillips, 2021):*

$$z_{i,X} \sim \mathcal{D}_i(X) = 2z_i^{initial} \oslash \left(1 + e^{-(A_i x_i + A_{-i} x_{-i})}\right), \forall i \in [n], \tag{1}$$

*where $z_{i,X} \in \mathbb{R}^{d_i}$ is the demand of firm $i$ induced by prices $X$ set by $n$ firms, $z_i^{initial} \in \mathbb{R}^{d_i}$ is the initial demand of firm $i$, the operator $\oslash$ denotes the Hadamard division (i.e., the element-wise division) (Cyganek, 2013), and $A_i \in \mathbb{R}^{d_i \times d_i}$ and $A_{-i} \in \mathbb{R}^{d_i \times \sum_{j \neq i}^n d_j}$ control demand shifts induced by $x_i$ and $x_{-i}$, respectively.*

In Example 1.1, the demand function $\mathcal{D}_i(X)$ may take various forms (Huang et al., 2013; Chen et al., 2006; Song et al., 2008; Kocabıyıkoğlu & Popescu, 2011), complicating the validation of Nash equilibria due to the potential mis-specification. For performatively stable equilibria, estimating $\beta$ is challenging without knowing $\mathcal{D}_i(X)$, as the range of demand shifts cannot be quantified. Moreover, prior methods relying on $\mathcal{W}_1$ distance to measure distribution shifts face high sample complexity, particularly for global firms (Bernard et al., 2018) with high-dimensional $d_i$, making accurate estimation impractical. Consequently, these limitations hinder the real-world applicability of existing theoretical results for performatively stable equilibria.

## 1.1 Contribution

This paper proposes a framework for achieving performatively stable equilibria in decision-dependent games for arbitrary data distribution maps, eliminating the need for impractical $\beta$-smoothness assumptions. Our contribution can be summarized as follows:

- **Gradient-based Sensitivity Definition and Theoretical Guarantees**: We introduce a gradient-based $\hat{\varepsilon}_i$-sensitivity measure in Section 3.1 that quantifies the impact of decision-induced distribution shifts for arbitrary $\mathcal{D}_i(X)$. By leveraging each player's loss function gradient, this measure directly captures the influence of distribution shifts on individual decisions. Using this definition, we derive convergence guarantees for performatively stable equilibria in Sections 3.2-3.3, effective even with finite samples, under practically feasible conditions of strong monotonicity. Our analyses rely on measurable $\hat{\varepsilon}_i$ and adjustable monotonicity parameter $\alpha$, which can be tuned by a regularization approach, ensuring wide applicability across diverse scenarios.

- **Algorithm Design and Systematic Validation**: We develop the **S**ensitivity-**I**nformed **R**epeated **R**etraining (SIR$^2$) algorithm in Section 4, which adjusts loss functions by tuning the monotonicity parameter $\alpha$ based on estimated $\hat{\varepsilon}_i$-sensitivity, guaranteeing convergence to performatively stable

equilibria. In Section 5, we evaluate the effectiveness of $\text{SIR}^2$ by comparing it against five state-of-the-art methods in the prediction error minimization game between prediction platforms, Cournot competition in crude oil trade, and the revenue maximization game in ride-share markets. Our method outperforms baselines in two key aspects: players' losses and equilibrium convergence speed.

## 1.2 Related Work

Performative prediction focuses on single-learner settings, where a model's predictions alter the distribution, which, in turn, influences the model's predictions. Performative optimality minimizes the performative loss by incorporating the data distribution map into the objective, using convex optimization (Miller et al., 2021; Yan & Cao, 2023; Lin & Zrnic, 2024) or gradient descent with estimated performative gradients (Izzo et al., 2021; 2022; Liu et al., 2024). In contrast, performative stability minimizes loss under the induced data distribution, preventing further distribution shifts through repeated retraining Perdomo et al. (2020); Mofakhami et al. (2023); Brown et al. (2022) or repeated gradient descent (Mendler-Dünner et al., 2020; Shan et al., 2023).

Decision-dependent games extend these concepts to multi-agent settings, where joint decisions shape the distribution. Existing work for Nash equilibria optimizes performative losses of all players, assuming linear data distribution maps. Specifically, in each iteration, each player alternates between: (1) updating the decision using the estimated performative gradient, and (2) updating the data distribution map by fitting the predefined function through collected decision-distribution pairs (Narang et al., 2023) or online stochastic approximation (Zhu et al., 2023). For performatively stable equilibria, current studies conduct analyses under three conditions: (1) an $\alpha$-strongly monotone game, (2) $\beta$-smooth loss functions, and (3) $\gamma$-Lipschitz data distribution maps with respect to $\mathcal{W}_1$ distance. Their theoretical results guarantee the convergence through repeated retraining (Narang et al., 2023) or repeated gradient descent (Narang et al., 2023; Cutler et al., 2024). Different from existing work, our framework eliminates the assumption of $\beta$-smoothness, using a gradient-based sensitivity measure to achieve practical convergence for arbitrary distribution maps.

## 2 Problem Statement

This section formalizes the problem under consideration. First, we introduce decision-dependent games. We then define the performatively stable equilibrium, a central concept in our analysis. Finally, we describe the repeated retraining procedure that forms the basis of our analytical framework.

In a decision-dependent game, each player seeks to minimize their own loss function through a best response to the anticipated optimal decisions of the other players. This loss function depends on both the decisions made by all players and the resulting decision-dependent data distribution, which is induced by the joint decision. For an index set $[n] = \{1, 2, \cdots, n\}$, each player $i \in [n]$ aims to solve the following decision-dependent optimization problem:

$$\min_{x_i \in \mathcal{X}_i} \mathbb{E}_{Z_i \sim \mathcal{D}_i(X)} \ell_i(x_i, x_{-i}, Z_i), \tag{2}$$

where $x_i$ is the decision vector of player $i$ with the dimension $d_i$, $\mathcal{X}_i \subset \mathbb{R}^{d_i}$ is the decision space of player $i$, $\mathcal{X} = \mathcal{X}_1 \times \mathcal{X}_2 \times \cdots \times \mathcal{X}_n \subset \mathbb{R}^d$ is the joint decision space with the dimension $d = \sum_{i=1}^n d_i$, $X = (x_1, x_2, \cdots, x_n)^T \in \mathcal{X}$ is the joint decision vector of all players, $x_{-i} = (x_1, \cdots, x_{i-1}, x_{i+1}, \cdots, x_n)^T$ is the decision vector of all players except player $i$, $\mathcal{D}_i(X)$ is the distribution for player $i$ induced by the joint decision vector $X$, $Z_i = \{z_{i_1}, z_{i_2}, \cdots, z_{i_m}\}$ is a sample set of $m$ data points drawn i.i.d. from $\mathcal{D}_i(X)$, and $\ell_i(\cdot)$ is the loss function of player $i$.

The decision-dependent game on the data distributions induced by joint decision $Y \in \mathcal{X}$ is composed of the loss functions of all $n$ players, which can be jointly formulated as follows:

$$\mathcal{G}(Y) := \left( \mathbb{E}_{Z_1 \sim \mathcal{D}_1(Y)} \ell_1(x_1, x_{-1}, Z_1), \cdots, \mathbb{E}_{Z_n \sim \mathcal{D}_n(Y)} \ell_n(x_n, x_{-n}, Z_n) \right)^T, \tag{3}$$

Accordingly, a joint decision $X^* = (x_1^*, x_2^*, \cdots, x_n^*)^T \in \mathbb{R}^d$ is called a Nash equilibrium of the game $\mathcal{G}(Y)$ in Eq. (3) if the following holds:

$$x_i^* \in \arg \min_{x_i \in \mathcal{X}_i} \mathbb{E}_{Z_i \sim \mathcal{D}_i(Y)} \ell_i(x_i, x_{-i}^*, Z_i), \forall i \in [n], \tag{4}$$

where $x_{-i}^* = (x_1^*, \cdots, x_{i-1}^*, x_{i+1}^*, \cdots, x_n^*)^T$ is the decision of all players except player $i$ in the Nash equilibrium. We denote such a Nash equilibrium of game $\mathcal{G}(Y)$ as $Nash(Y)$:

$$Nash(Y) := \{X^* \in \mathcal{X} : X^* \text{ is a Nash Equilibrium of game } \mathcal{G}(Y)\}. \tag{5}$$

In a decision-dependent game, a joint decision $X$ by all players may alter the underlying data distributions. This data distribution shift can create opportunities for players to reduce their losses by adapting decisions to the new distributions. In this context, a practical consequence is the performatively stable equilibrium, where the joint decision $X^{PS}$ is a Nash equilibrium under the decision-induced data distributions $\mathcal{D}_i(X^{PS}), \forall i \in [n]$. Training on the distributions $\mathcal{D}_i(X^{PS}), \forall i \in [n]$ results in the same $X^{PS}$, which prevents the distributions from further shifting. Formally, a performatively stable equilibrium is defined as follows:

**Definition 1.** *(Performatively stable equilibrium) A joint decision $X^{PS} \in \mathcal{X}$ is a performatively stable equilibrium if it is a Nash equilibrium on the data distribution induced by $X^{PS}$:*

$$X^{PS} = Nash(X^{PS}). \tag{6}$$

In this paper, we investigate the conditions under which a performatively stable equilibrium can be reached through the following repeated retraining procedure[1], where each player $i$ optimize the next decision $x_i^{t+1}$ according to the current distribution $\mathcal{D}_i(X^t)$. In this procedure, the performatively stable equilibrium is verified when the joint decision stabilizes, i.e., $X^{t+1} = X^t$.

**Definition 2.** *(Repeated retraining) Repeated retraining refers to the procedure where, at each step $t + 1$, the joint decision $X^{t+1}$ is determined based on the data distribution $\mathcal{D}_i(X^t)$ for all $i \in [n]$, induced by the previous joint decision $X^t$:*

$$X^{t+1} = Nash(X^t), \tag{7}$$

*where $X^0, X^1, \cdots X^t, X^{t+1}, \cdots \in \mathcal{X}_{RR}$, and $\mathcal{X}_{RR}$ denotes the closed convex hull of the sequence of joint decisions $\{X^t\}$ obtained during the repeated retraining procedure.*

In this paper, we employ repeated retraining, as an example, to derive theoretical guarantees and develop our algorithm for decision-dependent games. Notably, the proposed framework is flexible and can be readily extended to incorporate other decision update approaches, such as repeated gradient descent, as explored in prior work (Narang et al., 2023; Cutler et al., 2024).

## 3 Theoretical Results

In this section, we present our main theoretical results, including: (1) a gradient-based sensitivity measure of $\mathcal{D}_i(X)$, (2) guarantees for convergence to performatively stable equilibrium with feasible conditions, and (3) finite-sample convergence guarantees.

### 3.1 Gradient-based Sensitivity Measure

We introduce the $\hat{\varepsilon}_i$-sensitivity measure for $\mathcal{D}_i(X)$ of player $i$, using practically calculable gradients to quantify performative shifts without the impractical $\beta$-smoothness assumption. This definition stems from the observation that shifts in data distributions can be reflected by changes to each player's loss function gradient during the repeated retraining procedure.

---

[1]Note that, although the repeated gradient descent approaches can also achieve a performatively stable equilibrium, we adopt the repeated retraining in this paper to avoid frequent decision deployments required for partial model updates in repeated gradient descent methods, which increases practical costs. In addition, existing gradient-based methods often rely on a $\beta$-joint smoothness condition for equilibrium analysis, where $\beta$, a critical parameter to algorithm design, may be unverifiable in real-world scenarios due to the unknown data distribution maps.

**Definition 3.** *($\hat{\varepsilon}_i$-sensitivity) The data distribution map $\mathcal{D}_i(X)$ is $\hat{\varepsilon}_i$-sensitive for each $i \in [n]$ if, for all joint decisions $Y, X, X' \in \mathcal{X}_{RR}$, the following inequality holds during repeated retraining:*

$$\left\| \underset{Z_i \sim \mathcal{D}_i(X)}{\mathbb{E}} \nabla_{y_i} \ell_i(y_i, y_{-i}, Z_i) - \underset{Z_i' \sim \mathcal{D}_i(X')}{\mathbb{E}} \nabla_{y_i} \ell_i(y_i, y_{-i}, Z_i') \right\|_2 \leq \hat{\varepsilon}_i \|X - X'\|_2. \tag{8}$$

For comparison, we also present the definition of $\beta_i$-smoothness. A loss function $\ell_i(x_i, x_{-i}, Z_i)$ is $\beta_i$-smooth if, $\forall x_i, x_i' \in \mathcal{X}_i$ and $\forall z_{i_j}, z_{i_j}' \in \mathcal{Z}_i, \mathcal{Z}_i := \cup_{X \in \mathcal{X}} \sup(\mathcal{D}_i(X))$, the following conditions hold: $\|\nabla_{x_i} \ell_i(x_i, x_{-i}, z_{i_j}) - \nabla_{x_i} \ell_i(x_i, x_{-i}, z_{i_j}')\|_2 \leq \beta_i \|z_{i_j} - z_{i_j}'\|_2, \|\nabla_{x_i} \ell_i(x_i, x_{-i}, z_{i_j}) - \nabla_{x_i} \ell_i(x_i', x_{-i}, z_{i_j})\|_2 \leq \beta_i \|x_i - x_i'\|_2$. Since the distribution map $\mathcal{D}_i(X)$ is typically unknown, the sample space $\mathcal{Z}_i$ is difficult to characterize. Consequently, the constant $\beta_i$ is challenging to verify in practice. Although $\hat{\varepsilon}_i$-sensitivity serves an analogous role in convergence proofs to parameters in prior work (Perdomo et al., 2020; Brown et al., 2022; Narang et al., 2023), it is defined for the repeated retraining procedure, which makes it practically computable. Specifically, for arbitrary distribution map $\mathcal{D}_i(X)$, we can calculate $\hat{\varepsilon}_i$ during repeated retraining by:

$$\max_{t,t'} \frac{\|\mathbb{E}_{Z^t \sim \mathcal{D}_i(X^t)} \nabla_{x_i^t} \ell_i(x_i^t, x_{-i}^t, Z_i^t) - \mathbb{E}_{Z^{t'} \sim \mathcal{D}_i(X^{t'})} \nabla_{x_i^t} \ell_i(x_i^t, x_{-i}^t, Z_i^{t'})\|_2}{\|X^t - X^{t'}\|_2}, \forall t, t' = 1, 2, \cdots \tag{9}$$

This applicability enables the relaxation of unverifiable conditions and leads to verifiable convergence conditions. This measure bounds the gradient discrepancy under distributions induced by $X$ and $X'$ using $\hat{\varepsilon}_i$, which can be practically estimated via the supremum during repeated retraining iterations (Section 4). It characterizes the sensitivity of the data distribution map $\mathcal{D}_i(X)$ without requiring its explicit representation, allowing $\mathcal{D}_i(X)$ to be arbitrarily complex. It offers a verifiable criterion for the sensitivity of $\mathcal{D}_i(X)$, serving as a foundation for the following convergence analysis.

### 3.2 Theoretical Guarantees for Performatively Stable Equilibria

Before presenting our theoretical results, we introduce two key definitions, which are essential for understanding the subsequent theorems.

**Definition 4.** *(Individual gradients of game $\mathcal{G}(\cdot)$) The individual gradients of the joint decision $X$, under the distributions induced by $Y$, are defined as:*

$$\nabla_X \mathcal{G}(Y) := \left( \underset{Z_1 \sim \mathcal{D}_1(Y)}{\mathbb{E}} \nabla_{x_1} \ell_1(x_1, x_{-1}, Z_1), \cdots, \underset{Z_n \sim \mathcal{D}_n(Y)}{\mathbb{E}} \nabla_{x_n} \ell_n(x_n, x_{-n}, Z_n) \right)^T. \tag{10}$$

**Definition 5.** *($\alpha$-strong monotonicity) A game $\mathcal{G}(Y)$ is $\alpha$-strongly monotone for $Y \in \mathcal{X}$ if, for all joint decisions $X, X' \in \mathcal{X}$, the following inequality holds:*

$$\alpha \|X - X'\|_2^2 \leq (\nabla_X \mathcal{G}(Y) - \nabla_{X'} \mathcal{G}(Y))^T (X - X'). \tag{11}$$

The $\alpha$-strongly monotonicity guarantees the existence of a unique Nash equilibrium in each iteration of the repeated retraining (Rosen, 1965), ensuring tractable game dynamics. This condition is applied in existing research on decision-dependent games (Narang et al., 2023; Zhu et al., 2023; Cutler et al., 2024), facilitating the analysis's focus on the performative effect. Having established these definitions, we now present a corollary of Theorem 7 in (Narang et al., 2023) and a new theorem that provide practical convergence conditions for performatively stable equilibria.

**Corollary 1.** *If the game $\mathcal{G}(Y)$ is $\alpha$-strongly monotone and each data distribution map $\mathcal{D}_i(Y)$ is $\hat{\varepsilon}_i$-sensitive for $i \in [n]$, then the repeated retraining procedure of the game defined in Eq. (3) produces a sequence of joint decisions $\{X^0, X^1, \cdots, X^{t-1}, X^t, X^{t+1}, \cdots\} \subset \mathcal{X}_{RR}$ such that:*

$$\|X^{t+1} - X^t\|_2 \leq \frac{\sqrt{\sum_{i=1}^n \hat{\varepsilon}_i^2}}{\alpha} \|X^t - X^{t-1}\|_2 \leq \left( \frac{\sqrt{\sum_{i=1}^n \hat{\varepsilon}_i^2}}{\alpha} \right)^t \|X^1 - X^0\|_2. \tag{12}$$

*Proof sketch.* Here, we provide a proof sketch. The complete proof is given in Appendix B.

Given an $\alpha$-strongly monotone $\mathcal{G}(\cdot)$ and the definition of $Nash(X)$ in Eq. (5), by applying the first-order optimality conditions (Bubeck et al., 2015) and the Cauchy-Schwarz inequality (Hunter & Nachtergaele, 2001), we have, $\forall X, X' \in \mathcal{X}$:

$$\alpha \|Nash(X) - Nash(X')\|_2 \leq \|\nabla_{Nash(X')}\mathcal{G}(X') - \nabla_{Nash(X')}\mathcal{G}(X)\|_2. \tag{13}$$

From Definitions 3 and 4, $\forall Y, X, X' \in \mathcal{X}_{RR}$, we have:

$$\|\nabla_Y \mathcal{G}(X) - \nabla_Y \mathcal{G}(X')\|_2 \leq \sqrt{\sum_{i=1}^{n} \hat{\varepsilon}_i^2} \|X - X'\|_2. \tag{14}$$

We set $Y = Nash(X')$ in Eq. (14), and combined with Eq. (13), we have:

$$\|Nash(X) - Nash(X')\|_2 \leq \frac{\sqrt{\sum_{i=1}^{n} \hat{\varepsilon}_i^2}}{\alpha} \|X - X'\|_2. \tag{15}$$

Without loss of generality, we set $X = X^t$ and $X' = X^{t-1}$. Then, we obtain:

$$\|X^{t+1} - X^t\|_2 \leq \frac{\sqrt{\sum_{i=1}^{n} \hat{\varepsilon}_i^2}}{\alpha} \|X^t - X^{t-1}\|_2 \leq \left( \frac{\sqrt{\sum_{i=1}^{n} \hat{\varepsilon}_i^2}}{\alpha} \right)^t \|X^1 - X^0\|_2. \tag{16}$$

$\blacksquare$

Corollary 1 establishes the dynamics of players' decisions during repeated retraining. Leveraging this result, Theorem 2 demonstrates that, under certain conditions, the repeated retraining procedure converges to performatively stable equilibria[2].

**Theorem 2.** *If $\alpha > \sqrt{\sum_{i=1}^{n} \hat{\varepsilon}_i^2}$, then the repeated retraining procedure for the game defined in Eq. (3) converges to a performatively stable equilibrium $X^{PS}$ within the set $\mathcal{X}_{RR}$, i.e., $X^{PS} \in \mathcal{X}_{RR}$.*

The main contribution of Corollary 1 and Theorem 2 lies in relaxing the unverifiable conditions in a performative setting in prior research (Perdomo et al., 2020; Mendler-Dünner et al., 2020; Narang et al., 2023; Cutler et al., 2024). Our convergence analysis is inspired by these earlier works and leverages the contraction mapping theorem. By introducing the calculable $\hat{\varepsilon}_i$-sensitivity measure, we establish practical guidance for algorithm design to ensure convergence to a performatively stable equilibrium. In addition, the parameters in prior works (Narang et al., 2023; Cutler et al., 2024) are defined on the full decision space $\mathcal{X}$, and our $\hat{\varepsilon}_i$ is defined on the subspace $\mathcal{X}_{RR}$. Consequently, $\hat{\varepsilon}_i$ is smaller than or equal to these parameters, resulting in a more relaxed requirement on the strong monotonicity parameter $\alpha$.

### 3.3 Finite-Sample Convergence Analysis

Corollary 1 and Theorem 2 ensure guaranteed convergence to performatively stable equilibria under practically feasible conditions, assuming full distribution access. In reality, players rely on finite collected samples, thereby introducing challenges due to limited data and potential measurement noise. To address this, we extend our results in Theorem 3, deriving finite-sample convergence conditions to maintain the practicality of our framework. First, we derive a lemma that quantifies the discrepancy between empirical and expected individual gradients. This will facilitate our analysis of convergence to performatively stable equilibria with finite samples[3].

---

[2]The proof of Theorem 2 is given in Appendix C.

[3]The proof of Lemma 1 is given in Appendix D. Note that Lemma 1 (and the following Theorem 3) establish theoretical results by using the Central Limit Theorem, which holds as the sample size $m \to \infty$. For more practical scenarios, we provide two additional results: Lemma 2 (Appendix E) leverages the Berry-Esseen theorem to establish a probability bound for finite $m$, and Lemma 3 (Appendix F) provides a finite-sample bound using Chebyshev's inequality. Corresponding variants of Theorem 3 can be found in Appendix G.2.

**Lemma 1.** *Consider the empirical average individual gradients $\overline{\nabla_X \mathcal{G}_m(Y)}$ estimated over $m$ samples of a game $\mathcal{G}(Y)$:*

$$\overline{\nabla_X \mathcal{G}_m(Y)} := \left( \frac{1}{m} \sum_{j=1}^{m} \nabla_{x_1} \ell_1(x_1, x_{-1}, z_{1_j}), \cdots, \frac{1}{m} \sum_{j=1}^{m} \nabla_{x_n} \ell_n(x_n, x_{-n}, z_{n_j}) \right)^T . \tag{17}$$

*Then for any $\delta > 0$, with probability at least $F_{\chi^2(d)}(\frac{m\delta^2}{\sigma})$, the following inequality holds:*

$$\left\| \overline{\nabla_X \mathcal{G}_m(Y)} - \nabla_X \mathcal{G}(Y) \right\|_2 \leq \delta, \tag{18}$$

*where $\nabla_X \mathcal{G}(Y)$ is the expected individual gradients in Definition 4, $F_{\chi^2(d)}(\cdot)$ denotes the cumulative distribution function of the $\chi^2$ distribution with $d$ degrees of freedom, and $\sigma = \lambda_{\max}(\mathbf{\Sigma})$ denotes the largest eigenvalue of $\mathbf{\Sigma}$, the covariance matrix of the average individual gradients $\overline{\nabla_X \mathcal{G}_m(Y)}$.*

With Lemma 1, we now present the following theorem.

**Theorem 3.** *If a game $\mathcal{G}(Y)$ is $\alpha$-strongly monotone and each data distribution map $\mathcal{D}_i(X)$ is $\hat{\varepsilon}_i$-sensitive for all $i \in [n]$, and $\alpha > 2\sqrt{\sum_{i=1}^{n} \hat{\varepsilon}_i^2}$, then the repeated retraining procedure of the game defined in Eq. (3) with finite samples produces a sequence of joint decisions $\{X^0, X^1, \cdots, X^{t-1}, X^t, X^{t+1}, \cdots\} \subset \mathcal{X}_{RR}$ that converges to a performatively stable equilibrium $X^{PS}$. Specifically, with probability at least $\prod_{\mu=1}^{t} \left( F_{\chi^2(d)}(\frac{m_\mu \sum_{i=1}^{n} \hat{\varepsilon}_i^2 \delta^2}{\sigma_\mu}) \right)$, the following inequality holds:*

$$\|X^t - X^{PS}\|_2 \leq \delta, \forall t \geq \frac{\log\left( \frac{\delta}{\|X^0 - X^{PS}\|_2} \right)}{\log\left( \frac{2\sqrt{\sum_{i=1}^{n} \hat{\varepsilon}_i^2}}{\alpha} \right)}, \tag{19}$$

*where $\delta > 0$ controls the convergence rate, $m_\mu$ denotes the number of samples collected by each player at time step $\mu$, and $\sigma_\mu = \lambda_{\max}(\mathbf{\Sigma}_\mu)$ denotes the largest eigenvalue of $\mathbf{\Sigma}_\mu$ at time step $\mu$.*

*Proof sketch.* Here, we provide a proof sketch. The complete proof is given in Appendix G.

By applying the triangle inequality, we have:

$$\begin{aligned}
&\|\overline{\nabla_{Nash(X^t)} \mathcal{G}_{m_t}(X^t)} - \nabla_{Nash(X^t)} \mathcal{G}(X^{PS})\|_2 \\
&\leq \|\overline{\nabla_{Nash(X^t)} \mathcal{G}_{m_t}(X^t)} - \nabla_{Nash(X^t)} \mathcal{G}(X^t)\|_2 + \|\nabla_{Nash(X^t)} \mathcal{G}(X^t) - \nabla_{Nash(X^t)} \mathcal{G}(X^{PS})\|_2,
\end{aligned} \tag{20}$$

where $m_t$ is the number of samples each player collects at time step $t$, $\overline{\nabla_{Nash(X^t)} \mathcal{G}_{m_t}(X^t)}$ represents the average of individual gradients at the joint decision $Nash(X^t)$, computed over $m_t$ collected samples.

From Theorem 2 and Definition 3, we obtain: $\|\nabla_{Nash(X^t)} \mathcal{G}(X^t) - \nabla_{Nash(X^t)} \mathcal{G}(X^{PS})\|_2 \leq \sqrt{\sum_{i=1}^{n} \hat{\varepsilon}_i^2} \|X^t - X^{PS}\|_2$. According to Lemma 1, we have $\|\overline{\nabla_{Nash(X^t)} \mathcal{G}_{m_t}(X^t)} - \nabla_{Nash(X^t)} \mathcal{G}(X^t)\|_2 \leq \sqrt{\sum_{i=1}^{n} \hat{\varepsilon}_i^2} \delta$, with probability at least $F_{\chi^2(d)}(\frac{m_t \sum_{i=1}^{n} \hat{\varepsilon}_i^2 \delta^2}{\sigma_t})$. Combining these two inequalities with Eq. (20), when $\|X^t - X^{PS}\|_2 > \delta$ and $\alpha > \sqrt{\sum_{i=1}^{n} \hat{\varepsilon}_i^2}$, with probability at least $F_{\chi^2(d)}(\frac{m_t \sum_{i=1}^{n} \hat{\varepsilon}_i^2 \delta^2}{\sigma_t})$, we have:

$$\left\| \overline{\nabla_{Nash(X^t)} \mathcal{G}_{m_t}(X^t)} - \nabla_{Nash(X^t)} \mathcal{G}(X^{PS}) \right\|_2 \leq 2\sqrt{\sum_{i=1}^{n} \hat{\varepsilon}_i^2} \|X^t - X^{PS}\|_2. \tag{21}$$

$X^{PS}$ and $Nash(X^t)$ are the Nash equilibria of $\mathcal{G}(X^{PS})$ and $\mathcal{G}(X^t)$, respectively, and $\mathcal{G}(\cdot)$ is monotone. According to the first-order optimality condition (Bubeck et al., 2015) for an $\alpha$-strongly monotone $\mathcal{G}(\cdot)$, we

multiply both sides of Eq. (21) by $\|Nash(X^t) - X^{PS}\|_2$ and apply Cauchy-Schwarz inequality (Hunter & Nachtergaele, 2001), when $\alpha > \sqrt{\sum_{i=1}^{n} \hat{\varepsilon}_i^2}$, with probability at least $F_{\chi^2(d)}(\frac{m_t \sum_{i=1}^{n} \hat{\varepsilon}_i^2 \delta^2}{\sigma_t})$, we have:

$$\|Nash(X^t) - X^{PS}\|_2 \leq \frac{2\sqrt{\sum_{i=1}^{n} \hat{\varepsilon}_i^2}}{\alpha} \|X^t - X^{PS}\|_2. \tag{22}$$

Since the distribution maps $\mathcal{D}_i(X)$ are unknown, the sample sets obtained after decision deployment cannot be anticipated. To obtain a practically feasible result, we assume the covariance matrices and the corresponding eigenvalues are independent across time steps under unknown distribution maps $\mathcal{D}_i(X)$. By Theorem 2, if $\alpha > 2\sqrt{\sum_{i=1}^{n} \hat{\varepsilon}_i^2}$, then with probability at least $\prod_{\mu=1}^{t} \left( F_{\chi^2(d)}(\frac{m_\mu \sum_{i=1}^{n} \hat{\varepsilon}_i^2 \delta^2}{\sigma_\mu}) \right)$, we have:

$$\|X^t - X^{PS}\|_2 \leq \delta, \ \forall \ t \geq \frac{\log\left( \frac{\delta}{\|X^0 - X^{PS}\|_2} \right)}{\log\left( \frac{2\sqrt{\sum_{i=1}^{n} \hat{\varepsilon}_i^2}}{\alpha} \right)}. \tag{23}$$

∎

Theorem 3 guarantees the convergence to performatively stable equilibria with a probability. This guarantee is established by Lemma 1, which applies the Central Limit Theorem. The convergence can be achieved when $\alpha > 2\sqrt{\sum_{i=1}^{n} \hat{\varepsilon}_i^2}$, a condition readily satisfied by adding an convex regularizer to each loss function in a monotone game based on measured $\hat{\varepsilon}_i$, given sufficient sample size $m_t$ in each iteration. This result, therefore, informs practical algorithm design in the next section.

## 4 Algorithm

Based on the theoretical results, we propose the **S**ensitivity-**I**nformed **R**epeated **R**etraining (SIR$^2$) algorithm (Algorithm 1) to achieve performatively stable equilibria in decision-dependent games. SIR$^2$ iteratively optimizes each player's decision by adapting to the data distribution maps $\mathcal{D}_i(X)$, using a gradient-based $\hat{\varepsilon}_i$-sensitivity measure (Definition 3) to ensure convergence. By setting $\gamma = \max(0, c\sqrt{\sum_{i=1}^{n} \hat{\varepsilon}_i^2} - \psi)$ with $c > 2$, SIR$^2$ tunes the strong monotonicity parameter based on estimated $\hat{\varepsilon}_i$, which overcomes impractical $\beta$-smoothness assumptions of prior work and guarantees convergence across arbitrary distribution maps.

Specifically, in each iteration $t$, each player $i \in [n]$ first updates $\gamma$ based on the latest $\hat{\varepsilon}_i$ estimates (line 4). Then, for the monotone game $\mathcal{G} := (\mathbb{E}\ell_1(x_1, x_{-1}, Z_1), \cdots, \mathbb{E}\ell_n(x_n, x_{-n}, Z_n))$, each player $i$ adds a regularization term $\frac{\gamma}{2}\|x_i\|_2^2$ to its loss function, constructing an $\alpha$-strongly monotone game $\bar{\mathcal{G}} := (\mathbb{E}\bar{\ell}_1(x_1, x_{-1}, Z_1), \cdots, \mathbb{E}\bar{\ell}_n(x_n, x_{-n}, Z_n))$ with $\alpha = \psi + \gamma$ (line 5). Following this, each player $i$ calculates its decision $x_i^t$ in a Nash equilibrium under the game $\bar{\mathcal{G}}(X^{t-1})$ induced by $X^{t-1}$ (line 6). This step is achieved by solving the Nash equilibrium in time step $t$. For the strongly monotone game $\bar{\mathcal{G}}(X^{t-1})$, this computation is tractable using numerical methods, such as gradient-based optimization. After deploying the joint decision $X^t = (x_1^t, x_2^t, \cdots, x_n^t)^T$ (line 7), each player $i$ collects samples $Z_i^t$ from the distribution induced by the joint decision $X^t$ (line 8) and estimates $\hat{\varepsilon}_i$ (line 9). The procedure repeats until it converges to a performatively stable equilibrium.

By doing so, our algorithm ensures that the game is $\alpha$-strongly monotone with $\alpha > 2\sqrt{\sum_{i=1}^{n} \hat{\varepsilon}_i^2}$, satisfying the convergence condition of Theorem 3. Therefore, our SIR$^2$ algorithm converges to a performatively stable equilibrium $X^{PS}$ with finite samples at the convergence rate and probability specified in Theorem 3. Note that our algorithm guarantees the convergence to the performatively stable equilibrium of the regularized game $\bar{\mathcal{G}}$. According to our Theorem 3, the repeated retraining in the original game $\mathcal{G}$ may fail to converge to a performatively stable equilibrium, when the strong monotonicity parameter $\psi$ is small (A detailed example is provided in Appendix H). Since the performatively stable equilibrium may differ from the equilibrium of the original game, we aim to ensure that this equilibrium is as close as possible to the equilibrium of the original game. Theorem 4 proves that this quadratic regularizer $R(x)$ used in our Algorithm 1 minimizes

---

**Algorithm 1:** Sensitivity-Informed Repeated Retraining (SIR$^2$): Procedures for Player $i \in [n]$

---

1: **Input:** Parameter $c \in (2, +\infty)$, initial sensitivity $\hat{\varepsilon}_i > 0$,
   initial empty joint decision $X^0 = \varnothing$,
   initial time step $t = 1$, initial dataset $Z_i^0 \sim \mathcal{D}_i(X^0)$,
   $\psi$-strongly monotone game $\mathcal{G} := (\mathbb{E}\ell_1(x_1, x_{-1}, Z_1), \cdots, \mathbb{E}\ell_n(x_n, x_{-n}, Z_n))$
2: **Output:** The joint decision at performatively stable equilibrium $X^{PS}$
3: **while** not converged **do**
4: $\quad \gamma \leftarrow \max(0, c\sqrt{\sum_{i=1}^n \hat{\varepsilon}_i^2} - \psi)$;
5: $\quad \bar{\ell}_i(x_i, x_{-i}, Z_i) \leftarrow \ell_i(x_i, x_{-i}, Z_i) + \frac{\gamma}{2}\|x_i\|_2^2$;
6: $\quad x_i^t \leftarrow \arg\min_{x_i} \mathbb{E}_{Z_i^{t-1} \sim \mathcal{D}_i(X^{t-1})} \bar{\ell}_i(x_i, x_{-i}, Z_i^{t-1})$;
7: $\quad X^t \leftarrow (x_1^t, x_2^t, \cdots, x_n^t)^T$;
8: $\quad$ Collect data $Z_i^t \sim \mathcal{D}_i(X^t)$;
9: $\quad \hat{\varepsilon}_i \leftarrow \max\left(\hat{\varepsilon}_i, \frac{\left\|\mathbb{E}_{Z_i^t \sim \mathcal{D}_i(X^t)} \nabla_{x_i^t} \ell_i(x_i^t, x_{-i}^t, Z_i^t) - \mathbb{E}_{Z_i^{t-1} \sim \mathcal{D}_i(X^{t-1})} \nabla_{x_i^t} \ell_i(x_i^t, x_{-i}^t, Z_i^{t-1})\right\|_2}{\|X^t - X^{t-1}\|_2}\right)$;
10: $\quad t \leftarrow t + 1$;
11: **end while**
12: $X^{PS} \leftarrow X^t$;

---

the upper bound of the distance between the original and regularized equilibria among all $\gamma$-strongly convex regularizers[4].

*Theorem 4. Let $\mathcal{R}$ be the family of all $\gamma$-strongly convex regularizers. The distance $\|X^* - X^R\|_2$ has an upper bound that depends on the regularizer $R(\cdot) \in \mathcal{R}$, where $X^*$ and $X^R$ are the Nash equilibria of the original and regularized games, respectively. Among all choices of $R(\cdot) \in \mathcal{R}$, the quadratic regularizer $R(x) = \frac{\gamma}{2}\|x\|_2^2$ minimizes this upper bound.*

## 5 Experiments

In this section, we evaluate the performance of our proposed SIR$^2$ algorithm[5] in three multi-agent decision-dependent games: prediction error minimization between prediction platforms, Cournot competition in crude oil trade, and revenue maximization in ride-share markets.

To the best of our knowledge, five methods have been proposed for decision-dependent games in published works: Repeated Retraining (RR) (Narang et al., 2023), Repeated Gradient Descent (RGD) (Narang et al., 2023), Stochastic Forward-Backward (SFB) method (Cutler et al., 2024), Adaptive Gradient Method (AGM) (Narang et al., 2023), and Online Performative Gradient Descent (OPGD) (Zhu et al., 2023). We include all of them in our performance comparison. For our SIR$^2$, we set $c = 2.1$, to ensure that $\alpha > 2\sqrt{\sum_{i=1}^n \hat{\varepsilon}_i^2}$ (Theorem 3). In fact, our method is robust to the key parameter $c$, which controls the convergence rate to performatively stable equilibria. The sensitivity analysis of this parameter is presented in Appendix J.1. The performance comparison over wall-clock time is provided in Appendix J.2. More details on experimental settings can be found in Appendix J.3.

### 5.1 Prediction Error Minimization Game

#### 5.1.1 Game Abstraction

In the prediction error minimization game, platforms compete to make predictions that can shape the distribution they aim to predict, thereby affecting their own prediction accuracy. For example, in election forecasting, the prediction released by public media can lead to shifts in the polling of candidates (features)

---

[4]The proof of Theorem 4 is provided in Appendix I.
[5]The source code of our method is available at: 'anonymous.4open.science/r/multiplayer-performative-stable'

Table 1: Sum RMSE comparison for SIR$^2$ and five baselines on the Prediction Error Minimization Game, with $\sigma_{a_i}^2$ as the parameter in $\mathcal{D}_i(X)$. For each method, the average result and standard deviation over 10 trials are reported. The best result on each setting is highlighted in bold.

| | $\sigma_{a_i}^2 = 2.5$ | $\sigma_{a_i}^2 = 5.0$ | $\sigma_{a_i}^2 = 7.5$ | $\sigma_{a_i}^2 = 10.0$ |
|---|---|---|---|---|
| SIR$^2$ | **0.0272 ± 0.0021** | **0.0272 ± 0.0021** | **0.0272 ± 0.0021** | **0.0272 ± 0.0021** |
| RR | 0.1022 ± 0.0713 | 0.1145 ± 0.0411 | 0.1195 ± 0.0330 | 0.1229 ± 0.0370 |
| RGD | 0.2381 ± 0.0657 | 0.1735 ± 0.0184 | 0.1712 ± 0.0175 | 0.1709 ± 0.0175 |
| SFB | 0.2507 ± 0.0855 | 0.1752 ± 0.0210 | 0.1837 ± 0.0304 | 0.2187 ± 0.0845 |
| AGM | 4.3717 ± 3.4092 | 3.6357 ± 2.0388 | 3.8324 ± 2.1366 | 3.9956 ± 2.1163 |
| OPGD | 0.9663 ± 0.1773 | 0.4428 ± 0.0860 | 0.3926 ± 0.0871 | 0.3280 ± 0.1003 |

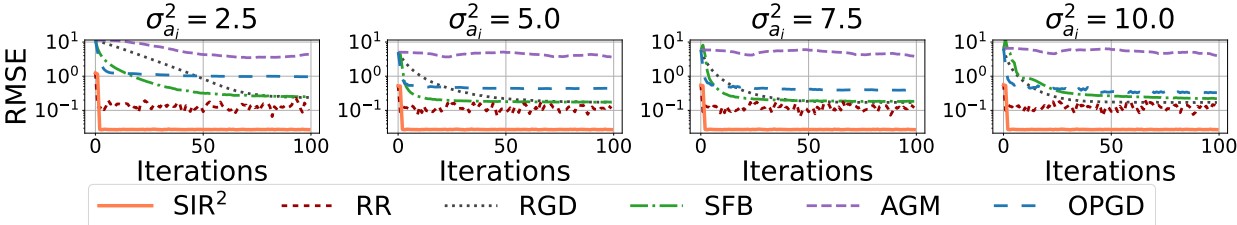

Figure 1: Sum RMSE comparison for SIR$^2$ and five baselines on the Prediction Error Minimization Game over iterations.

and the final election results (ground truth output). In our experiment, we consider two platforms, i.e., $n = 2$, in this game. We generate $m = 100$ data samples $\theta_i \sim \mathcal{N}_{10}(0, 0.1)$ and the corresponding outputs $z_i = \theta_i^T b_i$, where $b_i \sim \mathcal{N}_{10}(0, 0.1)$. Each platform $i \in [2]$ seeks to predict $z_i$ by solving a regression problem, i.e., minimizing the loss function: $\ell_i(x_i, z_i) = \frac{1}{m}\|z_i - \theta_i^T x_i\|_2^2$. This loss function is convex but not strongly convex in $x_i$, which implies that its strong convexity parameter is 0. Since this game is separable (i.e., $\ell_i(Z_i, x_i, x_{-i}) = \ell_i(Z_i, x_i)$ (Narang et al., 2023)), the strong monotonicity parameter of this game is the minimum of the strong convexity parameters of the loss functions, and this value is $\psi = 0$. We employ the `sklearn.linear\_model.Ridge` to solve the regression problems in each iteration. The performative effect shifts features and outputs, modeled by the data distribution map $\mathcal{D}_i(X)$ (Perdomo et al., 2020; Narang et al., 2023; Liu et al., 2024):

$$\theta_{i\ X} \sim \mathcal{D}_{\theta_i}(X) = \theta_i + \sum_{i=1}^{n} C_i x_i + u_i, \text{ and } z_{i\ X} \sim \mathcal{D}_{z_i}(X) = \theta_i^T b_i + a_i^T x_i + a_{-i}^T x_{-i} + w_i, \tag{24}$$

where $C_i \sim \mathcal{N}_{10 \times 10}(0, \sigma_c^2)$ is the matrix controlling feature shifts induced by decision $x_i$, $a_i \sim \mathcal{N}_{10}(0, \sigma_{a_i}^2)$ and $a_{-i} \sim \mathcal{N}_{10}(0, \sigma_{a_{-i}}^2)$ are the vectors controlling output shifts induced by the decisions $x_i$ and $x_{-i}$, respectively, and $u_i \sim \mathcal{N}_{10}(0, 0.01)$ and $w_i \sim \mathcal{N}(0, 0.01)$ are noise terms. In our experiment, we set $\sigma_{a_i}^2 = \{2.5, 5.0, 7.5, 1.0\}$, $\sigma_{a_{-i}}^2 = 12.5 - \sigma_{a_i}^2$, and $\sigma_c^2 = \sigma_{a_i}^2/n$ to simulate varying performative effects on features $\theta_i$ and outputs $z_i$ (higher $\sigma_{a_i}^2$ indicates greater sensitivity).

### 5.1.2 Result Analysis

Table 1 reports the sum of root mean square error (RMSE) across two platforms (i.e., $RMSE_{\text{platform 1}} + RMSE_{\text{platform 2}}$) for each method. The total RMSE of all methods over 100 iterations is shown in Fig. 1. The standard deviations of the total RMSE for all methods are provided in Fig. 8 in J.4. Our method, SIR$^2$, converges to a performatively stable equilibrium within 5 iterations and achieves the lowest RMSE ($< 0.03$) across all settings. In contrast, AGM and OPGD, which target Nash equilibria, fail to converge to their goals. A key reason is that these methods perform gradient descent using the estimated gradient of performative loss, although the performative loss may not be convex. We apply the closed-form solution $\frac{z_i}{\alpha}$ to minimize the loss function in each iteration. Consequently, they may converge to local optima where the decisions of two

Table 2: Total revenue comparison for $\text{SIR}^2$ and five baselines in the crude oil trade, with $\mu$ as the parameter in $\mathcal{D}_i(X)$. For each method, the average result and standard deviation over 10 trials are reported. The best result on each setting is highlighted in bold.

| | $\mu_A = 0.25$ | $\mu_A = 0.5$ | $\mu_A = 0.75$ | $\mu_A = 1.0$ |
|---|---|---|---|---|
| $\text{SIR}^2$ | $\mathbf{6.18 \times 10^{11} \pm 1.64 \times 10^9}$ | $\mathbf{5.60 \times 10^{11} \pm 2.63 \times 10^9}$ | $\mathbf{5.04 \times 10^{11} \pm 3.39 \times 10^9}$ | $\mathbf{4.48 \times 10^{11} \pm 3.77 \times 10^9}$ |
| RR | $8.50 \times 10^{10} \pm 2.38 \times 10^8$ | $7.78 \times 10^{10} \pm 1.82 \times 10^8$ | $7.09 \times 10^{10} \pm 2.29 \times 10^8$ | $6.44 \times 10^{10} \pm 1.97 \times 10^8$ |
| RGD | $1.18 \times 10^{11} \pm 8.68 \times 10^8$ | $1.04 \times 10^{11} \pm 6.45 \times 10^8$ | $9.17 \times 10^{10} \pm 9.93 \times 10^8$ | $7.85 \times 10^{10} \pm 8.36 \times 10^8$ |
| SFB | $2.35 \times 10^{11} \pm 4.19 \times 10^9$ | $1.97 \times 10^{11} \pm 2.97 \times 10^9$ | $1.62 \times 10^{11} \pm 4.88 \times 10^9$ | $1.24 \times 10^{11} \pm 4.66 \times 10^9$ |
| AGM | $4.33 \times 10^{11} \pm 1.06 \times 10^{11}$ | $4.50 \times 10^{11} \pm 9.35 \times 10^{10}$ | $2.76 \times 10^{11} \pm 1.48 \times 10^{11}$ | $2.35 \times 10^{11} \pm 1.91 \times 10^{11}$ |
| OPGD | $3.62 \times 10^{11} \pm 6.87 \times 10^{10}$ | $3.56 \times 10^{11} \pm 3.67 \times 10^{10}$ | $2.81 \times 10^{11} \pm 8.91 \times 10^{10}$ | $2.08 \times 10^{11} \pm 6.99 \times 10^{10}$ |

platforms are not mutually optimal. Our method performs superiorly because it leverages the $\hat{\varepsilon}_i$-sensitivity measure to adapt dynamically to $\mathcal{D}_i(X)$, bypassing $\beta$-smoothness assumption and the high sample complexity of $\mathcal{W}_1$ distance estimation, thus ensuring rapid and robust convergence with consistently low error. This adaptability allows our method to achieve less than 0.03 in total RMSE across various $\sigma_{a_i}^2$.

## 5.2 Cournot Competition in Crude Oil Trade

### 5.2.1 Game Abstraction

In the Cournot competition, $n = 28$ crude oil exporting countries, covering 99% of 2021 global exports, adjust crude oil export quantities to maximize revenue. These countries can only adjust their export quantities, and the price per barrel is determined by the global market. Utilizing 2021 crude oil trade data[6], we denote each country $i$'s export quantity as $q_i$, and we use an uniform cost $c = \$10$ per barrel across all countries. The crude oil price follows a publicly known linear inverse demand function $P_{z \sim \mathcal{D}(X)} = z - b \sum_{i=1}^{n}(q_i + x_i)$, where $x_i$ is the decision representing export adjustment for country $i$, $q_i + x_i$ is the resulting export quantity, $z \sim \mathcal{D}(X)$ is a demand intercept parameter influenced by the joint decision $X$, and $b$ quantifies the price sensitivity to the total supply. Each country $i$ minimizes its expected loss by adjusting exports $x_i$: $\ell_i(x_i, x_{-i}, z) = c \cdot (q_i + x_i) - (q_i + x_i) \cdot (z - b \sum_{j=1}^{n}(q_j + x_j))$, where $c \cdot (q_i + x_i)$ is the cost of country $i$, and $(q_i + x_i) \cdot (z - b \sum_{j=1}^{n}(q_j + x_j))$ is the revenue of country $i$. The strong monotonicity parameter of the game is the smallest eigenvalue of the Jacobian matrix $\mathcal{J}$ of the individual gradient $\nabla_X \mathcal{G}(Y)$ (Rockafellar & Wets, 1998). The eigenvalues of $\mathcal{J}$ in this game are $b(n-1)$ and $b$. Given that $n = 28$ in our experiment, the smallest eigenvalue is $b$, which implies that $\psi = b$. The data distribution map $\mathcal{D}(X)$, which is identical across all countries, implies that adjustments in total exports induce a nonlinear transformation of the price function $P(X)$:

$$z \sim \mathcal{D}(X) = z_0 - \mu \cdot \sinh^{-1}\left(\sum_{i=1}^{n} x_i\right) + w_i, \tag{25}$$

where $z_0 = 147.27$ is a base price representing the historic peak crude oil price reached in July 2008, $\mu$ controls the sensitivity of the data distribution map $\mathcal{D}(X)$, $\sinh^{-1}(\cdot)$ denotes the inverse hyperbolic sine transformation, and $w_i \sim \mathcal{N}(0, 0.01)$ is the noise term. In our experiment, we set $\mu = \{0.25, 0.5, 0.75, 1.0\}$ to simulate different performative effects on the price function $P(X)$, where larger values of $\mu$ correspond to higher sensitivities. For further details, please refer to Appendix J.5.

### 5.2.2 Result Analysis

Table 2 presents the total revenue of 28 countries across all methods in the Cournot competition. The results of all methods over 100 iterations are shown in Fig. 2, with standard deviations provided in Appendix J.6 (Fig. 9). Our $\text{SIR}^2$ algorithm outperforms all baseline methods in revenue and converges to a performatively stable equilibrium within 10 iterations, while RR, RGD, and SFB struggle in reaching equilibrium under shifted distributions due to reliance on impractical assumptions (e.g., $\beta$-joint smoothness). AGM and OPGD, despite attempting to learn data distribution maps, assume linear maps. As a result, they fail to capture

---

[6]The crude oil trade dataset is available at 'kaggle.com/datasets/toriqulstu/global-crude-petroleum-trade-1995-2021'.

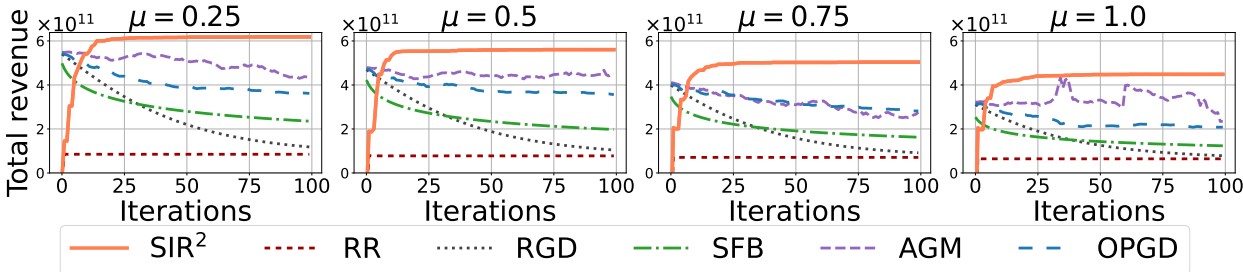

Figure 2: Total revenue comparison for $SIR^2$ and five baselines on the crude oil trade over iterations.

Table 3: Total revenue comparison for $SIR^2$ and five baselines on the Ride-Share Markets, with $\mu_A$ as the parameter in $\mathcal{D}_i(X)$. For each method, the average result and standard deviation over 10 trials are reported. The best result on each setting is highlighted in bold.

|        | $\mu_A = 0.25$ | $\mu_A = 0.5$ | $\mu_A = 0.75$ | $\mu_A = 1.0$ |
|--------|----------------|---------------|----------------|---------------|
| $SIR^2$ | **228348 ± 664** | **219390 ± 723** | **217054 ± 954** | **214854 ± 780** |
| RR     | 207515 ± 2713  | 135764 ± 4790 | 84074 ± 5266   | 49624 ± 3890  |
| RGD    | 176232 ± 4046  | 99141 ± 2923  | 65041 ± 2262   | 45986 ± 1860  |
| SFB    | 212102 ± 3375  | 157131 ± 4805 | 118173 ± 4942  | 92016 ± 4990  |
| AGM    | 186364 ± 2964  | 148872 ± 5652 | 151494 ± 9386  | 157927 ± 9953 |
| OPGD   | 160929 ± 2007  | 129170 ± 2337 | 117443 ± 1806  | 113418 ± 1453 |

complex real-world distribution dynamics, leading to oscillatory results. The detailed exporting quantities and prices achieved by all methods are provided in Figs. 10 and 11, respectively, in Appendix J.6.

### 5.3 Revenue Maximization Game in Ride-Share Markets

#### 5.3.1 Game Abstraction

In the revenue maximization game, companies set optimal prices to maximize revenue, where each company's price decisions influence both its own demand and that of its competitors. Specifically, a company that raises its prices reduces demand for its own services while increasing demand for competitors' services. In this game, two companies ($n = 2$), Lyft and Uber, set price adjustments to maximize revenue in 11 Boston locations[7]. Each company $i$ seeks to maximize its revenue $z_i^T x_i$ in the price interval by minimizing the regularized loss function: $\ell_i(x_i, z_i) = -z_i^T x_i + \frac{\alpha}{2}\|x_i\|_2^2$, where $\alpha > 0$ is the regularization parameter. For this separable game with $\alpha$-strongly convex loss functions in each $x_i$, the strong monotonicity parameter is $\psi = \alpha$. The data distribution map is designed based on the logistic demand function (Bowerman et al., 2003; Phillips, 2021):

$$z_i \sim \mathcal{D}_i(X) = \frac{2z_{0\ i}}{1 + e^{-(A_i x_i + A_{-i} x_{-i})}} + w_i, \tag{26}$$

where $z_{0\ i}$ is the initial demand of company $i$, $A_i, A_{-i} \in \mathbb{R}^{11 \times 11}$, and $w_i \sim \mathcal{N}_{11}(0, 0.01)$ is the noise term. We generate matrices $A_i$ and $A_{-i}$ as $A_i = diag(\mathcal{N}_{11}(-\mu_A, (\frac{\mu_A}{5})^2)) + E$ and $A_{-i} = diag(\mathcal{N}_{11}(\frac{\mu_A}{2}, (\frac{\mu_A}{10})^2)) + E$, respectively, where $E_{ij} \sim \mathcal{N}(0, 0.01)$, and the diagonal entries are truncated to ensure $A_i \preceq 0$ and $A_{-i} \succeq 0$. In our experiment, we vary $\mu_A = \{0.25, 0.5, 0.75, 1.0\}$, increasing the sensitivity as $\mu_A$ grows. For further details, please refer to Appendix J.7.

#### 5.3.2 Result Analysis

Table 3 summarizes the total revenue of two companies across all methods in the revenue maximization game. Consistent with the results in Section 5.1, our method outperforms baselines across all values of $\mu_A$,

---

[7]The Lyft and Uber Dataset is available at 'kaggle.com/datasets/brllrb/uber-and-lyft-dataset-boston-ma'.

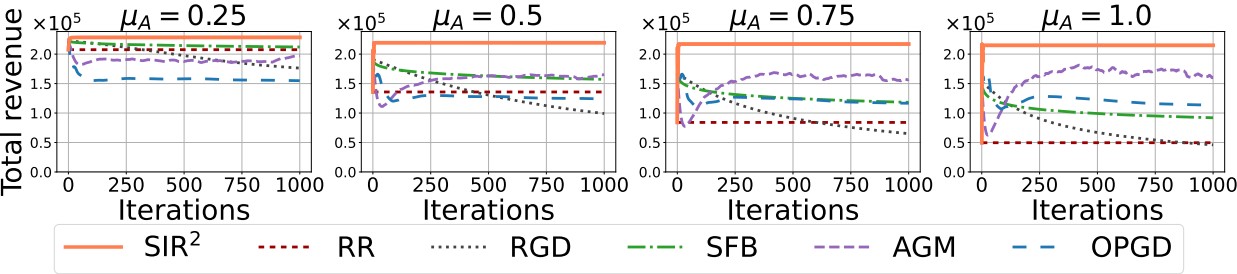

Figure 3: Total revenue comparison for $SIR^2$ and five baselines on the Ride-Share Markets over iterations.

achieving revenue gains of at least $\$16,246$ (at $\mu_A = 0.25$) and up to $\$65,560$ (at $\mu_A = 0.75$), demonstrating its effectiveness in adapting to $\mathcal{D}_i(X)$. Fig. 3 shows total revenues of two companies over 1000 iterations, with standard deviations in Appendix J.8 (Fig. 12). While initial revenue fluctuates due to the adaptation to $\mathcal{D}_i(X)$, our method converges to a performatively stable equilibrium within 10 iterations and maintains consistently high revenue. Our method achieves higher revenue primarily due to an appropriate strategy: it sets a slightly lower price (5%) than baselines. This strategy stimulates greater demand, ultimately outperforming other methods in total revenue. The detailed revenues, demands, and prices of each company achieved by all methods are provided in Figs. 13-15, respectively, in Appendix J.8.

## 6 Conclusion

In this paper, we propose a framework for decision-dependent games with decision-induced distribution shifts. By introducing a gradient-based $\hat{\varepsilon}_i$-sensitivity measure for $\mathcal{D}_i(X)$, we quantify decision-induced distribution shifts, relaxing the impractical $\beta$-smoothness assumption. Based on the definition, we derive convergence guarantees to performatively stable equilibria, which hold across arbitrary data distribution maps under practically feasible conditions, even with finite samples. Building upon the theoretical results, we propose an $SIR^2$ algorithm, which adapts loss functions using calculated $\hat{\varepsilon}_i$-sensitivity, so as to achieve rapid convergence. Empirical evaluations support our theoretical findings and demonstrate the superiority of $SIR^2$ in losses and convergence speed over baselines in the prediction error minimization game, the Cournot competition, and the revenue maximization game. The proposed framework offers insights for multi-agent learning in dynamic environments, such as reinforcement learning (Mandal et al., 2023) and federated learning (Jin et al., 2024).

The current version of our approach is potentially limited by: (1) the strong monotonicity condition, which may not hold in complex games with multiple Nash equilibria, and (2) the need for large sample sizes to guarantee the convergence with high probability in high-dimensional settings. To address these issues, in our future work, we will extend the framework to non-strongly monotone games by incorporating the analysis of mixed strategies and will enhance convergence guarantees for small sample sizes with alternative statistical tools, such as Chebyshev's inequality (Ferentios, 1982).

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

# Appendices

## A  An Example of Unverifiable $\beta$-smoothness

The $\beta$-smoothness condition (referring to individual $\beta_i$-smoothness for each loss function) applied in current research (Narang et al., 2023; Zhu et al., 2023; Cutler et al., 2024) assumes that for each player $i \in [n]$, its loss function gradient is $\beta_i$-Lipschitz continuous with respect to the data $z_i$ and decision $x_i$. Specifically, each loss function $\ell_i(x_i, x_{-i}, Z_i)$ is $\beta_i$-smooth if, $\forall x_i, x_i' \in \mathcal{X}_i$ and $\forall z_{i_j}, z_{i_j}' \in \mathcal{Z}_i$, $\mathcal{Z}_i := \cup_{X \in \mathcal{X}} \sup(\mathcal{D}_i(X))$, the following holds:

$$
\begin{aligned}
\|\nabla_{x_i} \ell_i(x_i, x_{-i}, z_{i_j}) - \nabla_{x_i} \ell_i(x_i, x_{-i}, z_{i_j}')\|_2 &\leq \beta_i \|z_{i_j} - z_{i_j}'\|_2, \\
\|\nabla_{x_i} \ell_i(x_i, x_{-i}, z_{i_j}) - \nabla_{x_i} \ell_i(x_i', x_{-i}, z_{i_j})\|_2 &\leq \beta_i \|x_i - x_i'\|_2.
\end{aligned}
\tag{27}
$$

Since $\mathcal{D}_i(X)$ is typically unknown, quantifying the extent of distribution shifts to verify $\beta_i$ is challenging. Here, we use the 1-dimensional case of Prediction Error Minimization Game (Section 5.1) as a practical example: Each platform $i \in [n]$ seeks to predict outcome $z_{i_j} \in \mathbb{R}$ with feature $\theta_{i_j} \in \mathbb{R}$ by solving a regression problem, i.e., minimizing the loss function: $\ell_i(x_i, z_i, \theta_i) = \mathbb{E}_{\{z_i, \theta_i\} \sim \mathcal{D}_i(X)} \|z_i - \theta_i x_i\|_2^2$, where $\{z_i, \theta_i\}$ denotes the dataset for platform $i$, and $x_i \in \mathbb{R}$ is the decision of platform $i$.

According to the definition of $\beta$-smoothness in Eq. (27), we have:

$$
\begin{aligned}
\| - 2(z_{i_j} - \theta_{i_j} x_i)\theta_{i_j} + 2(z_{i_j}' - \theta_{i_j} x_i)\theta_{i_j}\|_2 &= \| - 2\theta_{i_j}\|_2 \|z_{i_j} - z_{i_j}'\|_2, \\
\| - 2(z_{i_j} - \theta_{i_j} x_i)\theta_{i_j} + 2(z_{i_j} - \theta_{i_j}' x_i)\theta_{i_j}'\|_2 &= \| - 2z_{i_j} + 2(\theta_{i_j} + \theta_{i_j}')x_i\|_2 \|\theta_{i_j} - \theta_{i_j}'\|_2, \\
\| - 2(z_{i_j} - \theta_{i_j} x_i)\theta_{i_j} + 2(z_{i_j} - \theta_{i_j} x_i')\theta_{i_j}\|_2 &= \|2\theta_{i_j}^2\|_2 \|x_i - x_i'\|_2,
\end{aligned}
\tag{28}
$$

where $-2(z_{i_j} - \theta_{i_j} x_i)\theta_{i_j}$ is the loss function gradient $\nabla_{x_i} \ell_i(x_i, z_{i_j}, \theta_{i_j})$.

In this example, $\beta_i$ must satisfy: $\beta_i \geq \max(\| - 2\theta_{i_j}\|_2, \| - 2z_{i_j} + 2(\theta_{i_j} - \theta_{i_j}')x_i\|_2, \|2\theta_{i_j}^2\|_2)$. However, when the data distribution map $\mathcal{D}_i(X)$ is unknown, the upper bound of $\| - 2\theta_{i_j}\|_2, \| - 2z_{i_j} + 2(\theta_{i_j} + \theta_{i_j}')x_i\|_2, \|2\theta_{i_j}^2\|_2$ cannot be determined. Consequently, the value of $\beta_i$ can not be verified in this case.

## B  Proof of Corollary 1

Given an $\alpha$-strongly monotone game $\mathcal{G}(\cdot)$ (as defined in Eq. (11)) and the definition of $Nash(X)$ in Eq. (5), we have, for all $X, X' \in \mathcal{X}$:

$$
\alpha \|Nash(X) - Nash(X')\|_2^2 \leq (\nabla_{Nash(X)} \mathcal{G}(X) - \nabla_{Nash(X')} \mathcal{G}(X)))^T (Nash(X) - Nash(X')).
\tag{29}
$$

By the first-order optimality conditions (Bubeck et al., 2015), for all $X, X' \in \mathcal{X}$,

$$
\begin{aligned}
\nabla_{Nash(X)} \mathcal{G}(X)^T (Nash(X) - Nash(X')) &\leq 0, \\
\nabla_{Nash(X')} \mathcal{G}(X')^T (Nash(X) - Nash(X')) &\geq 0,
\end{aligned}
\tag{30}
$$

and thus we have:

$$
\nabla_{Nash(X)} \mathcal{G}(X)^T (Nash(X) - Nash(X')) \leq \nabla_{Nash(X')} \mathcal{G}(X')^T (Nash(X) - Nash(X')).
\tag{31}
$$

Combining Eqs. (29) and (31) and applying the Cauchy-Schwarz inequality (Hunter & Nachtergaele, 2001), we obtain:

$$
\begin{aligned}
&\alpha \|Nash(X) - Nash(X')\|_2^2 \\
&\leq (\nabla_{Nash(X)} \mathcal{G}(X) - \nabla_{Nash(X')} \mathcal{G}(X))^T (Nash(X) - Nash(X')) \\
&\leq (\nabla_{Nash(X')} \mathcal{G}(X') - \nabla_{Nash(X')} \mathcal{G}(X))^T (Nash(X) - Nash(X')) \\
&\leq \|\nabla_{Nash(X')} \mathcal{G}(X') - \nabla_{Nash(X')} \mathcal{G}(X)\|_2 \|Nash(X) - Nash(X')\|_2 \\
&\implies \alpha \|Nash(X) - Nash(X')\|_2 \leq \|\nabla_{Nash(X')} \mathcal{G}(X') - \nabla_{Nash(X')} \mathcal{G}(X)\|_2.
\end{aligned}
\tag{32}
$$

From the definition of individual gradients (Definition 4) and that of the $\hat{\varepsilon}_i$-sensitivity (Definition 3), for all $Y, X, X' \in \mathcal{X}_{RR}$, we have:

$$
\begin{aligned}
&\|\nabla_Y \mathcal{G}(X) - \nabla_Y \mathcal{G}(X')\|_2^2 \\
&= \sum_{i=1}^n \left\| \underset{Z_i \sim \mathcal{D}_i(X)}{\mathbb{E}} \nabla_{y_i} \ell_i(y_i, y_{-i}, Z_i) - \underset{Z_i' \sim \mathcal{D}_i(X')}{\mathbb{E}} \nabla_{y_i} \ell_i(y_i, y_{-i}, Z_i') \right\|_2^2 \\
&\leq \sum_{i=1}^n \hat{\varepsilon}_i^2 \|X - X'\|_2^2 \\
&\Longrightarrow \|\nabla_Y \mathcal{G}(X) - \nabla_Y \mathcal{G}(X')\|_2 \leq \sqrt{\sum_{i=1}^n \hat{\varepsilon}_i^2} \|X - X'\|_2.
\end{aligned}
\tag{33}
$$

Since $Y$ can be any joint decision in the set $\mathcal{X}_{RR}$ as stated in Definition 3, without loss of generality, we set $Y = Nash(X')$. Then the inequality in Eq. (33) becomes the following:

$$
\|\nabla_{Nash(X')} \mathcal{G}(X) - \nabla_{Nash(X')} \mathcal{G}(X')\|_2 \leq \sqrt{\sum_{i=1}^n \hat{\varepsilon}_i^2} \|X - X'\|_2.
\tag{34}
$$

Combining Eqs. (32) and (34), we have:

$$
\|Nash(X) - Nash(X')\|_2 \leq \frac{\sqrt{\sum_{i=1}^n \hat{\varepsilon}_i^2}}{\alpha} \|X - X'\|_2.
\tag{35}
$$

Since $X, X'$ can be any joint decisions in the set $\mathcal{X}_{RR}$ as stated in Definition 3, without loss of generality, we set $X = X^t$ and $X' = X^{t-1}$, so that $Nash(X) = X^{t+1}$, $Nash(X') = X^t$. Then, we obtain:

$$
\|X^{t+1} - X^t\|_2 \leq \frac{\sqrt{\sum_{i=1}^n \hat{\varepsilon}_i^2}}{\alpha} \|X^t - X^{t-1}\|_2 \leq \left( \frac{\sqrt{\sum_{i=1}^n \hat{\varepsilon}_i^2}}{\alpha} \right)^t \|X^1 - X^0\|_2.
\tag{36}
$$

∎

## C  Proof of Theorem 2

We first proof that $\{X^t\}$ is a Cauchy sequence when $\alpha > \sqrt{\sum_{i=1}^n \hat{\varepsilon}_i^2}$. For any two positive integers $q$ and $p$ (assume that $q > p$), by repeatedly applying the triangle inequality, we have:

$$
\|X^q - X^p\|_2 \leq \|X^q - X^{p+1}\|_2 + \|X^{p+1} - X^p\|_2 \leq \cdots \leq \sum_{r=p}^{q-1} \|X^{r+1} - X^r\|_2.
\tag{37}
$$

From Corollary 1, we have $\|X^{t+1} - X^t\|_2 \leq \left( \frac{\sqrt{\sum_{i=1}^n \hat{\varepsilon}_i^2}}{\alpha} \right)^t \|X^1 - X^0\|_2$. Combining this with Eq. (37), we get:

$$
\|X^q - X^p\|_2 \leq \sum_{r=p}^{q-1} \|X^{r+1} - X^r\|_2 \leq \sum_{r=p}^{q-1} \left( \frac{\sqrt{\sum_{i=1}^n \hat{\varepsilon}_i^2}}{\alpha} \right)^r \|X^1 - X^0\|_2.
\tag{38}
$$

Let $\mathcal{B} = \frac{\sqrt{\sum_{i=1}^n \hat{\varepsilon}_i^2}}{\alpha}$. Since $\alpha > \sqrt{\sum_{i=1}^n \hat{\varepsilon}_i^2}$, we have $\mathcal{B} < 1$. The sum of the geometric series (Friberg, 2007) is:

$$
\sum_{r=p}^{q-1} \mathcal{B}^r = \frac{\mathcal{B}^p (1 - \mathcal{B}^{q-p})}{1 - \mathcal{B}} \leq \frac{\mathcal{B}^p}{1 - \mathcal{B}}.
\tag{39}
$$

Substituting this into Eq. equation 38, we have:

$$\|X^q - X^p\|_2 \leq \frac{\mathcal{B}^p}{1 - \mathcal{B}} \|X^1 - X^0\|_2. \tag{40}$$

For any $k > 0$, we want to find a number $N$ such that for all $q > p \geq N$, $\|X^q - X^p\|_2 \leq k$. Solving for $p$, we get:

$$\frac{\mathcal{B}^p}{1 - \mathcal{B}} \|X^1 - X^0\|_2 \leq k \iff \mathcal{B}^p \leq \frac{k(1 - \mathcal{B})}{\|X^1 - X^0\|_2}$$

$$\iff p \log \mathcal{B} \leq \log \left( \frac{k(1 - \mathcal{B})}{\|X^1 - X^0\|_2} \right) \iff p \geq \frac{\log \left( \frac{k(1-\mathcal{B})}{\|X^1-X^0\|_2} \right)}{\log \mathcal{B}}. \tag{41}$$

Thus, we can choose any positive $N > \frac{\log \left( \frac{k(1-\mathcal{B})}{\|X^1-X^0\|_2} \right)}{\log \mathcal{B}}$. Since such an $N$ always exists for any $k > 0$, the sequence $\{X^t\}$ is a Cauchy sequence (Lang, 2002).

As a result, the joint decision sequence generated by the repeated retraining procedure, $\{X^t\}$, converges to a single joint decision: $\lim_{t \to \infty} X^t = X^{t \to \infty}$. This implies that the difference between neighbouring joint decisions becomes negligible: $\lim_{t \to \infty} \|X^{t+1} - X^t\|_2 = 0$.

Since Eq. (35) holds for any joint decisions $X, X' \in \mathcal{X}_{RR}$ and $\mathcal{X}_{RR}$ is a closed convex hull, the $Nash(\cdot)$ is Lipschitz continuous on $\mathcal{X}_{RR}$. Then, we can have $\lim_{t \to \infty} Nash(X^t) = Nash(\lim_{t \to \infty} X^t)$.

Using the definition of repeated retraining given in Eq. (7), i.e., $X^{t+1} = Nash(X^t)$, we have: $X^{t \to \infty} = Nash(X^{t \to \infty})$. Thus, we have $\lim_{t \to \infty} \|Nash(X^t) - X^t\|_2 = 0$. Further, according to the definition of the performatively stable equilibrium $X^{PS}$ given in Eq. (6), we can conclude that the sequence $\{X^t\}$ converges to a performatively stable equilibrium: $\lim_{t \to \infty} X^t = X^{PS}$. Finally, as $\mathcal{X}_{RR}$ is a closed convex hull that contains all joint decisions during the repeated retraining procedure, it must also contain the limit point $X^{t \to \infty}$. Therefore, the performatively stable equilibrium $X^{PS}$, being equal to $X^{t \to \infty}$, belongs to the set $\mathcal{X}_{RR}$: $X^{PS} \in \mathcal{X}_{RR}$. ∎

## D   Proof of Lemma 1

We decompose the loss function of player $i$ into two components:

$$\ell_i(x_i, x_{-i}, Z_i) = \ell_{i1}(x_i, x_{-i}, Z_i) + \ell_{i2}(x_i, x_{-i}), \tag{42}$$

where $\ell_{i1}(x_i, x_{-i}, Z_i)$ represents the $Z_i$-dependent component, and $\ell_{i2}(x_i, x_{-i})$ represents the $Z_i$-independent component. We then define the following gradients:

- $\nabla_{x_i} \ell_i {}_{z_{i_j} \sim \mathcal{D}_i} = \nabla_{x_i} \ell_{i1}(x_i, x_{-i}, z_{i_j}) + \nabla_{x_i} \ell_{i2}(x_i, x_{-i})$: the gradient of the loss function for a single sample $z_{i_j}$ from data distribution $\mathcal{D}_i$.

- $\nabla_{x_i} \ell_i {}_{Z_{i_m} \sim \mathcal{D}_i} = \frac{1}{m} \sum_{z_{i_j} \in Z_{i_m}} \nabla_{x_i} \ell_{i1}(x_i, x_{-i}, z_{i_j}) + \nabla_{x_i} \ell_{i2}(x_i, x_{-i})$: the average gradient of the loss function over a set of $m$ i.i.d. samples from data distribution $\mathcal{D}_i$, where $Z_{i_m}$ is the sample set containing these $m$ samples, and $z_{i_j}$ is the $j^{th}$ sample in set $Z_{i_m}$.

- $\nabla_{x_i} \ell_i {}_{\mathcal{D}_i} = \mathop{\mathbb{E}}_{Z_i \sim \mathcal{D}_i} \nabla_{x_i} \ell_i(x_i, x_{-i}, Z_i)$: the expected gradient of the loss function over the underlying, ground-truth distribution $\mathcal{D}_i$.

- $\nabla_X \mathcal{G}_{z_{\bullet_j} \sim \mathcal{D}} = \left( \nabla_{x_1} \ell_1 {}_{z_{1_j} \sim \mathcal{D}_1}, \cdots, \nabla_{x_n} \ell_n {}_{z_{n_j} \sim \mathcal{D}_n} \right)^T$: the individual gradients of the game for samples $z_{i_j}$ from data distribution $\mathcal{D}_i, i \in [n]$, where each player $i$ obtains a sample $z_{i_j}$.

- $\nabla_X \mathcal{G}_{Z_{\bullet_m} \sim \mathcal{D}} = \left( \nabla_{x_1} \ell_1 {}_{Z_{1_m} \sim \mathcal{D}_1}, \cdots, \nabla_{x_n} \ell_n {}_{Z_{n_m} \sim \mathcal{D}_n} \right)^T$: the average individual gradients of the game over a set of $m$ i.i.d. samples from data distribution $\mathcal{D}_i, i \in [n]$, where each player $i$ obtains $m$ samples.

- $\nabla_X \mathcal{G}_{\mathcal{D}} = (\nabla_{x_1} \ell_1 \, \mathcal{D}_1, \cdots, \nabla_{x_n} \ell_n \, \mathcal{D}_n)^T$: the expected individual gradients of the game over the underlying, ground-truth distribution $\mathcal{D}_i, i \in [n]$.

Then, the covariance matrix of $\nabla_X \mathcal{G}_{z_{\bullet_j} \sim \mathcal{D}}, j = 1, 2, \cdots, m$ is given by:

$$
\begin{aligned}
\mathbf{\Sigma} &= \mathbf{Cov}\left[\nabla_X \mathcal{G}_{z_{\bullet_j} \sim \mathcal{D}}, \nabla_X \mathcal{G}_{z_{\bullet_j} \sim \mathcal{D}}\right] \\
&= \frac{1}{m} \sum_{j=1}^{m} \left(\nabla_X \mathcal{G}_{z_{\bullet_j} \sim \mathcal{D}} - \nabla_X \mathcal{G}_{Z_{\bullet_m} \sim \mathcal{D}}\right)\left(\nabla_X \mathcal{G}_{z_{\bullet_j} \sim \mathcal{D}} - \nabla_X \mathcal{G}_{Z_{\bullet_m} \sim \mathcal{D}}\right)^T,
\end{aligned}
\tag{43}
$$

where $\mathbf{\Sigma}$ is the covariance matrix.

According to the Central Limit Theorem (Bauer, 2001), we have:

$$
\sqrt{m}\left(\nabla_X \mathcal{G}_{Z_{\bullet_m} \sim \mathcal{D}} - \nabla_X \mathcal{G}_{\mathcal{D}}\right) \sim \mathcal{N}_d(\mathbf{0}, \mathbf{\Sigma}),
\tag{44}
$$

where $\mathcal{N}_d(\mathbf{0}, \mathbf{\Sigma})$ denotes the $d$-dimensional zero-mean Gaussian distribution with covariance matrix $\mathbf{\Sigma}$. The eigendecomposition of $\mathbf{\Sigma}$ yields:

$$
\mathbf{\Sigma} = \mathbf{U}\mathbf{\Lambda}\mathbf{U}^T,
\tag{45}
$$

where $\mathbf{U}$ is the $d \times d$ matrix whose $j^{th}$ column is the $j^{th}$ eigenvector $\mathbf{u}_j$ of $\mathbf{\Sigma}$, and $\mathbf{\Lambda}$ is the diagonal matrix whose diagonal elements are the corresponding eigenvalues of $\mathbf{\Sigma}$: $\mathbf{\Lambda}_{jj} = \lambda_j$.

Let $\mathbf{a} = \mathbf{\Lambda}^{-\frac{1}{2}}\mathbf{U}^T\sqrt{m}\left(\nabla_X \mathcal{G}_{Z_{\bullet_m} \sim \mathcal{D}} - \nabla_X \mathcal{G}_{\mathcal{D}}\right)$, we have:

$$
\begin{aligned}
\mathbf{a} &\sim \mathcal{N}_d(\mathbf{0}, \mathbf{\Lambda}^{-\frac{1}{2}}\mathbf{U}^T\mathbf{\Sigma}\mathbf{U}\mathbf{\Lambda}^{-\frac{1}{2}}) \\
\Longrightarrow \mathbf{a} &\sim \mathcal{N}_d(\mathbf{0}, \mathbf{\Lambda}^{-\frac{1}{2}}\mathbf{U}^T\left(\mathbf{U}\mathbf{\Lambda}\mathbf{U}^T\right)\mathbf{U}\mathbf{\Lambda}^{-\frac{1}{2}}) \\
\Longrightarrow \mathbf{a} &\sim \mathcal{N}_d(\mathbf{0}, \mathbf{I}_d),
\end{aligned}
\tag{46}
$$

where $\mathbf{I}_d$ denotes the $d \times d$ identity matrix, and the vector $\mathbf{a}$ follows the standard $d$-dimensional multivariate normal distribution $\mathcal{N}_d(\mathbf{0}, \mathbf{I}_d)$. Accordingly, the quadratic form $\mathbf{a}^T\mathbf{a}$ follows a $\chi^2$ distribution with $d$ degrees of freedom (Pearson, 1893)[8]:

$$
\begin{aligned}
\mathbf{a}^T\mathbf{a} &= m\left(\nabla_X \mathcal{G}_{Z_{\bullet_m} \sim \mathcal{D}} - \nabla_X \mathcal{G}_{\mathcal{D}}\right)^T \mathbf{U}\mathbf{\Lambda}^{-\frac{1}{2}}\mathbf{\Lambda}^{-\frac{1}{2}}\mathbf{U}^T\left(\nabla_X \mathcal{G}_{Z_{\bullet_m} \sim \mathcal{D}} - \nabla_X \mathcal{G}_{\mathcal{D}}\right) \\
&= m\left(\nabla_X \mathcal{G}_{Z_{\bullet_m} \sim \mathcal{D}} - \nabla_X \mathcal{G}_{\mathcal{D}}\right)^T \mathbf{\Sigma}^{-1}\left(\nabla_X \mathcal{G}_{Z_{\bullet_m} \sim \mathcal{D}} - \nabla_X \mathcal{G}_{\mathcal{D}}\right) \\
&\sim \chi^2(d).
\end{aligned}
\tag{47}
$$

Then, for any positive real number $\tau$, we have:

$$
P\left[m\left(\nabla_X \mathcal{G}_{Z_{\bullet_m} \sim \mathcal{D}} - \nabla_X \mathcal{G}_{\mathcal{D}}\right)^T \mathbf{\Sigma}^{-1}\left(\nabla_X \mathcal{G}_{Z_{\bullet_m} \sim \mathcal{D}} - \nabla_X \mathcal{G}_{\mathcal{D}}\right) \leq \tau\right] = F_{\chi^2(d)}(\tau),
\tag{48}
$$

where $F_{\chi^2(d)}(\tau)$ denotes the cumulative distribution function of the $\chi^2$ distribution with $d$ degrees of freedom, evaluated at $\tau$.

For any real symmetric matrix $\mathbf{B} \in \mathbb{R}^{d \times d}$ and vector $\mathbf{v} \in \mathbb{R}^d$, the inequality $\mathbf{v}^T\mathbf{B}\mathbf{v} \geq \lambda_{min}(\mathbf{B})\|\mathbf{v}\|_2^2$ holds, where $\lambda_{min}(\mathbf{B})$ denotes the smallest eigenvalue of $\mathbf{B}$. Therefore, we have:

$$
\begin{aligned}
&m\left(\nabla_X \mathcal{G}_{Z_{\bullet_m} \sim \mathcal{D}} - \nabla_X \mathcal{G}_{\mathcal{D}}\right)^T \mathbf{\Sigma}^{-1}\left(\nabla_X \mathcal{G}_{Z_{\bullet_m} \sim \mathcal{D}} - \nabla_X \mathcal{G}_{\mathcal{D}}\right) \\
&\geq m\lambda_{min}(\mathbf{\Sigma}^{-1})\|\nabla_X \mathcal{G}_{Z_{\bullet_m} \sim \mathcal{D}} - \nabla_X \mathcal{G}_{\mathcal{D}}\|_2^2.
\end{aligned}
\tag{49}
$$

---

[8]For the case of non-invertible $\mathbf{\Sigma}$, please refer to Appendix G.1 for details.

Using this inequality, we can derive the following:

$$
\begin{aligned}
&P\left[m\lambda_{min}(\mathbf{\Sigma}^{-1})\|\nabla_X\mathcal{G}_{Z_{\bullet_m}\sim\mathcal{D}} - \nabla_X\mathcal{G}_\mathcal{D}\|_2^2 \leq \tau\right]\\
&= P\left[\|\nabla_X\mathcal{G}_{Z_{\bullet_m}\sim\mathcal{D}} - \nabla_X\mathcal{G}_\mathcal{D}\|_2^2 \leq \frac{\tau\lambda_{max}(\mathbf{\Sigma})}{m}\right]\\
&= P\left[\|\nabla_X\mathcal{G}_{Z_{\bullet_m}\sim\mathcal{D}} - \nabla_X\mathcal{G}_\mathcal{D}\|_2 \leq \sqrt{\frac{\tau\lambda_{max}(\mathbf{\Sigma})}{m}}\right]\\
&\geq P\left[m\left(\nabla_X\mathcal{G}_{Z_{\bullet_m}\sim\mathcal{D}} - \nabla_X\mathcal{G}_\mathcal{D}\right)^T\mathbf{\Sigma}^{-1}\left(\nabla_X\mathcal{G}_{Z_{\bullet_m}\sim\mathcal{D}} - \nabla_X\mathcal{G}_\mathcal{D}\right) \leq \tau\right]\\
&= F_{\chi^2(d)}(\tau).
\end{aligned}
\tag{50}
$$

Let $\delta = \sqrt{\frac{\tau\lambda_{max}(\mathbf{\Sigma})}{m}}$ and $\sigma = \lambda_{max}(\mathbf{\Sigma})$, then $\tau = \frac{m\delta^2}{\sigma}$. When each player $i$ samples $m$ samples, we can state that the following inequality holds with probability at least $F_{\chi^2(d)}(\frac{m\delta^2}{\sigma})$:

$$
\|\nabla_X\mathcal{G}_{Z_{\bullet_m}\sim\mathcal{D}} - \nabla_X\mathcal{G}_\mathcal{D}\|_2 \leq \delta.
\tag{51}
$$

Consider the game on the data distribution induced by the joint decision $Y$, we can set $\nabla_X\mathcal{G}_{Z_{\bullet_m}\sim\mathcal{D}} = \overline{\nabla_X\mathcal{G}_m(Y)}$ and $\nabla_X\mathcal{G}_\mathcal{D} = \nabla_X\mathcal{G}(Y)$ to complete the proof. ∎

## E   Lemma 2

**Lemma 2.** *For any $\delta > 0$, and given the third power of the $L_2$ norm $\rho = \sum_{j=1}^m \mathbb{E}\left[\left\|m^{-\frac{1}{2}}\mathbf{\Lambda}^{-\frac{1}{2}}\mathbf{U}^T\left(\nabla_X\mathcal{G}_{z_{\bullet_j}\sim\mathcal{D}} - \nabla_X\mathcal{G}_\mathcal{D}\right)\right\|_2^3\right]$ exists, the following inequality holds with probability at least $F_{\chi^2(d)}\left(\frac{m\delta^2}{\sigma}\right) - Cd^{\frac{1}{4}}\rho$:*

$$
\left\|\overline{\nabla_X\mathcal{G}_m(Y)} - \nabla_X\mathcal{G}(Y)\right\|_2 \leq \delta,
\tag{52}
$$

*where $C$ is a finite positive universal constant, $\mathbf{\Sigma}$ is the covariance matrix of $\overline{\nabla_X\mathcal{G}_m(Y)}$ with eigen decomposition $\mathbf{\Sigma} = \mathbf{U}\mathbf{\Lambda}\mathbf{U}^T$, $\mathbf{U}$ is the eigenvector matrix, and $\mathbf{\Lambda}$ is the eigenvalue matrix.*

**Proof.** *We define $\mathbf{b}_j = \mathbf{\Lambda}^{-\frac{1}{2}}\mathbf{U}^T\left(\nabla_X\mathcal{G}_{z_{\bullet_j}\sim\mathcal{D}} - \nabla_X\mathcal{G}_\mathcal{D}\right)$ and $\bar{B} = \mathbf{\Lambda}^{-\frac{1}{2}}\mathbf{U}^T\left(\nabla_X\mathcal{G}_{Z_{\bullet_m}\sim\mathcal{D}} - \nabla_X\mathcal{G}_\mathcal{D}\right)$. Then, we have $\mathbb{E}[\mathbf{b}_j] = 0$ and $Cov(\mathbf{b}_j) = \mathbf{\Lambda}^{-\frac{1}{2}}\mathbf{U}^T\mathbf{\Sigma}\mathbf{U}\mathbf{\Lambda}^{-\frac{1}{2}} = \mathbf{I}_d$.*

*According to the multidimensional Berry–Esseen theorem (Bentkus, 2005), when $\rho = \sum_{i=1}^m \mathbb{E}[\|m^{-\frac{1}{2}}\mathbf{b}_i\|_2^3]$ exists, for all convex set $D$, sample size $m$, and normal distribution $Z \sim \mathcal{N}_d(\mathbf{0}, \mathbf{I}_d)$, we have:*

$$
|P[\sqrt{m}\bar{B} \in D] - P[Z \in D]| \leq Cd^{\frac{1}{4}}\rho,
\tag{53}
$$

*where $C$ is a universal constant.*

*Define the convex set $D = \{v \in \mathbb{R}^d : \|v\|_2^2 \leq \tau\}$. Then:*

$$
|P[m\|\bar{B}\|_2^2 \leq \tau] - P[\|Z\|_2^2 \leq \tau]| \leq Cd^{\frac{1}{4}}\rho.
\tag{54}
$$

*Since $Z \sim \mathcal{N}_d(0, \mathbf{I}_d)$, for any positive real number $\tau$, we have $P\left[\|Z\|_2^2 \leq \tau\right] = F_{\chi^2(d)}(\tau)$.*

*Using Eq. (49), we derive:*

$$
\begin{aligned}
&P\left[m\lambda_{min}(\mathbf{\Sigma}^{-1})\|\nabla_X\mathcal{G}_{Z_{\bullet_m}\sim\mathcal{D}} - \nabla_X\mathcal{G}_\mathcal{D}\|_2^2 \leq \tau\right] \\
&= P\left[\|\nabla_X\mathcal{G}_{Z_{\bullet_m}\sim\mathcal{D}} - \nabla_X\mathcal{G}_\mathcal{D}\|_2^2 \leq \frac{\tau\lambda_{max}(\mathbf{\Sigma})}{m}\right] \\
&= P\left[\|\nabla_X\mathcal{G}_{Z_{\bullet_m}\sim\mathcal{D}} - \nabla_X\mathcal{G}_\mathcal{D}\|_2 \leq \sqrt{\frac{\tau\lambda_{max}(\mathbf{\Sigma})}{m}}\right] \\
&\geq P\left[m\left(\nabla_X\mathcal{G}_{Z_{\bullet_m}\sim\mathcal{D}} - \nabla_X\mathcal{G}_\mathcal{D}\right)^T\mathbf{\Sigma}^{-1}\left(\nabla_X\mathcal{G}_{Z_{\bullet_m}\sim\mathcal{D}} - \nabla_X\mathcal{G}_\mathcal{D}\right) \leq \tau\right] \\
&= P[m\|\bar{B}\|_2^2 \leq \tau] \\
&\geq F_{\chi^2(d)}(\tau) - Cd^{\frac{1}{4}}\rho,
\end{aligned}
\tag{55}
$$

*where $\rho = \sum_{j=1}^m \mathbb{E}\left[\left\|m^{-\frac{1}{2}}\mathbf{\Lambda}^{-\frac{1}{2}}\mathbf{U}^T\left(\nabla_X\mathcal{G}_{z_{\bullet_j}\sim\mathcal{D}} - \nabla_X\mathcal{G}_\mathcal{D}\right)\right\|_2^3\right]$.*

*Let $\delta = \sqrt{\frac{\tau\lambda_{max}(\mathbf{\Sigma})}{m}}$ and $\sigma = \lambda_{max}(\mathbf{\Sigma})$. Then $\tau = \frac{m\delta^2}{\sigma}$. When each player $i$ samples $m$ samples, the following holds with probability at least $F_{\chi^2(d)}(\frac{m\delta^2}{\sigma}) - Cd^{\frac{1}{4}}\rho$:*

$$
\|\nabla_X\mathcal{G}_{Z_{\bullet_m}\sim\mathcal{D}} - \nabla_X\mathcal{G}_\mathcal{D}\|_2 \leq \delta.
\tag{56}
$$

*For the game on the data distribution induced by the joint decision $Y$, set $\nabla_X\mathcal{G}_{Z_{\bullet_m}\sim\mathcal{D}} = \overline{\nabla_X\mathcal{G}_m(Y)}$ and $\nabla_X\mathcal{G}_\mathcal{D} = \nabla_X\mathcal{G}(Y)$ to complete the proof.*

*Note that, the exact value of $C$ is currently under investigation. Recent results, such as (Raič, 2019), may provide tight bounds.* ∎

## F   Lemma 3

**Lemma 3.** *For any $\delta > \frac{\|\boldsymbol{\sigma}_{sample}\|_2}{\sqrt{m}}$, with probability $1 - \frac{\|\boldsymbol{\sigma}_{sample}\|_2^2}{m\delta^2}$, the following inequality holds:*

$$
\left\|\overline{\nabla_X\mathcal{G}_m(Y)} - \nabla_X\mathcal{G}(Y)\right\|_2 \leq \delta,
\tag{57}
$$

*where $\boldsymbol{\sigma}_{sample} = (\sigma_{sample}^{(1)}, \sigma_{sample}^{(2)}, \cdots, \sigma_{sample}^{(d)})^T$ is the vector of standard deviations for the components of the sample gradient calculated with a single sample $z_{i_j}$ by each player $i$.*

**Proof.** *Consider the sample gradient $\nabla_X\mathcal{G}_{z_{\bullet_j}\sim\mathcal{D}}$ and the average gradient $\nabla_X\mathcal{G}_{Z_{\bullet_m}\sim\mathcal{D}}$. We have the following:*

$$
\mathbb{E}\left[\nabla_X\mathcal{G}_{Z_{\bullet_m}\sim\mathcal{D}}\right] = \mathbb{E}\left[\frac{1}{m}\sum_{j=1}^m\nabla_X\mathcal{G}_{z_{\bullet_j}\sim\mathcal{D}}\right] = \frac{1}{m}\sum_{j=1}^m\mathbb{E}\left[\nabla_X\mathcal{G}_{z_{\bullet_j}\sim\mathcal{D}}\right] = \mathbb{E}\left[\nabla_X\mathcal{G}_{z_{\bullet_j}\sim\mathcal{D}}\right] = \nabla_X\mathcal{G}_\mathcal{D}.
\tag{58}
$$

*Thus, the expectation of the average gradient equals the expected gradient.*

*In terms of standard deviation, let $\boldsymbol{\sigma}_{sample} = (\sigma_{sample}^{(1)}, \sigma_{sample}^{(2)}, \cdots, \sigma_{sample}^{(d)})^T$ denote the vector of standard deviations for the components of the sample gradient $\nabla_X\mathcal{G}_{z_{\bullet_j}\sim\mathcal{D}}$, and let $\boldsymbol{\sigma}_{avg} = (\sigma_{avg}^{(1)}, \sigma_{avg}^{(2)}, \cdots, \sigma_{avg}^{(d)})^T$ denotes the standard deviations for the components of the average gradient $\nabla_X\mathcal{G}_{Z_{\bullet_m}\sim\mathcal{D}}$. Then, for each dimension $k \in \{1, 2, \cdots, d\}$ we have:*

$$
\left(\sigma_{avg}^{(k)}\right)^2 = \mathrm{Var}\left(\frac{1}{m}\sum_{j=1}^m\nabla_X\mathcal{G}_{z_{\bullet_j}\sim\mathcal{D}}^{(k)}\right) = \frac{1}{m^2}\sum_{j=1}^m\mathrm{Var}\left(\nabla_X\mathcal{G}_{z_{\bullet_j}\sim\mathcal{D}}^{(k)}\right) = \frac{1}{m^2}\cdot m\cdot\left(\sigma_{sample}^{(k)}\right)^2 = \frac{1}{m}\left(\sigma_{sample}^{(k)}\right)^2.
\tag{59}
$$

*Thus, the standard deviation satisfies:*

$$
\frac{1}{\sqrt{m}}\sigma_{sample}^{(k)} = \sigma_{avg}^{(k)} \implies \frac{1}{\sqrt{m}}\|\boldsymbol{\sigma}_{sample}\|_2 = \|\boldsymbol{\sigma}_{avg}\|_2.
\tag{60}
$$

*Applying the multidimensional Chebyshev's inequality, for any $q > 1$, we obtain the following probabilistic bound:*

$$P\left(\|\nabla_X \mathcal{G}_{Z_{\bullet_m} \sim \mathcal{D}} - \nabla_X \mathcal{G}_\mathcal{D}\|_2 \leq q\|\boldsymbol{\sigma}_{avg}\|_2\right)$$
$$= P\left(\|\nabla_X \mathcal{G}_{Z_{\bullet_m} \sim \mathcal{D}} - \nabla_X \mathcal{G}_\mathcal{D}\|_2 \leq \frac{q}{\sqrt{m}}\|\boldsymbol{\sigma}_{sample}\|_2\right) \geq 1 - \frac{1}{q^2}. \quad (61)$$

*We set $\delta = \frac{q}{\sqrt{m}}\|\boldsymbol{\sigma}_{sample}\|_2$, then for any $\delta > \frac{1}{\sqrt{m}}\|\boldsymbol{\sigma}_{sample}\|_2$, we have:*

$$P\left(\|\nabla_X \mathcal{G}_{Z_{\bullet_m} \sim \mathcal{D}} - \nabla_X \mathcal{G}_\mathcal{D}\|_2 \leq \delta\right) \geq 1 - \frac{\|\boldsymbol{\sigma}_{sample}\|_2^2}{m\delta^2}. \quad (62)$$

■

## G  Proof of Theorem 3

By applying the triangle inequality, we have:

$$\|\nabla_{Nash(X^t)} \mathcal{G}_{Z_{\bullet_{m_t}} \sim \mathcal{D}(X^t)} - \nabla_{Nash(X^t)} \mathcal{G}_{\mathcal{D}(X^{PS})}\|_2$$
$$\leq \|\nabla_{Nash(X^t)} \mathcal{G}_{Z_{\bullet_{m_t}} \sim \mathcal{D}(X^t)} - \nabla_{Nash(X^t)} \mathcal{G}_{\mathcal{D}(X^t)}\|_2 + \quad (63)$$
$$\|\nabla_{Nash(X^t)} \mathcal{G}_{\mathcal{D}(X^t)} - \nabla_{Nash(X^t)} \mathcal{G}_{\mathcal{D}(X^{PS})}\|_2,$$

where $m_t$ is the number of samples each player collects at time step $t$, $Z_{\bullet_{m_t}} \sim \mathcal{D}(X^t)$ denotes the set of $m_t$ samples drawn from the data distribution $\mathcal{D}_i(X^t)$ for each $i \in [n]$, $\nabla_{Nash(X^t)} \mathcal{G}_{Z_{\bullet_{m_t}} \sim \mathcal{D}(X^t)}$ represents the average of individual gradients at the joint decision $Nash(X^t)$, computed over $m_t$ collected samples, $\nabla_{Nash(X^t)} \mathcal{G}_{\mathcal{D}(X^t)}$ is the expected gradient of the game at $Nash(X^t)$ with respect to the true data distribution at time step $t$, and $\nabla_{Nash(X^t)} \mathcal{G}_{\mathcal{D}(X^{PS})}$ is the expected gradient of the game at $Nash(X^t)$ with respect to the stable distribution $\mathcal{D}_i(X^{PS}), i \in [n]$.

From Theorem 2, we know that $X^{PS} \in \mathcal{X}_{RR}$ when $\alpha > \sqrt{\sum_{i=1}^n \hat{\varepsilon}_i^2}$. As stated in Definition 3, $Y$, $X$, and $X'$ can be any joint decisions in the set $\mathcal{X}_{RR}$. Therefore, by setting $Y = Nash(X^t)$, $X = X^t$, and $X' = X^{PS}$ in Eq. (33), we obtain:

$$\|\nabla_{Nash(X^t)} \mathcal{G}_{\mathcal{D}(X^t)} - \nabla_{Nash(X^t)} \mathcal{G}_{\mathcal{D}(X^{PS})}\|_2 \leq \sqrt{\sum_{i=1}^n \hat{\varepsilon}_i^2} \|X^t - X^{PS}\|_2. \quad (64)$$

Similarly, by setting $X = Nash(X^t)$, $m = m_t$, and $\mathcal{D} = \mathcal{D}(X^t)$, according to Eq. (51), the following inequality holds with probability at least $F_{\chi^2(d)}\left(\frac{m_t \sum_{i=1}^n \hat{\varepsilon}_i^2 \delta^2}{\sigma_t}\right)$:

$$\|\nabla_{Nash(X^t)} \mathcal{G}_{Z_{\bullet_{m_t}} \sim \mathcal{D}(X^t)} - \nabla_{Nash(X^t)} \mathcal{G}_{\mathcal{D}(X^t)}\|_2 \leq \sqrt{\sum_{i=1}^n \hat{\varepsilon}_i^2} \delta. \quad (65)$$

Combining Eqs. (63-65), when $\|X^t - X^{PS}\|_2 > \delta$ and $\alpha > \sqrt{\sum_{i=1}^n \hat{\varepsilon}_i^2}$, we have:

$$\|\nabla_{Nash(X^t)} \mathcal{G}_{Z_{\bullet_{m_t}} \sim \mathcal{D}(X^t)} - \nabla_{Nash(X^t)} \mathcal{G}_{\mathcal{D}(X^{PS})}\|_2$$
$$\leq \sqrt{\sum_{i=1}^n \hat{\varepsilon}_i^2} \delta + \sqrt{\sum_{i=1}^n \hat{\varepsilon}_i^2} \|X^t - X^{PS}\|_2 \leq 2\sqrt{\sum_{i=1}^n \hat{\varepsilon}_i^2} \|X^t - X^{PS}\|_2, \quad (66)$$

with probability at least $F_{\chi^2(d)}\left(\frac{m_t \sum_{i=1}^n \hat{\varepsilon}_i^2 \delta^2}{\sigma_t}\right)$.

Since $X^{PS}$ and $Nash(X^t)$ are the Nash equilibria of game $\mathcal{G}(X^{PS})$ and $\mathcal{G}(X^t)$, respectively, and the game $\mathcal{G}(\cdot)$ is monotone, according to the first-order optimality condition (Bubeck et al., 2015), the following inequalities hold:

$$
\begin{aligned}
(X^{PS} - Nash(X^t))^T \nabla_{Nash(X^t)} \mathcal{G}_{Z_{\bullet_{m_t}} \sim \mathcal{D}(X^t)} &\geq 0, \\
(Nash(X^t) - X^{PS})^T \nabla_{X^{PS}} \mathcal{G}_{\mathcal{D}(X^{PS})} &\geq 0.
\end{aligned}
\tag{67}
$$

Combining the two inequalities in Eq. (67), we have:

$$
\begin{aligned}
&(X^{PS} - Nash(X^t))^T \nabla_{Nash(X^t)} \mathcal{G}_{Z_{\bullet_{m_t}} \sim \mathcal{D}(X^t)} + (Nash(X^t) - X^{PS})^T \nabla_{X^{PS}} \mathcal{G}_{\mathcal{D}(X^{PS})} \\
=& (Nash(X^t) - X^{PS})^T \left( \nabla_{X^{PS}} \mathcal{G}_{\mathcal{D}(X^{PS})} - \nabla_{Nash(X^t)} \mathcal{G}_{Z_{\bullet_{m_t}} \sim \mathcal{D}(X^t)} \right) \\
=& (Nash(X^t) - X^{PS})^T \left( \nabla_{X^{PS}} \mathcal{G}_{\mathcal{D}(X^{PS})} - \nabla_{Nash(X^t)} \mathcal{G}_{\mathcal{D}(X^{PS})} \right) + \\
& (Nash(X^t) - X^{PS})^T \left( \nabla_{Nash(X^t)} \mathcal{G}_{\mathcal{D}(X^{PS})} - \nabla_{Nash(X^t)} \mathcal{G}_{Z_{\bullet_{m_t}} \sim \mathcal{D}(X^t)} \right) \\
\geq& 0.
\end{aligned}
\tag{68}
$$

The last inequality in Eq. (68) can be rewritten as follows:

$$
\begin{aligned}
&(Nash(X^t) - X^{PS})^T \left( \nabla_{Nash(X^t)} \mathcal{G}_{\mathcal{D}(X^{PS})} - \nabla_{X^{PS}} \mathcal{G}_{\mathcal{D}(X^{PS})} \right) \\
\leq& (Nash(X^t) - X^{PS})^T \left( \nabla_{Nash(X^t)} \mathcal{G}_{\mathcal{D}(X^{PS})} - \nabla_{Nash(X^t)} \mathcal{G}_{Z_{\bullet_{m_t}} \sim \mathcal{D}(X^t)} \right).
\end{aligned}
\tag{69}
$$

As the game $\mathcal{G}(\cdot)$ is $\alpha$-strongly monotone, we have:

$$
\alpha \| Nash(X^t) - X^{PS} \|_2^2 \leq \left( \nabla_{Nash(X^t)} \mathcal{G}_{\mathcal{D}(X^{PS})} - \nabla_{X^{PS}} \mathcal{G}_{\mathcal{D}(X^{PS})} \right)^T (Nash(X^t) - X^{PS}).
\tag{70}
$$

Multiplying both sides of Eq. (66) by $\left\| Nash(X^t) - X^{PS} \right\|_2$ and applying Cauchy-Schwarz inequality (Hunter & Nachtergaele, 2001), when $\alpha > \sqrt{\sum_{i=1}^n \hat{\varepsilon}_i^2}$, we have:

$$
\begin{aligned}
&\left( \nabla_{Nash(X^t)} \mathcal{G}_{\mathcal{D}(X^{PS})} - \nabla_{Nash(X^t)} \mathcal{G}_{Z_{\bullet_{m_t}} \sim \mathcal{D}(X^t)} \right)^T (Nash(X^t) - X^{PS}) \\
\leq& \left\| Nash(X^t) - X^{PS} \right\|_2 \left\| \nabla_{Nash(X^t)} \mathcal{G}_{\mathcal{D}(X^{PS})} - \nabla_{Nash(X^t)} \mathcal{G}_{Z_{\bullet_{m_t}} \sim \mathcal{D}(X^t)} \right\|_2 \\
\leq& 2 \sqrt{\sum_{i=1}^n \hat{\varepsilon}_i^2} \left\| Nash(X^t) - X^{PS} \right\|_2 \| X^t - X^{PS} \|_2,
\end{aligned}
\tag{71}
$$

with probability at least $F_{\chi^2(d)}(\frac{m_t \sum_{i=1}^n \hat{\varepsilon}_i^2 \delta^2}{\sigma_t})$. Combining Eqs. (69-71), when $\alpha > \sqrt{\sum_{i=1}^n \hat{\varepsilon}_i^2}$, we have:

$$
\| Nash(X^t) - X^{PS} \|_2 \leq \frac{2\sqrt{\sum_{i=1}^n \hat{\varepsilon}_i^2}}{\alpha} \| X^t - X^{PS} \|_2,
\tag{72}
$$

with probability at least $F_{\chi^2(d)}(\frac{m_t \sum_{i=1}^n \hat{\varepsilon}_i^2 \delta^2}{\sigma_t})$.

Similar to the proof of Theorem 2, let $\mathcal{B} = \frac{\sqrt{\sum_{i=1}^n \hat{\varepsilon}_i^2}}{\alpha}$. When $\alpha > 2\sqrt{\sum_{i=1}^n \hat{\varepsilon}_i^2}$, we have $\mathcal{B} < \frac{1}{2}$. By Theorem 2, $X^{PS} \in \mathcal{X}_{RR}$ when $\mathcal{B} < \frac{1}{2}$. As a result, we can conclude that, when each player collects a finite number of samples, the sequence $\{X^t\}$ converges towards the performatively stable equilibrium $X^{PS}$ during the repeated retraining procedure. Since the distribution maps $\mathcal{D}_i(X)$ are unknown, the sample sets obtained after decision deployment cannot be anticipated. To obtain a practically feasible result, we assume the covariance matrices and the corresponding eigenvalues are independent across time steps under unknown distribution maps $\mathcal{D}_i(X)$. Specifically, with probability at least $\prod_{\mu=1}^t F_{\chi^2(d)} \left( \frac{m_\mu \sum_{i=1}^n \hat{\varepsilon}_i^2 \delta^2}{\sigma_\mu} \right)$, we have:

$$
\| X^t - X^{PS} \|_2 \leq 2\mathcal{B} \| X^{t-1} - X^{PS} \|_2 \leq (2\mathcal{B})^t \| X^0 - X^{PS} \|_2.
\tag{73}
$$

Given $\mathcal{B} < \frac{1}{2}$, for any arbitrarily small positive number $\delta$, to ensure that $\|X^t - X^{PS}\|_2 \leq \delta$, it suffices to satisfy:

$$(2\mathcal{B})^t \|X^0 - X^{PS}\|_2 \leq \delta \iff (2\mathcal{B})^t \leq \frac{\delta}{\|X^0 - X^{PS}\|_2}$$

$$\iff t\log(2\mathcal{B}) \leq \log\left(\frac{\delta}{\|X^0 - X^{PS}\|_2}\right) \iff t \geq \frac{\log\left(\frac{\delta}{\|X^0 - X^{PS}\|_2}\right)}{\log(2\mathcal{B})}. \tag{74}$$

Therefore, if $\mathcal{B} < \frac{1}{2}$, then with probability at least $\prod_{\mu=1}^t F_{\chi^2(d)}\left(\frac{m_\mu \sum_{i=1}^n \hat{\varepsilon}_i^2 \delta^2}{\sigma_\mu}\right)$, we have:

$$\|X^t - X^{PS}\|_2 \leq \delta, \ \forall \ t \geq \frac{\log\left(\frac{\delta}{\|X^0 - X^{PS}\|_2}\right)}{\log(2\mathcal{B})}. \tag{75}$$

∎

### G.1 Extension of Theorem 3 for not invertible $\mathbf{\Sigma}$

Begin with Lemma 1, according to the Central Limit Theorem (Bauer, 2001), we have:

$$\sqrt{m}\left(\nabla_X \mathcal{G}_{Z_{\bullet_m} \sim \mathcal{D}} - \nabla_X \mathcal{G}_{\mathcal{D}}\right) \sim \mathcal{N}_d(\mathbf{0}, \mathbf{\Sigma}). \tag{76}$$

For a non-invertible covariance matrix $\mathbf{\Sigma}$, we consider another vector $\mu \sim \mathcal{N}_d(\mathbf{0}, \eta\mathbf{I}_d)$, independent of the distribution given in Eq. (76), where $\eta$ is a very small positive number. According to the property of the Gaussian distribution, we have:

$$\sqrt{m}\left(\nabla_X \mathcal{G}_{Z_{\bullet_m} \sim \mathcal{D}} - \nabla_X \mathcal{G}_{\mathcal{D}}\right) + \sqrt{m}\mu \sim \mathcal{N}_d(\mathbf{0}, \mathbf{\Sigma} + m\eta\mathbf{I}_d). \tag{77}$$

Then we perform the same procedure in the proof of Lemma 1 (Appendix D) to $\sqrt{m}\left(\nabla_X \mathcal{G}_{Z_{\bullet_m} \sim \mathcal{D}} - \nabla_X \mathcal{G}_{\mathcal{D}}\right) + \sqrt{m}\mu$. Let $\delta = \sqrt{\frac{\tau\lambda_{max}(\mathbf{\Sigma} + m\eta\mathbf{I_d})}{m}}$ and $\sigma = \lambda_{max}(\mathbf{\Sigma} + \mathbf{m}\eta\mathbf{I_d})$, then we have $\tau = \frac{m\delta^2}{\sigma}$. When each player $i$ samples $m$ samples, we can state that the following inequality holds, with probability at least $F_{\chi^2(d)}\left(\frac{m\delta^2}{\sigma}\right)$:

$$\|\nabla_X \mathcal{G}_{Z_{\bullet_m} \sim \mathcal{D}} - \nabla_X \mathcal{G}_{\mathcal{D}} + \mu\|_2 \leq \delta. \tag{78}$$

By the triangle inequality, we know that $\|\nabla_X \mathcal{G}_{Z_{\bullet_m} \sim \mathcal{D}} - \nabla_X \mathcal{G}_{\mathcal{D}}\|_2 \leq \|\nabla_X \mathcal{G}_{Z_{\bullet_m} \sim \mathcal{D}} - \nabla_X \mathcal{G}_{\mathcal{D}} + \mu\|_2 + \|\mu\|_2$. Then, with probability at least $F_{\chi^2(d)}\left(\frac{m\delta^2}{\sigma}\right)$, we have:

$$\|\nabla_X \mathcal{G}_{Z_{\bullet_m} \sim \mathcal{D}} - \nabla_X \mathcal{G}_{\mathcal{D}}\|_2 \leq \delta + \|\mu\|_2. \tag{79}$$

For the scale of $\|\mu\|_2$, since $\frac{\mu}{\sqrt{\eta}} \sim \mathcal{N}_d(\mathbf{0}, \mathbf{I}_d)$, the quadratic form $\frac{\mu^T \mu}{\eta} \sim \chi^2(d)$ (Pearson, 1900). Then, for any positive real number $\tau$, we have:

$$P\left[\frac{\mu^T \mu}{\eta} \leq \tau\right] = P\left[\mu^T \mu \leq \eta\tau\right] = F_{\chi^2(d)}(\tau). \tag{80}$$

Thus, with the probability $F_{\chi^2(d)}(\frac{\delta'^2}{\eta})$, we have:

$$\|\mu\|_2 \leq \delta'. \tag{81}$$

From Eqs. (79) and (81), with a small positive $\eta$, the average empirical gradient $\nabla_X \mathcal{G}_{Z_{\bullet_m} \sim \mathcal{D}}$ approaches the expected gradient $\nabla_X \mathcal{G}_{\mathcal{D}}$ with high probability as the sample size $m$ increases. Specifically, as $\eta \to 0$, Eq. (81) holds for an arbitrarily small $\delta'$ with probability approaching 1. This result is consistent with Lemma 1. We follow the same procedure in the proof of Theorem 3 (Appendix G) by replacing $\delta$ with $\delta + \|\mu\|_2$. Since $\|\mu\|_2 \to 0$ almost surely as $\eta \to 0$, the conclusion of Theorem 3 remains valid. ∎

### G.2 Variants of Theorem 3

The convergence guarantee in Theorem 3 relies on the discrepancy between empirical and expected individual gradients at each iteration. By replacing the probability in Eq. (75) with that provided by Lemmas 2 and 3, we obtain the following variants of Theorem 3:

**Variant of Theorem 3 based on Lemma 2 (Berry-Esseen theorem):** For any $\delta > 0$, with probability at least $\prod_{\mu=1}^t \left( F_{\chi^2(d)} \left( \frac{m_\mu \delta^2}{\sigma_\mu} \right) - C d^{\frac{1}{4}} \rho_\mu \right)$, the following inequality holds:

$$\|X^t - X^{PS}\|_2 \leq \delta, \forall t \geq \frac{\log \left( \frac{\delta}{\|X^0 - X^{PS}\|_2} \right)}{\log \left( \frac{2\sqrt{\sum_{i=1}^n \hat{\varepsilon}_i^2}}{\alpha} \right)}. \tag{82}$$

**Variant of Theorem 3 based on Lemma 3 (Chebyshev's inequality):** For any $\delta > \max_\mu \frac{\|\boldsymbol{\sigma}_{sample_\mu}\|_2}{\sqrt{m_\mu}}$, with probability at least $\prod_{\mu=1}^t \left( 1 - \frac{\|\boldsymbol{\sigma}_{sample_\mu}\|_2^2}{m_\mu \delta^2} \right)$, the following inequality holds:

$$\|X^t - X^{PS}\|_2 \leq \delta, \forall t \geq \frac{\log \left( \frac{\delta}{\|X^0 - X^{PS}\|_2} \right)}{\log \left( \frac{2\sqrt{\sum_{i=1}^n \hat{\varepsilon}_i^2}}{\alpha} \right)}. \tag{83}$$

## H An Example of Non-Convergence to a Performatively Stable Equilibrium in Repeated Retraining

Here, we apply the 1-dimensional case of the Prediction Error Minimization Game (Section 5.1) as a practical example. Each platform $i \in [n]$ seeks to predict outcome $z_i \in \mathbb{R}^{m_i}$ with feature set $\theta_i \in \mathbb{R}^{m_i}$ by solving a regression problem, i.e., minimizing the loss function: $\ell_i(x_i, z_i, \theta_i) = \|z_i - \theta_i x_i\|_2^2$, where $m_i$ is the sample number, $x_i \in \mathbb{R}$ is the regression model parameter. To facilitate the illustration, we assume the outcome $z_i$ is only affected by the player's own model $x_i$ through the distribution map:

$$z_{i\ x_i} \sim \mathcal{D}_i(x_i) = z_i^0 + \mu \theta_i x_i, \tag{84}$$

where $z_{i\ x_i}$ is the outcome for platform $i$ after deploying model $x_i$, and $z_i^0$ is the initial outcome.

- Without regularization term, the closed-form solution of minimizing $\ell_i$ at time step $t$ is $x_i^t = \frac{\theta_i^T z_i^0}{\theta_i^T \theta_i} + \mu x_i^{t-1}$. We then calculate the difference between models in neighboring iterations:

$$|x_i^t - x_i^{t-1}| = \left| \frac{\theta_i^T z_i^0}{\theta_i^T \theta_i} + \mu x_i^{t-1} - \frac{\theta_i^T z_i^0}{\theta_i^T \theta_i} - \mu x_i^{t-2} \right| = |\mu(x_i^{t-1} - x_i^{t-2})|. \tag{85}$$

  In this case, if $|\mu| > 1$, then $|x_i^t - x_i^{t-1}| > |x_i^{t-1} - x_i^{t-2}|$, which implies that the repeated retraining procedure diverges and will not converge to a performatively stable equilibrium.

- By adding an extra regularization term $\frac{\alpha}{2}\|x_i\|_2^2$ to the loss function, we have the closed-form solution at time step $t$: $x_i^t = \frac{\theta_i^T (z_i^0 + \mu \theta_i x_i^{t-1})}{\theta_i^T \theta_i + \frac{\alpha}{2}}$. We can then calculate the difference between models in neighboring iterations:

$$|x_i^t - x_i^{t-1}| = |\frac{\theta_i^T (z_i^0 + \mu \theta_i x_i^{t-1})}{\theta_i^T \theta_i + \frac{\alpha}{2}} - \frac{\theta_i^T (z_i^0 + \mu \theta_i x_i^{t-2})}{\theta_i^T \theta_i + \frac{\alpha}{2}}| = \frac{|\mu \theta_i^T \theta_i||x_i^{t-1} - x_i^{t-2}|}{|\theta_i^T \theta_i + \frac{\alpha}{2}|}. \tag{86}$$

  In this case, even if $|\mu| > 1$, we can set $\alpha > 2(|\mu|-1)\theta_i^T \theta_i$ to enforce $|\mu \theta_i^T \theta_i| < |\theta_i^T \theta_i + \frac{\alpha}{2}|$. Consequently, it ensures $|x_i^t - x_i^{t-1}| < |x_i^{t-1} - x_i^{t-2}|$, thereby guaranteeing convergence to a performatively stable equilibrium.

## I  Proof of Theorem 4

For any $\gamma$-strongly convex regularization term $R(\cdot)$, the regularized game becomes:

$$\bar{\mathcal{G}}^R(Y) := \left( \underset{Z_1 \sim \mathcal{D}_1(Y)}{\mathbb{E}} [\ell_1(x_1, x_{-1}, Z_1) + R(x_1)], \cdots, \underset{Z_n \sim \mathcal{D}_n(Y)}{\mathbb{E}} [\ell_n(x_n, x_{-n}, Z_n) + R(x_n)] \right). \tag{87}$$

Since the regularized game $\bar{\mathcal{G}}^R(Y)$ is $(\psi + \gamma)$-strongly monotone, the individual gradient $\nabla_X \bar{\mathcal{G}}^R(Y)$ satisfies:

$$(\nabla_{X^*} \bar{\mathcal{G}}^R(Y) - \nabla_{X^R} \bar{\mathcal{G}}^R(Y))^T (X^* - X^R) \geq (\psi + \gamma) \|X^* - X^R\|_2^2. \tag{88}$$

Applying the Cauchy-Schwarz inequality (Hunter & Nachtergaele, 2001) to Eq. (88), we obtain:

$$\begin{aligned}
&\|\nabla_{X^*} \bar{\mathcal{G}}^R(Y) - \nabla_{X^R} \bar{\mathcal{G}}^R(Y)\|_2 \|X^* - X^R\|_2 \\
&\geq (\nabla_{X^*} \bar{\mathcal{G}}^R(Y) - \nabla_{X^R} \bar{\mathcal{G}}^R(Y))^T (X^* - X^R) \\
&\geq (\psi + \gamma) \|X^* - X^R\|_2^2, \\
&\Longrightarrow \|\nabla_{X^*} \bar{\mathcal{G}}^R(Y) - \nabla_{X^R} \bar{\mathcal{G}}^R(Y)\|_2 \geq (\psi + \gamma) \|X^* - X^R\|_2.
\end{aligned} \tag{89}$$

By first-order optimality conditions, we have:

$$\begin{aligned}
\nabla_{X^*} \mathcal{G}(Y) = \mathbf{0} &\Longrightarrow \nabla_{X^*} \bar{\mathcal{G}}^R(Y) = \nabla_{X^*} R(X^*), \\
\nabla_{X^R} \bar{\mathcal{G}}^R(Y) &= \mathbf{0}.
\end{aligned} \tag{90}$$

Substituting $\nabla_{X^*} \bar{\mathcal{G}}^R(Y)$ and $\nabla_{X^R} \bar{\mathcal{G}}^R(Y)$ into Eq. (89) yields:

$$\|\nabla_{X^*} R(X^*)\|_2 \geq (\psi + \gamma) \|X^* - X^R\|_2 \Longrightarrow \frac{\|\nabla_{X^*} R(X^*)\|_2}{(\psi + \gamma)} \geq \|X^* - X^R\|_2. \tag{91}$$

Considering the gradient of $R(\cdot)$ is $L$-Lipschitz continuous, we have:

$$\begin{aligned}
&\|\nabla_X R(X) - \nabla_{X'} R(X')\|_2 \leq L \|X - X'\|_2, \\
&\Longrightarrow \|\nabla_X R(X)\|_2 = \|\nabla_X R(X) - \nabla_{\mathbf{0}} R(\mathbf{0}) + \nabla_{\mathbf{0}} R(\mathbf{0})\|_2 \\
&\leq \|\nabla_X R(X) - \nabla_{\mathbf{0}} R(\mathbf{0})\|_2 + \|\nabla_{\mathbf{0}} R(\mathbf{0})\|_2 \leq L \|X\|_2 + \|\nabla_{\mathbf{0}} R(\mathbf{0})\|_2.
\end{aligned} \tag{92}$$

Combining Eqs. (91) and (92), we have:

$$\frac{L \|X^*\|_2 + \|\nabla_{\mathbf{0}} R(\mathbf{0})\|_2}{(\psi + \gamma)} \geq \frac{\|\nabla_{X^*} R(X^*)\|_2}{(\psi + \gamma)} \geq \|X^* - X^R\|_2. \tag{93}$$

This indicates that the distance between the original and regularized equilibria, i.e., $\|X^* - X^R\|_2$, is upper bounded by $\frac{L \|X^*\|_2 + \|\nabla_{\mathbf{0}} R(\mathbf{0})\|_2}{(\psi + \gamma)}$.

For $\gamma$-strongly convex $R(\cdot)$, by definition:

$$\begin{aligned}
R(X) - R(X') &\geq \nabla_{X'} R(X')^T (X - X') + \frac{\gamma}{2} \|X - X'\|_2^2, \\
R(X') - R(X) &\geq \nabla_X R(X)^T (X' - X) + \frac{\gamma}{2} \|X - X'\|_2^2.
\end{aligned} \tag{94}$$

Summing these two inequalities yields:

$$0 \geq (\nabla_{X'} R(X') - \nabla_X R(X))^T (X - X') + \gamma \|X - X'\|_2^2. \tag{95}$$

By the Cauchy-Schwartz inequality (Hunter & Nachtergaele, 2001) and $L$-Lipschitz continuity of the gradient:

$$
\begin{aligned}
(\nabla_X R(X) - \nabla_{X'} R(X'))^T (X - X') &\leq \|\nabla_{X'} R(X') - \nabla_X R(X)\|_2 \|X - X'\|_2 \leq L\|X - X'\|_2^2, \\
\Longrightarrow (\nabla_{X'} R(X') - \nabla_X R(X))^T (X - X') &\geq -L\|X - X'\|_2^2.
\end{aligned}
\tag{96}
$$

Combining Eqs. (95) and (96):

$$
0 \geq -L\|X - X'\|_2^2 + \gamma\|X - X'\|_2^2 \Longrightarrow L \geq \gamma.
\tag{97}
$$

The upper bound of $\|X^* - X^R\|_2$, i.e., $\frac{L\|X^*\|_2 + \|\nabla_{\mathbf{0}} R(\mathbf{0})\|_2}{(\psi + \gamma)}$, is minimized when: (1) $L$ is minimized, i.e., $L = \gamma$, and (2) $\|\nabla_{\mathbf{0}} R(\mathbf{0})\|_2 = 0$. Clearly, the proposed quadratic regularizer $R(X) = \frac{\gamma}{2}\|X\|_2^2$ satisfies these two conditions, as it is $\gamma$-Lipschitz in gradient, and has $\nabla R(\mathbf{0}) = \mathbf{0}$, yielding the tightest upper bound $\frac{\gamma\|X^*\|_2}{(\psi + \gamma)}$. Thus, $R(X) = \frac{\gamma}{2}\|X\|_2^2$ minimizes the upper bound of the distance $\|X^* - X^R\|_2$ among all $\gamma$-strongly convex regularizers. $\blacksquare$

## J  Experimental Details

### J.1  Parameter Sensitivity Analysis

We evaluate the robustness of $\text{SIR}^2$ to $c$, a key parameter that controls the convergence rate to performatively stable equilibria. Theorem 3 guarantees convergence for $c > 2$. We test $c \in \{2.1, 4, 6, 8, 10\}$ in the revenue maximization game, averaging total revenue over 10 trials across 15 iterations for $\mu_A \in \{0.25, 0.5, 0.75, 1.0\}$ (Fig. 4). Our method exhibits robust performance, converging within approximately 5 iterations with revenue variance below 5% despite large variations in $c$.

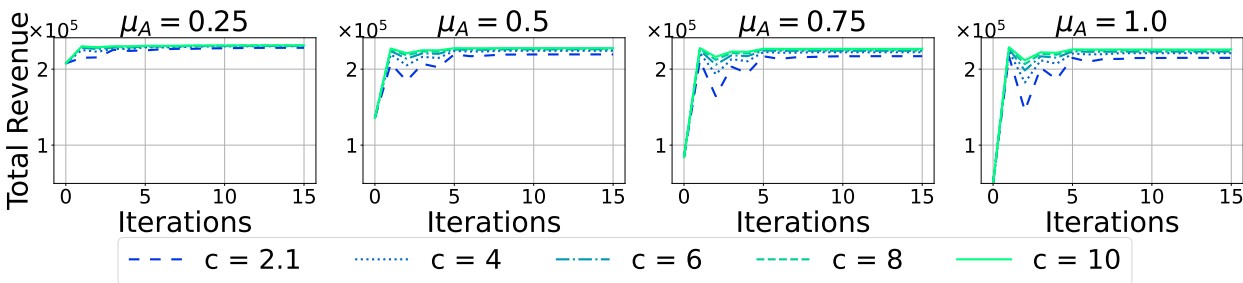

Figure 4: Average total revenue of $\text{SIR}^2$ with different $c$ in Revenue Maximization in Ride-Share Markets with $\mu_A = \{0.25, 0.5, 0.75, 1.0\}$ over 10 trails.

### J.2  Comparison of Wall-Clock Time

We present the wall-clock time comparison between $\text{SIR}^2$ and the baseline algorithms in Figs. 5-7. The results show that although $\text{SIR}^2$ requires more time in each iteration due to the computation of the full equilibrium, it converges in fewer iterations compared with baselines. Consequently, the total wall-clock time for $\text{SIR}^2$ to converge is comparable to that of the baselines. Moreover, within the same time period, although the baseline methods complete more iterations, $\text{SIR}^2$ achieves better performance.

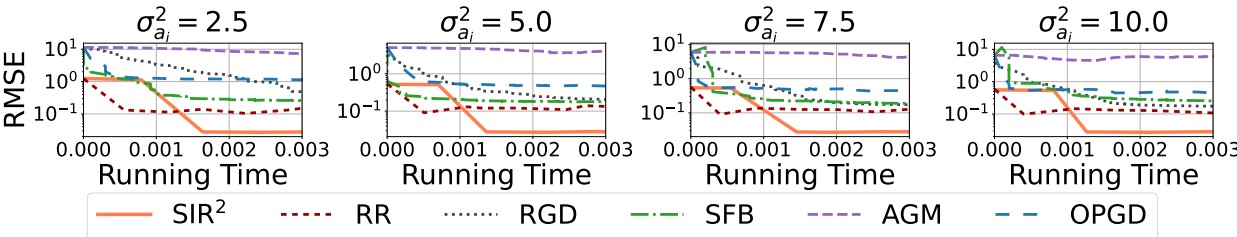

Figure 5: Sum RMSE comparison for SIR$^2$ and five baselines on the Prediction Error Minimization Game over wall-clock time.

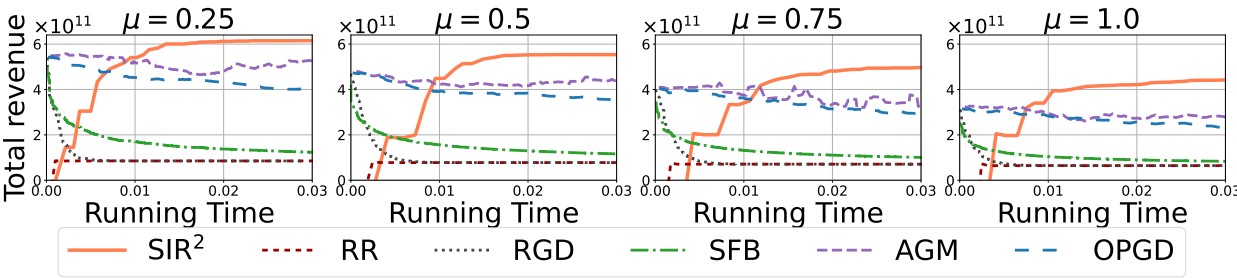

Figure 6: Total revenue comparison for SIR$^2$ and five baselines on the crude oil trade over wall-clock time.

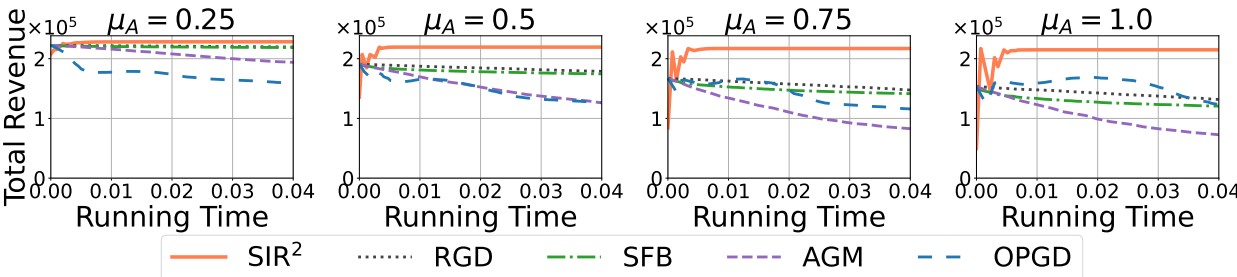

Figure 7: Total revenue comparison for SIR$^2$ and five baselines on the Ride-Share Markets over wall-clock time.

### J.3    Additional Experimental Settings

Regarding the machine configuration, all experiments were conducted on a laptop equipped with a 13th Gen Intel Core i9-13900HX CPU (24 threads) and 64 GB of RAM. The Python implementation required 50 MB of storage, and all experiments were completed within 20 minutes.

In this work, we compare our method with five state-of-the-art methods, including Repeated Retraining (RR) (Narang et al., 2023), RGD (Narang et al., 2023), SFB (Cutler et al., 2024), AGM (Narang et al., 2023) and OPGD (Zhu et al., 2023). Notably, the first two methods, RGD and SFB, aim for performatively stable equilibria, while the latter two, AGM and OPGD, target Nash equilibria. Note that prior work on cooperative or constrained settings (Wang et al., 2023; Li et al., 2022; Yan & Cao, 2024) is not directly comparable due to our focus on non-cooperative games without constraints.

In each experiment, each player $i \in [n]$ deploys an initial decision $x_i^0$ based on the initial dataset $Z_i^0 \sim \mathcal{D}_i(X^0)$. At time step $t$, each player $i$ optimizes the decision $x_i^t$ under the distribution $\mathcal{D}_i(X^{t-1})$ and then collects samples from the data distribution induced by the joint decision $X^t$. We evaluate the performance of decision $X^t$ on the induced distribution $\mathcal{D}_i(X^t)$, using root mean square error (RMSE) for prediction games and total

revenue for ride-share markets. Regarding the stopping criterion, while our algorithm typically stabilizes within a few iterations, the stabilized level varies across datasets, leading us to use a fixed time step ($T = 100$ for prediction games and Cournot competition, and $T = 1000$ for ride-share markets) as a stopping criterion rather than a fixed error/revenue level. We repeat our experiments 10 times for each setting and report the average results of all methods.

### J.4 Additional Experimental Results on Prediction Error Minimization Game

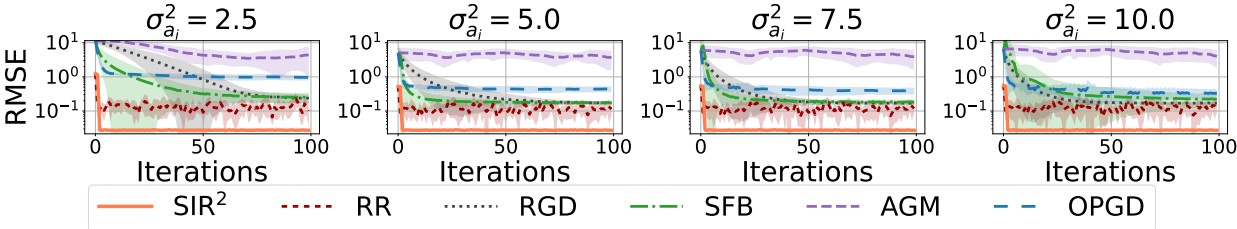

Figure 8: Performance comparison (in terms of the sum RMSE and standard deviation) of our proposed method, $\text{SIR}^2$, against five baseline approaches: RR, RGD, SFB, AGM and OPGD on the Prediction Error Minimization Game with $\sigma^2_{a_i} = \{0.25, 0.5, 0.75, 1.0\}$ across iterations.

### J.5 Cournot Competition Game Details

To solve this equilibrium, we first calculate the best response of each country by the first-order optimality condition $\forall i \in [n]$:

$$\nabla_{x_i} \ell_i(z, x_i, x_{-i}) = 2bx_i + 2bq_i + b\sum_{j \neq i}^{n}(q_j + x_j) + c - z = 0,$$
$$\Longrightarrow 2bx_i + b\sum_{j \neq i}^{n} x_j = -2bq_i - b\sum_{j \neq i}^{n} q_j - c + z. \tag{98}$$

Then, we solve the Nash equilibrium by solving the above equations simultaneously for all countries:

$$2bx_1 + b\sum_{j \neq 1}^{n} x_j = -2bq_1 - b\sum_{j \neq 1}^{n} q_j - c + z$$
$$2bx_2 + b\sum_{j \neq 2}^{n} x_j = -2bq_2 - b\sum_{j \neq 2}^{n} q_j - c + z$$
$$\vdots \tag{99}$$
$$2bx_n + b\sum_{j \neq n}^{n} x_j = -2bq_n - b\sum_{j \neq n}^{n} q_j - c + z$$

We reformulate it as system of linear equations:

$$\mathbf{A}X = \mathbf{b}, \tag{100}$$

with the matrix $\mathbf{A}$ and vector $\mathbf{b}$ defined as:

$$\mathbf{A} = \begin{bmatrix} 2b & b & \cdots & b \\ b & 2b & \cdots & b \\ b & \vdots & \ddots & b \\ b & b & \cdots & 2b \end{bmatrix}, \mathbf{b} = \begin{bmatrix} -2bq_1 - b\sum_{j \neq 1}^{n} q_j - c + z \\ -2bq_2 - b\sum_{j \neq 2}^{n} q_j - c + z \\ \vdots \\ -2bq_n - b\sum_{j \neq n}^{n} q_j - c + z \end{bmatrix}. \tag{101}$$

Since the eigenvalue of $\mathbf{A}$ is $b(n + 1)$ and $b$ with $b > 0$, this Cournot competition is $b$-strongly monotone, ensuring the existence of a unique Nash equilibrium, which can be solved by:

$$X = \mathbf{A}^{-1}\mathbf{b}. \tag{102}$$

In addition, the gradient-based methods (RGD, SFB, AGM, and OPGD) update their decisions via the gradient $\nabla_{x_i} \ell_i(z, x_i, x_{-i})$.

### J.6 Additional Experimental Results on Cournot Competition

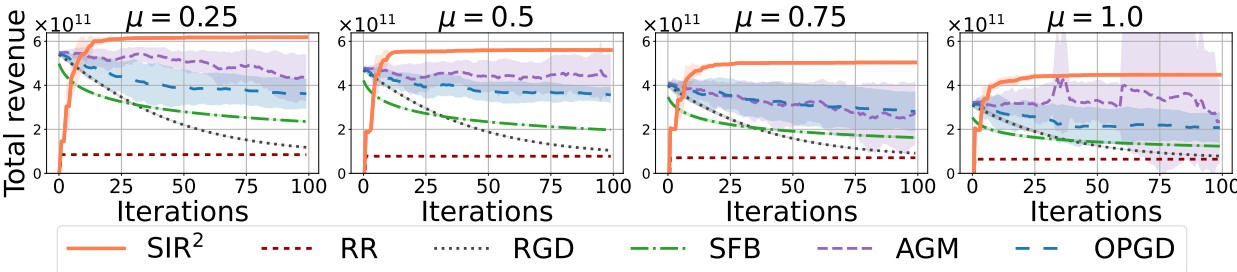

Figure 9: Performance comparison (in terms of the total revenue and the standard deviation) of our proposed method, $\text{SIR}^2$, against five baseline approaches: RR, RGD, SFB, AGM and OPGD on the crude oil trade with $\mu = \{0.25, 0.5, 0.75, 1.0\}$ across iterations.

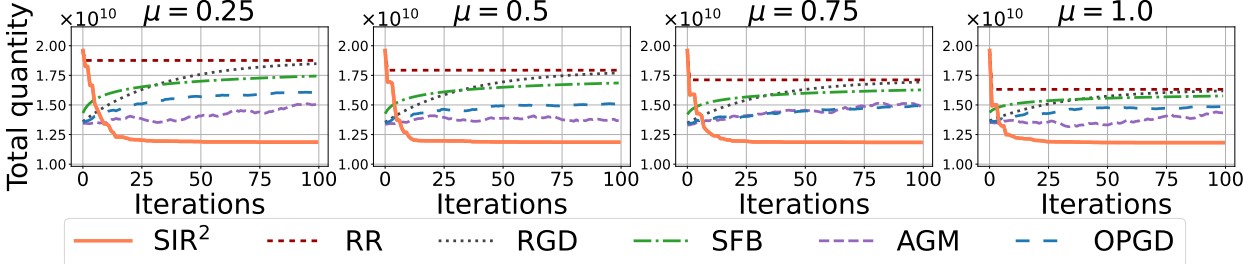

Figure 10: Quantity of each company obtained by our proposed method, $\text{SIR}^2$, and five baseline approaches: RR, RGD, SFB, AGM and OPGD on the crude oil trade with $\mu = \{0.25, 0.5, 0.75, 1.0\}$ across iterations.

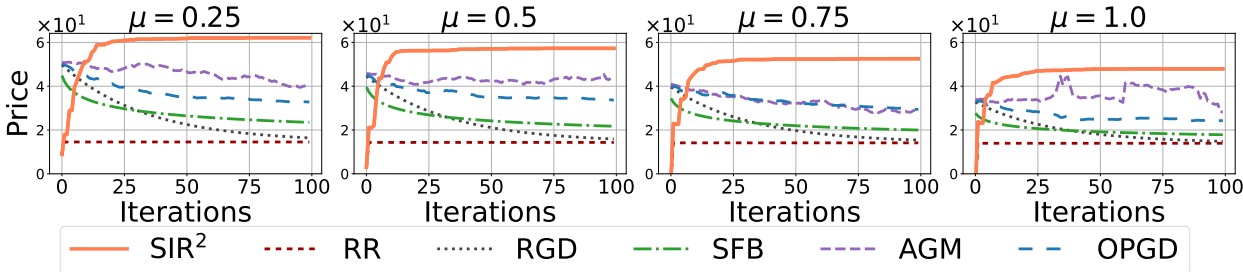

Figure 11: Crude oil price of each company obtained by our proposed method, $\text{SIR}^2$, and five baseline approaches: RR, RGD, SFB, AGM and OPGD on the crude oil trade with $\mu = \{0.25, 0.5, 0.75, 1.0\}$ across iterations.

### J.7 Revenue Maximization Game Details

In this game, two companies ($n = 2$), Lyft and Uber, set prices adjustments to maximize revenue in 11 Boston locations. The dataset's ride-share records are grouped into price intervals $[10, 15), [15, 20), [20, 25), [25, 30),$

$[30, +\infty)$, represented by their lower bounds ($p = 10, 15, 20, 25, 30$) (Narang et al., 2023). The game is conducted independently per price interval. Records are assigned to intervals according to their actual prices. For each interval, we calculate the demand for each company across 11 locations, denoted as $z_i \in \mathbb{R}^{11}$. For each price interval, company $i$ sets its price adjustments $x_i \in \mathbb{R}^{11}$ for the 11 locations, with each element in $x_i$ representing the price adjustment in the corresponding location, yielding the actual price vector $x_i + p$. Each company $i$ seeks to maximize its revenue $z_i^T x_i$ in the price interval by minimizing the regularized loss function: $\ell_i(x_i, z_i) = -z_i^T x_i + \frac{\alpha}{2}\|x_i\|_2^2$, where $\alpha > 0$ is the regularization parameter.

### J.8    Additional Experimental Results on Revenue Maximization Game

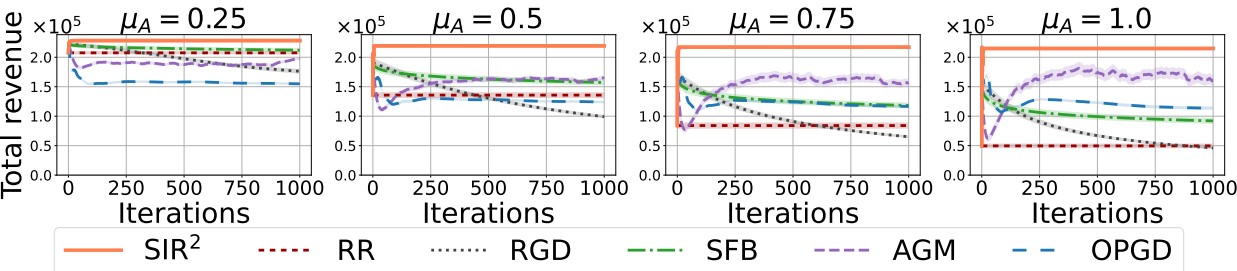

Figure 12: Performance comparison (in terms of the total revenue and the standard deviation) of our proposed method, SIR$^2$, against five baseline approaches: RR, RGD, SFB, AGM and OPGD on the Ride-Share Markets with $\mu_A = \{0.25, 0.5, 0.75, 1.0\}$ across iterations.

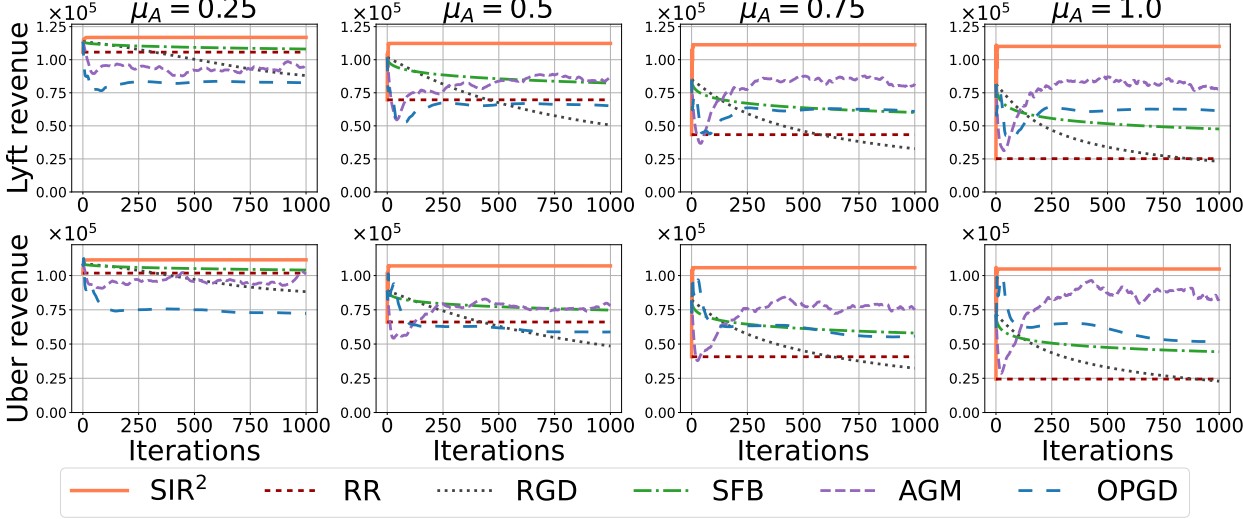

Figure 13: Revenues of each company obtained by our proposed method, SIR$^2$, and five baseline approaches: RR, RGD, SFB, AGM and OPGD on the Ride-Share Markets with $\mu_A = \{0.25, 0.5, 0.75, 1.0\}$ across iterations.

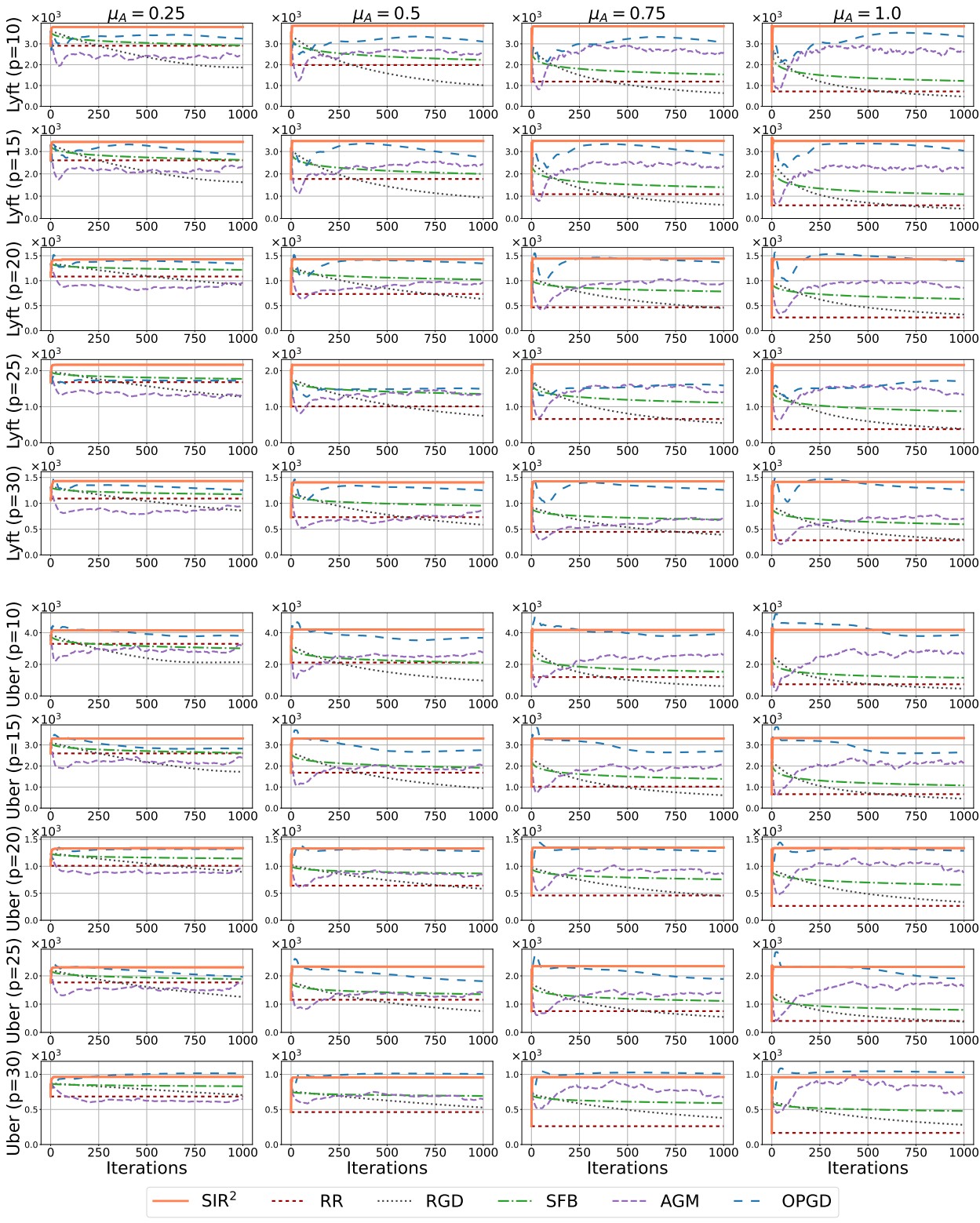

Figure 14: Demands of each company in each price interval obtained by our proposed method, $SIR^2$, and five baseline approaches: RR, RGD, SFB, AGM and OPGD on the Ride-Share Markets with $\mu_A = \{0.25, 0.5, 0.75, 1.0\}$ across iterations.

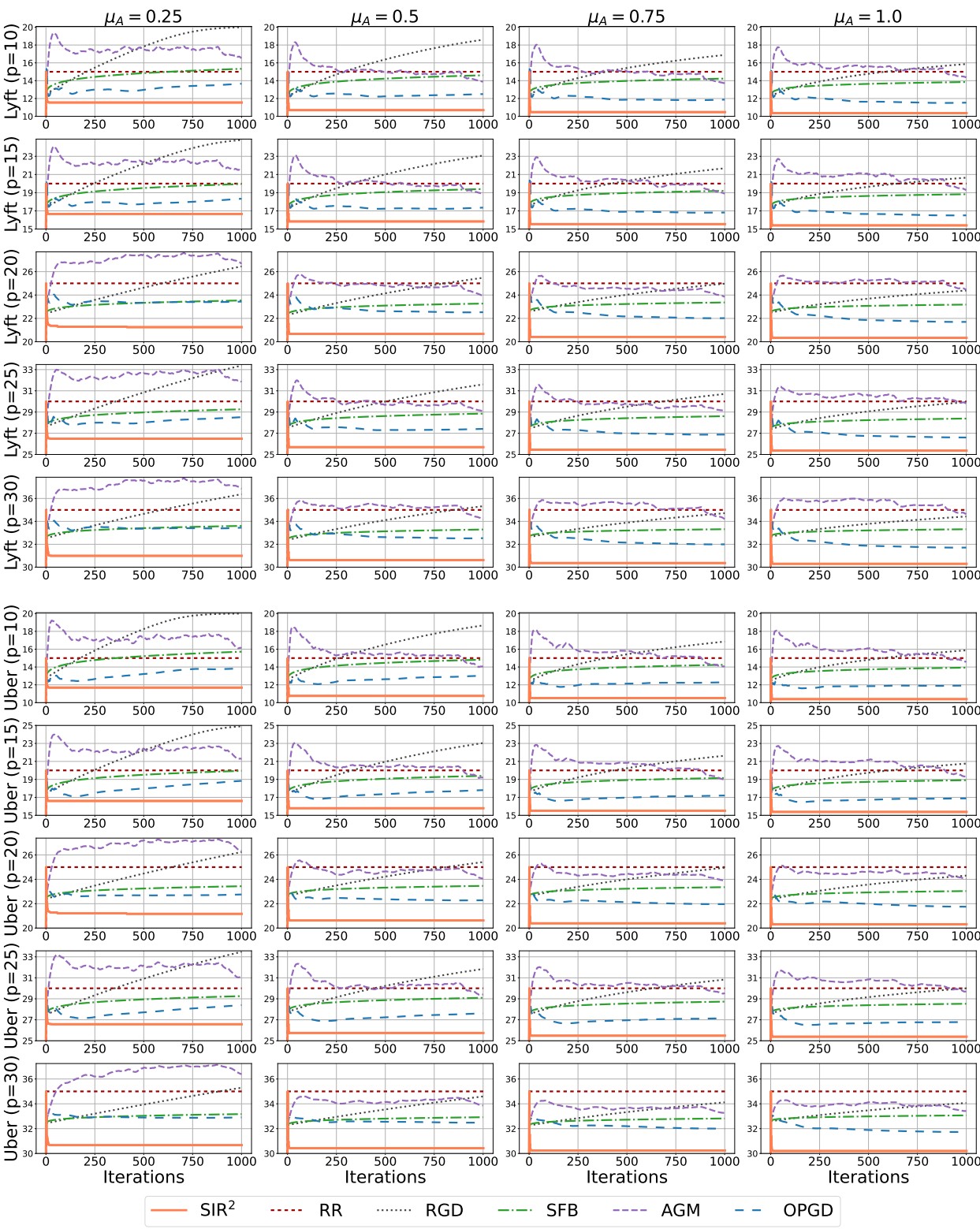

Figure 15: Average prices of each company in each price interval set by our proposed method, $\text{SIR}^2$, and five baseline approaches: RR, RGD, SFB, AGM and OPGD on the Ride-Share Markets with $\mu_A = \{0.25, 0.5, 0.75, 1.0\}$ across iterations.

