# OpenReview forum: "Towards Performatively Stable Equilibria in Decision-Dependent Games for Arbitrary Data Distribution Maps"
_TMLR — Rejected by TMLR_

### Review · Reviewer_H5NA · 2025-09-16

**Summary Of Contributions:**

This paper studies methods for computing performatively stable equilibria in a multi-agent setting. Existing works rely on the smoothness of the loss functions, which is hard to verify in practice. The authors propose a different notion called “sensitivity,” which replaces the smoothness assumption and can be estimated more easily. They further show that, for strongly monotone games, the repeated retraining method converges to the performatively stable equilibrium under suitable assumptions.


# Strengths
The motivation is clear, and the related work is adequately addressed.
The theoretical results are mostly sound, and the proofs are easy to follow.

# Weaknesses
Here are some technical questions for which I would appreciate clarification from the authors.

1. **Assumption of Theorems 2 and 3.** In both theorems, the strong monotonicity parameter $\alpha$ is required to be larger than the sensitivity parameters $\hat{\varepsilon}_i$. It is stated at the end of Section 3 that this assumption can be satisfied by tuning $\alpha$ based on the measured $\hat{\varepsilon}_i$. I don’t understand how this is always possible, since we cannot choose $\alpha$ to be arbitrarily large.
2. **Convergence of Algorithm 1 (SIR$^2$).** In Algorithm 1, at each iteration a strongly convex regularizer is added to each player's loss function, and the repeated retraining method is applied to the regularized game $\bar{\mathcal{G}}$. However, it is stated that SIR$^2$ converges to a performatively stable equilibrium. This is somewhat confusing to me. Does SIR$^2$ converge to the equilibrium of the original game $\mathcal{G}$ or of the regularized game $\bar{\mathcal{G}}$?
3. **Asymptotic convergence in Theorem 2.** Theorem 2 only guarantees asymptotic convergence to the performatively stable equilibrium, whereas Theorem 3 provides a non-asymptotic complexity bound. Is it possible to prove a non-asymptotic convergence bound under the setting of Theorem 2?
4. **Unclear statement of Theorem 4.** It is unclear to me what the phrase “the quadratic form $R$ minimizes the upper bound of $\lVert X^\ast - X^R \rVert$ …” means.


# Minor Questions
1. **Selection of the strong monotonicity parameter in experiments.** SIR$^2$ (Algorithm 1) requires the strong monotonicity parameter $\psi$ as input. How is this parameter selected in the numerical experiments?
2. In Eq. (27), should there be expectations $\mathbb{E}_{\{z_i, \theta_i\} \sim \mathcal{D}_i(X)}$ inside the norms?

**Audience:**

Yes

**Audience Explanation:**

There are several recent works that study performative prediction in the single-agent setting, the multi-agent setting, and reinforcement learning. This work relaxes the smoothness assumption and will be of interest to some readers.

**Broader Impact Concerns:**

There are no ethical concerns.

**Claims And Evidence:**

Yes

**Claims Explanation:**

For the theoretical parts, I have skimmed through the proof sketches and the full proofs of Theorems 1 and 2. They are easy to follow and seem convincing to me.
My only concern is the convergence of the algorithm SIR$^2$ (Algorithm 1) mentioned above, and I would appreciate any clarification from the authors.

**Requested Changes:**

Please refer to the weaknesses listed above.

---

> ### Author Response · Authors · 2025-10-30
> **Response 1**
>
> We sincerely thank the reviewer for the valuable time and effort dedicated to reviewing our manuscript, as well as the acknowledgment of motivation, literature review, and theoretical analysis.
>
> We also greatly appreciate the constructive feedback and detailed suggestions expertly provided on convergence guarantees and algorithm implementation. Below, we respond to the reviewer's valuable feedback. All revisions made in the manuscript are highlighted in blue for easy identification.
>
> ---
>
> - Regarding the assumption of Theorems 2 and 3, we appreciate the comment and opportunity to clarify our statement.
>
>   It is correct that parameter $\alpha$ should be larger than $\sqrt{\sum_{i=1}^n\hat{\varepsilon}_i^2}$ in Theorem 2 and $2\sqrt{\sum _{i=1}^n\hat{\varepsilon} _i^2}$ in Theorem 3 to ensure the convergence to a performatively stable equilibrium. Although the strong monotonicity parameter $\psi$ is determined by the original game structure, we can tune $\alpha$ by adding a regularizer $\frac{\gamma}{2}||x_i|| _2^2$ to each loss function to ensure the convergence, as shown in Algorithm 1. By doing so, we can tune the strong monotonicity parameter from $\psi$ to $\psi+\gamma=\alpha$, ensuring $\alpha>2\sqrt{\sum _{i=1}^n\hat{\varepsilon} _i^2}$ in practice.
>
>   **Revision:** We have explained that the regularization approach can tune the strong monotonicity parameter $\alpha$ in Section 3.3 (last paragraph).
>
> ---
>
> - Regarding the convergence of our algorithm, we thank the reviewer for this insightful comment.
>
>   We confirm that Algorithm 1 converges to the equilibrium of the regularized game. This approach addresses that the repeated retraining in the original game may fail to converge to a performatively stable equilibrium, when the strong monotonicity parameter $\psi$ is small and the distribution maps are highly sensitive (large $\hat{\varepsilon}_i$). By introducing a regularizer $\frac{\gamma}{2}||x_i|| _2^2$ to each loss function, we ensure the convergence of a performatively stable equilibrium according to our theoretical results.
>
>   **Revision:** We have clarified that our algorithm converges to an equilibrium of the regularized game in Section 4 (last paragraph), and provided an example to show that repeated retraining for the original game may fail to converge in Appendix H.
>
> ---
>
> - Regarding the asymptotic convergence in Theorem 2, we appreciate the reviewer for the interest in our theoretical results. We can establish this result based on our current theoretical framework. Since we proved $X^{PS}\in\mathcal{X} _{RR}$ in Theorem 2, we can set $X'=Nash(X')=X^{PS},X=X^{t-1},Nash(X)=X^t$ in Eq. (15). Then we have:
>   $$||X^t-X^{PS}|| _2\le\frac{\sqrt{\sum _{i=1}^n\hat{\varepsilon} _i^2}}{\alpha}||X^{t-1}-X^{PS}|| _2\le(\frac{\sqrt{\sum _{i=1}^n\hat{\varepsilon} _i^2}}{\alpha})^t||X^0-X^{PS}||_2.$$
>
>   To ensure $||X^t-X^{PS}|| _2\le\delta$, we obtain:
>   $$||X^t-X^{PS}|| _2\le\delta,\forall t\ge\frac{\log(\frac{\delta}{||X^0-X^{PS}|| _2})}{\log(\frac{\sqrt{\sum _{i=1}^n\hat{\varepsilon} _i^2}}{\alpha})}.$$
>
> ---
>
> - Regarding the statement in Theorem 4, we thank the reviewer for providing the opportunity to clarify this point.
>
>   Since our algorithm converges to a performatively stable equilibrium of the regularized game, this equilibrium can differ from the equilibrium of the original game. Thus, we aim to ensure that this equilibrium is as close as possible to the equilibrium of the original game.
>
>   Theorem 4 shows that, among all $\gamma$-strongly convex regularizers $\mathcal{R}$, the quadratic regularizer $R(x)=\frac{\gamma}{2}||x|| _2^2$ achieves the smallest possible upper bound on the distance between the original equilibrium and the regularized equilibrium.
>
>   To improve clarity, we have revised Theorem 4.
>
>   **Revision:** We have explained the motivation of Theorem 4 in Section 4 (last paragraph) and revised Theorem 4 in Section 4.
>
> ---

---

> ### Author Response · Authors · 2025-10-30
> **Response 2**
>
> - Regarding the strong monotonicity parameter $\psi$ in our experiment, we thank the reviewer for the suggestion.
>
>   - The games in Sections 5.1 and 5.3 are separable games, where the loss function of each player $i$ depends only on their own decision $x_i$ and sample $Z_i$, i.e., $\ell_i(Z_i,x_i,x_{-i}) = \ell_i(Z_i,x_i)$. For such games, the strong monotonicity parameter $\psi$ equals the smallest strong convexity parameter among all players' loss functions. Parameter $\psi=0$ in Section 5.1 and $\psi=\alpha$ in Section 5.3.
>
>   - The game in Section 5.2 is not separable. We compute the strong monotonicity parameter $\psi$ by analyzing the Jacobian matrix $\mathcal{J}$ of the individual gradient $\nabla_X\mathcal{G}(Y)$. The gradient of player $i$'s loss function is $\nabla _{x _i}\ell _i=2bx _i+2bq _i+b\sum _{j\neq i}^n(q _j+x _j)+c-z$. The resulting Jacobian matrix is a symmetric matrix with diagonal entries $2 b$ and non-diagonal entries $b$. According to [1], the strong monotonicity parameter $\psi$ is the smallest eigenvalue of $\mathcal{J}$. The eigenvalues of $\mathcal{J}$ are $b(n-1)$ and $b$. Therefore, for $n = 28$ as in our experiment, the smallest eigenvalue is $b$, which implies $\psi = b$.
>
>   **Revision:** We have clarified the value of $\psi$ in Sections 5.1.1, 5.2.1, and 5.3.1.
>
> ---
>
> - Regarding Eq. (27) (Eq. (28) in the revised manuscript), we thank the reviewer for the observation.
>
>   We clarify that there should not be expectations inside the norm of Eq. (28). $\beta_i$-smoothness requires each individual instance to satisfy the Lipschitz condition of the loss function gradient rather than the expectation of the sample set or distribution.
>
> ---
>
> **Reference:**
>
> [1] R Tyrrell Rockafellar and Roger JB Wets. Variational analysis. Springer, 1998.

---

### Review · Reviewer_UCcE · 2025-10-14

**Summary Of Contributions:**

Summary:

The paper studies the problem of finding performatively stable equilibria in decision-dependent games, i.e. fixed points under repeated retraining. The paper introduces a gradient based $\epsilon_i$-sensitivity condition to overcome existing $\beta$-smoothness assumption in the existing works. Convergence results are provided for repeated retraining (RR) under strong monotonicity. Since the $\epsilon$ sensitivity conditions cannot always be known Sensitivity-Informed Repeated Retraining (SIR) is proposed to estimate $\epsilon$-sensitivity condition from samples under the RR paradigm.

Strengths:

The paper provides a comprehensive empirical comparison across different domains.

Weaknesses:

The main weaknesses are incorrect portrayal of related work, including incorrect novelty, and the theoretical claims that do not seem to be fully correct. Some other minor weaknesses include assumptions made in the proofs that are not fully reflected in the main paper, as well as algorithmic limitations when compared to other methods. Below I go into detail on these points.

**Audience:**

No

**Audience Explanation:**

Due to the strong overlap of some of the main results with related work and the implicit use of the central limit theorem to establish the convergence results, it is hard to argue that this paper would be of interest to the community.

**Claims And Evidence:**

No

**Claims Explanation:**

Incorrect portrayal of the related work by Narang et al., 2023 including the same result that is not properly cited.

The paper cites Narang et al., 2023 as an existing work that that is limited by $\beta$-smoothness assumptions, however, Theorem 1 in this paper is exactly Theorem 7 in Narang et al., 2023. The result and the proof are the same. To clarify, Narang et al., 2023 do assume that the losses satisfy a Lipchitz assumption for which this paper refers to as $\beta$-smoothness, however these assumptions are only used for the result in Lemma 5 in Narang et al. which is identical to the $\epsilon_i$-sensitivity condition in this paper. Therefore, using the notion of $\epsilon_i$-sensitivity to establish convergence of repeated retraining is not novel (referred to as Deviation in the vector of individual gradients Lemma 5 in Narang et al). Moreover, the results seem different because Narang et al establishes a contraction of the mapping $Nash(X)$ while Theorem 1 seems to be showing a contraction of successive points in training $\|X^{t+1}-X^t\|\leq \rho \|X^t-X^{t-1}\|$ but it is indeed the same result as seen by eq 14. This statement is weaker as  it does not provide an existence of solution or necessarily guarantee convergence of the sequence without using other results such as Theorem 2.

Claims on high-probability results for finite-samples:

The paper claims to provide high-probability convergence results for a finite-number of samples however the proofs do not give insight on how many samples are needed. The reason being is that the proof uses the Central Limit Theorem which already includes a limit of infinite samples (see eq 43 in the appendix), effectively, for the results to hold for any number of samples the paper is implicitly assuming that the noise in gradient estimation is Normally distributed. This assumption is used through the paper to establish all the probabilistic results. Moreover, the result of Lemma 1 provides some form of concentration of noise for one timestep but this result is used for all time steps in Theorem 3. Therefore, one should include a union bound, that is some argument that is true across multiple timesteps at once. For example, if two events hold with probability 0.9 it is not true that they will both hold simultaneously with probability 0.9. From my understanding, and correct me if I am wrong, such a logic is missing in Theorem 3.

Algorithmic limitations:

The convergence of the SIR algorithm is argued via Theorem 3 which suggests that at each step SIR is solving a Nash equilibrium problem, however, this is inconsistent with the pseudo-code which suggests a best-response computation. Assuming one requires computing a Nash as suggested by the Theory, in general this is a computationally challenging problem, which seems to be a big limitation when compared to say other gradient based approaches. Additionally, the paper does not provide explanations (as far I can tell), on how this Nash is solved for in practice. Furthermore, I would expect that because we usually cannot solve for exact Nash in practice that this is also part of the limitations of the Theory, in general one would also want to reason about inexactly solving for the Nash equilibrium in RR at each step. Moreover, to make the empirical comparisons fair one should also compare wall-clock time as one iteration of SIR might be more expensive when compared to alternatives.

**Requested Changes:**

Critical changes:
+ revised discussion and comparison with Narang et al., 2023 including
    - proper citation of Theorem 7
    - attribution of $\epsilon$-sensitivity to Lemma 5
+ explicitly mention the use of the central limit theorem to analyze the stochasticity of SIR
+ address theoretical concerns including using a union bound
+ discuss algorithmic limitations especially the step of solving for a Nash equilibrium (including figured with wall clock time in experiments)

Other suggestions /questions:
+ $\beta$-smoothness should be formally defined in the main text as it is referred to often, in the Appendix two conditions are stated for the same definition it seems. It would improve clarity of the discussion especially with respect to related work.
+ what does this mean?
> due to the iterative exploration of non-convex performative loss gradients.
+ I believe Theorem 2 is missing a step, I think you need to take the limit inside Nash() and argue that is continuous. This is guaranteed by Theorem 7 in Narang et al., 2023
+ When discussing the related works adding citations can be helpful for example it is unclear which works are referred to in the following statement
> Additionally, prior research has applied the W1 distance to quantify distribution shifts
+ If $\psi$ is too small than there might not exist a performative stable equilibria, is this a problem?

---

> ### Author Response · Authors · 2025-10-30
> **Response 1**
>
> We are sincerely grateful for the reviewer's commitment of valuable time in carefully evaluating our manuscript and for recognizing our empirical evaluation.
>
> We also greatly appreciate the constructive feedback and detailed suggestions expertly provided on the contribution and theoretical analysis. Below, we respond to the reviewer's valuable feedback. All revisions made in the manuscript are highlighted in blue for easy identification.
>
> ---
>
> - Regarding the discussion and comparison with [1], we appreciate the suggestion and for providing the chance to clarify this point. We apologize for the insufficient discussion on [1].
>
>   - We agree that [1] is a key relevant work, which is discussed in Sections 1, 2, 3, and 5. Our main contribution is to relax the unverifiable conditions in a performative setting. Our $\hat{\varepsilon}_i$-sensitivity is computable for arbitrary distribution maps, which allows our theorems to guide the design of an algorithm that converges to a performatively stable equilibrium.
>
>     We confirm that Theorems 1 and 2 show equivalent theoretical strength to Theorem 7 in [1]. Our Theorem 1 analyzes successive point dynamics without a condition like $\rho<1$ in Theorem 7 in [1], and our Theorem 2 establishes the existence and convergence of the performatively stable equilibrium under an analogous condition $\alpha>\sqrt{\sum_{i=1}^n\hat{\varepsilon} _i^2}$.
>
>   - We agree that $\hat{\varepsilon}_i$-sensitivity is connected to Lemma 5 in [1].
>   While $\hat{\varepsilon}_i$ serves as a proof step analogous to Lemma 5 in [1], their distinct definitions lead to different practical applicability. We reduce the theoretical requirements by eliminating unnecessary conditions (e.g., $\mathcal{Z} _i:=\cup _{X\in\mathcal{X}} \sup(\mathcal{D} _i(X))$), which are often unverifiable with unknown data distribution maps. By focusing on the repeated retraining procedure, the proposed $\hat{\varepsilon} _i$-sensitivity simultaneously enables performatively stable equilibrium convergence analysis and practical computation. Specifically, for arbitrary distribution map $\mathcal{D} _i(X)$, we can calculate $\hat{\varepsilon} _i$ during repeated retraining by:
>   $$\max _{t,t^\prime}\frac{||\mathbb{E} _{Z^t\sim\mathcal{D} _i(X^t)}\nabla _{x^t_i}\ell _i(x^t_i,x^t _{-i},Z^t_i)-\mathbb{E} _{Z^{t^\prime}\sim\mathcal{D} _i(X^{t^\prime})}\nabla _{x^t_i}\ell _i(x^t_i,x^t _{-i},Z^{t^\prime} _i)|| _2}{||X^t-X^{t^\prime}|| _2},\forall t,t^\prime = 1,2,\cdots$$
>   In contrast, $\beta_i$-smoothness may be unverifiable as shown in Appendix A.
>
>   **Revision:** We have discussed the difference between $\hat{\varepsilon}_i$ and parameters in existing work in Section 3.1 (last paragraph), the difference between our Theorems 1 and 2 and theorems in current works in Section 3.2 (last paragraph)
>
> ---
>
> - Regarding the use of the Central Limit Theorem, we appreciate the reviewer for the insightful suggestion. We completely agree with the reviewer that the Central Limit Theorem provides an asymptotic approximation to the normal distribution as the sample size $m\rightarrow\infty$. To make our work more rigorous and practically useful, we further introduce two approaches.
>
>   - The Berry–Esseen theorem quantifies the maximum error between the normal distribution and the distribution of the average gradient.
>   - Chebyshev's inequality bounds the probability that the average gradient deviates from the expected gradient, without assuming the distribution of the sample.
>
>   **Revision:** We have acknowledged that Lemma 1 and Theorem 3 rely on the Central Limit Theorem and hold as $m\rightarrow\infty$ in the footnote of Section 3.3. We have provided two lemmas (one using the Berry–Esseen theorem while another utilizing Chebyshev's inequality) and their corresponding proofs in Appendices E and F, and corresponding variants of Theorem 3 in Appendix G.2.
>
> ---
>
> - Regarding the probabilistic guarantee of Theorem 3, we thank the reviewer for pointing out this issue. We agree that the probability should consider all previous time steps, since the convergence $||X^t-X^{PS}||<\delta$ requires Eq. (22) to hold at every time step. The probability in Theorem 3 should be $\prod _{\mu=1}^t(F _{\chi^2(d)}(\frac{m _\mu\sum _{i=1}^n\hat{\varepsilon} _i^2\delta^2}{\sigma _\mu}))$, which incorporates all historical sample sizes. This ensures that $X^t$ will enter and stay in the $\delta$-ball around $X^{PS}$ with the stated probability.
>
>   **Revision:** We have revised the probability in Section 3.3 (Theorem 3 and proof) and Appendix G.
>
> ---
>
> **Reference:**
>
> [1] A. Narang, E. Faulkner, D. Drusvyatskiy, M. Fazel, and L. J. Ratliff. Multiplayer performative prediction: Learning in decision-dependent games. JMLR, 24(202):1–56, 2023.

---

> > ### Comment · Reviewer_UCcE · 2025-10-30
> > **Concerns with overlap on related work still not addressed**
> >
> > I thank the others for their response, however, my concern regarding overlap between Theorem 1 and Theorem 7 in [1] is still not addressed. More precisely, *Theorem 1 in this paper is exactly Theorem 7 in Narang et al., 2023. [1]*
> >
> > I would like to point out the following:
> > + Narang et. al show a contraction of repeated retraining under a condition that is equivalent to $\hat{\epsilon}_i$-sensitivity
> > + Theorem 1 skips one step in the proof by assuming $\hat{\epsilon}_i$-sensitivity
> > + The results are exactly the same, with the same contraction factor
> >
> > Here is a sketch of their proof:
> > 1. Use Lipschitz assumptions to prove $\hat{\epsilon}_i$-sensitivity holds (this is Lemma 5 in [1])
> > 2. With $\hat{\epsilon}_i$-sensitivity show that repeatedly solving for Nash is a contraction with factor $\rho = \frac{\sqrt{\sum_i^n \hat{\epsilon}_i}}{\alpha}$
> >
> > Note that step 2 is exactly Theorem 1.
> >
> > > We confirm that Theorems 1 and 2 show equivalent theoretical strength to Theorem 7 in [1]. Our Theorem 1 analyzes successive point dynamics without a condition like $\rho<1$ in Theorem 7 in [1], and our Theorem 2 establishes the existence and convergence of the performatively stable equilibrium under an analogous condition $\alpha>\sqrt{\sum_{i=1}^n\hat{\varepsilon} _i^2}$. and our Theorem 2 establishes the existence and convergence of the performatively stable equilibrium under an analogous condition
> >
> > The above statement does not address the issue because:
> > 1. Analyzing successive points is a special case of a contraction, in fact Theorem 1 in this paper shows a contraction but for some reason is presented as the weaker result of contraction of successive points.
> > 2.  "Removing condition $\rho<1$ and using an analogous condition", these conditions are the same.
> >
> > Therefore the result in Theorem 1 is simply skipping the first step of Theorem 7 in [1]. And if one were to compare the proofs it is easy to see they are almost identical.
> >
> > > While $\hat{\epsilon}_i$-sensitivity serves as a proof step analogous to Lemma 5 in [1], their distinct definitions lead to different practical applicability. We reduce the theoretical requirements by eliminating unnecessary conditions
> >
> > The condition in Lemma 5 in [1] looks identical to $\hat{\epsilon}_i$-sensitivity where $ \hat{\epsilon}_i = \beta_i \gamma_i $. There are no mention of other conditions in the results of Lemma 5.
> >
> > [1] A. Narang, E. Faulkner, D. Drusvyatskiy, M. Fazel, and L. J. Ratliff. Multiplayer performative prediction: Learning in decision-dependent games. JMLR, 24(202):1–56, 2023.

---

> > ### Comment · Reviewer_UCcE · 2025-11-17
> > **Question on updated stochastic result**
> >
> > I thank the authors for clarifying many of my doubts and updating the manuscript accordingly. Due to the many changes it is difficult for me to check everything but that being said I still have a remaining question. In the updated manuscript the following change is included:
> >
> > > Regarding the probabilistic guarantee of Theorem 3, we thank the reviewer for pointing out this issue. We agree that the probability should consider all previous time steps, since the convergence $||X^t-X^{PS}||<\delta$ requires Eq. (22) to hold at every time step. The probability in Theorem 3 should be $\prod _{\mu=1}^t(F _{\chi^2(d)}(\frac{m _\mu\sum _{i=1}^n\hat{\varepsilon} _i^2\delta^2}{\sigma _\mu}))$, which incorporates all historical sample sizes. This ensures that $X^t$ will enter and stay in the $\delta$-ball around $X^{PS}$ with the stated probability.
> >
> > Doesn't this assume that the event in Lemma 1 (eq 50) are independent at each timestep? This is not clear as the bound depends on the covariance matrix (eq 43) which depends on the iterates, therefore it may not be true that the eigenvalues of the covariance matrix are are independent?  In general one has to use a union bound if there is no assumption on dependence.
> >
> > I also appreciate the authors updating the manuscript to specifically mention they are using the CLT but there are some statements that should be altered for example, I think it is not accurate to say that Theorem 3 provides results in high probability with finite-samples if it is using Lemma 1 with the CLT (which assumes infinite samples).

---

> > > ### Author Response · Authors · 2025-11-18
> > > **Response to updated stochastic result**
> > >
> > > ---
> > >
> > > - We appreciate the reviewer for this insightful comment regarding the independence across time steps in Theorem 3. The reviewer is correct that the covariance matrix and its eigenvalues depend on the iterates. We agree that a union bound for the probabilistic guarantee regarding the distribution maps $\mathcal{D}_i(X)$ would be mathematically rigorous.
> > >
> > >   We acknowledge that the probabilistic bound in Lemma 1 depends on the covariance matrix and is applied at each iteration in Theorem 3. The main reason is that the distribution maps $\mathcal{D}_i(X)$ are unknown and can be arbitrarily complex. In this case, the sample sets from distributions after decision deployment can not be anticipated. Since the covariance matrix is computed from the sample sets at each time step, the dependence between covariance matrices across time steps is also unknown. Therefore, to obtain a practically feasible result, we assume the covariance matrices and the corresponding eigenvalues are independent across time steps under unknown distribution maps $\mathcal{D}_i(X)$.
> > >
> > >   Consequently, the product form of the probability $\prod_{\mu=1}^t(\cdot)$ in Theorem 3 is obtained from this independence assumption.
> > >
> > >   **Revision:** We have stated that Theorem 3 models the sample sets and covariance matrices as independent across time steps in Section 3.3 (last paragraph of Proof sketch of Theorem 3) and in Appendix G (last paragraph).
> > >
> > > ---
> > >
> > > - We thank the reviewer for the suggestion regarding the precision of our theoretical statement. We agree that our original phrasing could be confusing. To address this, we have revised the description of Theorem 3 to be accurate as follows:
> > >
> > >   >Theorem 3 guarantees the convergence to performatively stable equilibria with a probability. This guarantee is established by Lemma 1, which applies the Central Limit Theorem.
> > >
> > >   **Revision:** We have revised the statement of Theorem 3 in Section 3.3 (last paragraph).

---

> ### Author Response · Authors · 2025-10-30
> **Response 2**
>
> ---
>
> - Regarding the computation of equilibrium in each iteration, we appreciate the insightful consideration.
>
>   We confirm that the Nash equilibrium in Theorem 3 and the best-response process in Algorithm 1 are consistent. In a strongly monotone game, the Nash equilibrium can be obtained by calculating all players' best responses. This computation is compatible with gradient-based optimization, leading to the Nash equilibrium in each iteration before decision deployment.
>
>   We agree that the Nash equilibrium computation is challenging. Our framework focuses on strongly monotone games, where a Nash equilibrium is tractable through numerical methods. In our experiment, the equilibria for the separable games in Sections 5.1 and 5.3 are solved by optimizing each player's loss function. The equilibrium for the game in Section 5.2 is obtained by solving a linear system (detailed in Appendix J.5). Our core contribution lies in establishing the performatively stable equilibrium for arbitrary data distribution maps, rather than addressing computational complexity.
>
>   We agree that wall-clock time is an important metric for a fair comparison. The results in Appendix J.2 show that although SIR$^2$ requires more time in each iteration due to the computation of the full equilibrium, it converges in fewer iterations compared with baselines. Consequently, the total wall-clock time for SIR$^2$ to converge is comparable to that of the baselines.
>
>   **Revision:** We have explained the equilibrium calculation in Section 4 (second paragraph). We have provided a wall-clock time comparison in Appendix J.2.
>
> ---
>
> - Regarding the definition of $\beta_i$-smoothness, we thank the reviewer for the suggestion. The $\beta_i$-smoothness requires the Lipschitz smoothness in both data $z$ and decision $x$, and thus there are two conditions for this definition.
>
>   **Revision:** We have added the $\beta_i$-smoothness definition in Section 3.1 (last paragraph).
>
> ---
>
> - Regarding the exploration of performative gradient, we appreciate the reviewer for the comment. It means that AGM and OPGD perform gradient descent using the estimated gradient of performative loss, although the performative loss may not be convex.
>
>   **Revision:** We have updated the phrase in Section 5.1.2.
>
> ---
>
> - Regarding the proof in Theorem 2, we thank the reviewer for this insightful suggestion. The continuity of $Nash(\cdot)$ can be established by Eq. (35) in the proof of Theorem 1 (Appendix B). Specifically, since Eq. (35) holds for any joint decisions $X,X'\in\mathcal{X} _{RR}$ and $\mathcal{X} _{RR}$ is a closed convex hull, the $Nash(\cdot)$ is Lipschitz continuous on $\mathcal{X} _{RR}$. With this continuity property, we can take the limit inside the $Nash(\cdot)$.
>
>   **Revision:** We have proved the continuity of $Nash(\cdot)$ in Appendix C (last paragraph).
>
> ---
>
> - Regarding the citations of related work that applied $\mathcal{W}_1$ distance, we thank the reviewer for the suggestion.
>
>   **Revision:** We have added citations in Section 1 (fourth paragraph).
>
> ---
>
> - Regarding the small monotonicity parameter $\psi$, we appreciate the reviewer for this insightful question. The reviewer is correct that the performatively stable equilibrium may not exist for the original game with a small monotonicity parameter $\psi$. To address this, our algorithm adds $\frac{\gamma}{2}||x_i||_2^2$ to each loss function, ensuring the regularized game is $\psi+\gamma$-strongly monotone. By choosing $\gamma$ such that $\psi+\gamma>2\sqrt{\sum _{i=1}^n\hat{\varepsilon} _i^2}$, our Theorem 3 guarantees the convergence.
>
> ---

---

> > ### Comment · Reviewer_UCcE · 2025-10-30
> > **Best response calculation does not correspond to Nash**
> >
> > > We confirm that the Nash equilibrium in Theorem 3 and the best-response process in Algorithm 1 are consistent. In a strongly monotone game, the Nash equilibrium can be obtained by calculating all players' best responses. This computation is compatible with gradient-based optimization, leading to the Nash equilibrium in each iteration before decision deployment.
> >
> > This is not correct, in general one cannot reduce the computation of a Nash equilibrium to a best response calculation for each player even in strongly monotone games. Take for example the following 2 player zero-sum min-max game $\min \max \frac{x^2}{2} + x y -\frac{y^2}{2}$, this is smooth and strongly monotone. Let us consider the dynamics governed by iteratively doing best response, that is $ x_{t+1} = \arg \min \frac{x^2}{2} +x y_t -\frac{y_t^2}{2} = -y_t $ similarly $ y_{t+1} = x _t $.
> >
> > Therefore the dynamics of the system are $$ z_{t+1} = A z_t$$ where $A  = [[ 0, -1], [1, 0]]$ and $ z_t = [x_t, y_t]^\top$. for which is not guaranteed to converge and can cycle take for example the starting point $[1,1]$ then we see that the dynamics enter into the following cycle:
> > $$
> > [1,1] \Rightarrow  [-1,1] \Rightarrow  [-1,-1]  \Rightarrow  [1,-1] \Rightarrow  [1,1]
> > $$.

---

> ### Author Response · Authors · 2025-10-31
> **Response to the best response calculation**
>
> ---
>
> We sincerely thank the reviewer for considering our response and providing further comments regarding the best response calculation in our Algorithm and a clear example, which is helpful for us to understand the concern. This comment helps us to improve the clarity of our manuscript.
>
> The reviewer is correct that the iterative best-response dynamics described in the comment can fail to converge to a Nash equilibrium.
>
> However, we wish to clarify that the process in our Algorithm 1 (Step 6) is different. The best-response calculation in our algorithm anticipates the best response of other players. Specifically, all players simultaneously compute the Nash equilibrium by solving the system of optimality conditions (best response functions). The players choose the best response not to the previous decision $x^{t-1} _{-i}$, but to decisions $x^t _{-i}$ (the best response of other players) in time step $t$.
>
> ---
>
> For the example that the reviewer provided, our algorithm solves the equilibrium directly by considering the optimality conditions of both players, which is shown as follows.
>
> 1. We first calculate the gradient of the loss function for two players, and then set it to 0, which are the first-order optimality conditions (best response) of two players.
>
>   - For player $x$, we have:
> $$\nabla_x\ell_i = x + y=0.$$
>
>   - For player $y$, we have:
> $$\nabla_y\ell_i = x -y=0.$$
>
> 2. Solving this linear equation system yields the unique Nash equilibrium $(x^{Nash},y^{Nash})=(0,0)$.
>
> This process does not perform iterative updates based on previous decisions and computes the Nash equilibrium after the distribution shifts.
>
> ---
>
> **Revision:**
>
> - We have clarified that the best response computation in each iteration is defined as solving the Nash equilibrium for the game in time step $t$. (Section 4, second paragraph)
>
> - We have modified step 6 in Algorithm 1 clarify the best response process as $x_i^t $ $\leftarrow $ $ \arg \min_{x_i} \mathbb{E} _{Z^{t-1} _i\sim \mathcal{D} _i(X^{t-1})} \bar{\ell} _i(x _i,x _{-i}^{t},Z^{t-1} _i)$. (Section 4)

---

> ### Author Response · Authors · 2025-10-31
> **Response to the connection and distinction to related work**
>
> ---
>
> We sincerely appreciate the reviewer for providing valuable feedback on the connection and distinction to related work.
>
> ---
>
> We agree that our Theorems 1 and 2, as well as Theorem 7 in [1], guarantee the convergence under a condition. We acknowledge that the proofs for these theorems share a similar idea. In fact, our proofs of Theorems 1 and 2 are inspired not only by [1] but also by the analysis in [2,3], which investigates the repeated retraining (repeated risk minimization) procedure in the performative setting. The proofs in our work and in [1,2,3] all leverage the contraction mapping theorem (Banach fixed-point theorem), and thus, show contraction of successive points.
>
> We clarify that the core novelty of our work does not lie in the convergence proof itself. Instead, our contribution is the relaxation of unverifiable conditions required in prior work, making our framework practically applicable. The calculable condition $\hat{\varepsilon}_i$, supported by the theorems, is able to guide the algorithm design that guarantees the convergence to a performatively stable equilibrium for arbitrary data distribution maps.
>
> ---
>
> The $\hat{\varepsilon}_i$-sensitivity differs from Lemma 5 in [1] both theoretically and practically, and $\hat{\varepsilon} _i$ is not generally equal to $\rho = \beta _i \gamma _i$. The difference lies in the definition.
>
> - Practically, $\hat{\varepsilon} _i$ can be calculated, while $\beta_i$ might not be verifiable. This is because we define $\hat{\varepsilon} _i$-sensitivity on the space $\mathcal{X} _{RR}$ induced by the repeated retraining procedure. Since the decisions in $\mathcal{X} _{RR}$ are observable during repeated retraining, $\hat{\varepsilon} _i$ can be computed, as shown in Eq. (9). In contrast, $\beta_i$-smoothness holds for all samples that can be induced by any decision in the full decision space, i.e., $z _i\in \mathcal{Z} _i\:=\cup _{X\in\mathcal{X}}\sup(\mathcal{D} _i(X))$. Thus, $\beta _i$ may be difficult to verify for unknown distribution maps, as shown in Appendix A.
>
> - Theoretically, $\hat{\varepsilon}_i$ is smaller than or equal to $\beta_i\gamma_i$, leading to a more relaxed convergence condition. This is because $\hat{\varepsilon}_i$ is defined on $\mathcal{X} _{RR}$, which is a subspace of $\mathcal{X}$. This leads to a more relaxed requirement for the convergence guarantee. Specifically, our convergence guarantee requires the strong monotonicity parameter $\alpha$ to satisfy $\alpha>\sqrt{\sum _{i=1}^n\hat{\varepsilon} _i^2}$, while the result in [1] requires $\alpha'$ to satisfy $\alpha'>\sqrt{\sum _{i=1}^n\beta _i^2\gamma _i^2}$. Since $\hat{\varepsilon} _i\le\beta _i\gamma _i$, our requirement on $\alpha$ is more relaxed than that on $\alpha'$. Thus, the convergence requirement in our result is relatively looser.
>
> ---
>
> **Revision:**
>
> - We have acknowledged that our proof idea was inspired by prior works in Section 3.2 (last paragraph).
>
> - We have discussed the value of $\hat{\varepsilon}_i$ and parameters in prior works in Section 3.2 (last paragraph).
>
> ---
>
> **Reference:**
>
> [1] A. Narang, E. Faulkner, D. Drusvyatskiy, M. Fazel, and L. J. Ratliff. Multiplayer performative prediction: Learning in decision-dependent games. JMLR, 24(202):1–56, 2023.
>
> [2] J. Perdomo, T. Zrnic, C. Mendler-Dünner, and M. Hardt. Performative prediction. ICML’20, pp. 7599–7609. PMLR, 2020
>
> [3] G. Brown, S. Hod, and I. Kalemaj. Performative prediction in a stateful world. AISTATS’22, pp. 6045–6061. PMLR, 2022.
>
> [4] N. Fournier and A. Guillin. On the rate of convergence in wasserstein distance of the empirical
> measure. Probability Theory and Related Fields, 162(3):707–738, 2015

---

> > ### Comment · Reviewer_UCcE · 2025-10-31
> > **Question about  $\hat{\varepsilon} _i$**
> >
> > > Practically, $\hat{\varepsilon} _i$ can be calculated, while $\beta_i$ might not be verifiable. This is because we define $\hat{\varepsilon} _i$-sensitivity on the space $\mathcal{X} _{RR}$ induced by the repeated retraining procedure.
> >
> > Thank you for clarifying, so from my understanding the main difference is that you are defining a sensitivity measure of the $\mathcal{X} _{RR}$ instead of $\mathcal{X}$, when comparing the inequality of in Lemma 5 of [1] (that is set $\beta_i \gamma_i = \hat{\varepsilon} _i$). However, in general it seems that we cannot control what space of  $\mathcal{X}$ for which $\mathcal{X} _{RR}$ will be a subset? Therefore, without further information these conditions can be equal?
> >
> > However, I think I understand better why your result is a contraction of successive points instead of showing a contraction with respect to the fixed point, that is because you want to apply the sensitivity with respect to the consecutive points instead of with respect to $x_{t}$ and $x^\star$ ($x^\star$ being the Nash). It is clear from [1] that if one were to define the sensitivity with respect to successive points that is $y$ and $y'$ are chosen as in Theorem 1 then you would get the same result as Theorem 1. Both of these insights however weren't clear. I think it is more accurate to state Theorem 1 as a corollary of Theorem 7 in [1], since they show a contraction if you simply take  $y$ and $y'$  to be successive points in the repeated retraining procedure.
> >
> > [1] A. Narang, E. Faulkner, D. Drusvyatskiy, M. Fazel, and L. J. Ratliff. Multiplayer performative prediction: Learning in decision-dependent games. JMLR, 24(202):1–56, 2023.

---

> > > ### Author Response · Authors · 2025-11-01
> > > **Response to sensetivity definition**
> > >
> > > ---
> > >
> > > We appreciate the reviewer for this further comment regarding the $\hat{\varepsilon} _i$-sensitivity. The reviewer is correct that we cannot control the specific subspace $\mathcal{X} _{RR}$ relative to $\mathcal{X}$, and the conditions $\hat{\varepsilon} _i$ and $\beta _i \gamma _i$ can be equal in certain cases:
> > >
> > > - If the repeated retraining procedure explores the full decision space, then $\mathcal{X} _{RR} = \mathcal{X}$, leading to $\hat{\varepsilon} _i=\beta _i\gamma _i$. However, in practice, since $\mathcal{X} _{RR}$ contains the decisions during repeated retraining, $\mathcal{X} _{RR}$ generally is a proper subset of $\mathcal{X}$.
> > >
> > > - If the $\beta_i$-smoothness condition over $\mathcal{X} _{RR}$ is the same as that over $\mathcal{X}$, then we also have that these two conditions are equal. However, as shown in Appendix A, the value of $\beta_i$ over $\mathcal{X}$ may be difficult to verify with unknown distribution maps, making this equality generally unverifiable in practice.
> > >
> > > ---
> > >
> > > Regarding the contraction of successive points, the reviewer is correct that $\hat{\varepsilon}_i$-sensitivity can not be directly applied to $X^{PS}$. Since the $\hat{\varepsilon}_i$ is defined on the subspace $\mathcal{X} _{RR}$, and performatively stable equilibrium $X^{PS}$ may not lies in this subspace $\mathcal{X} _{RR}$, we can not directly replace the $X$ or $X'$ with $X^{PS}$ in Eq. (15). To address this, our Theorem 2 shows that $X^{PS}\in\mathcal{X} _{RR}$ under a condition of $\alpha>\sqrt{\sum _{i = 1}^n \hat{\varepsilon} _i^2}$.
> > >
> > > We agree that it is appropriate to state Theorem 1 as a corollary of Theorem 7 in [1], since the proofs share a similar structure, with the main difference lying in the parameter definition.
> > >
> > > ---
> > >
> > > **Revision:**
> > >
> > > - We have stated our original Theorem 1 as a corollary of Theorem 7 in [1] in Section 3.1 (first paragraph, Corollary 1 in revised manuscript).
> > >
> > > ---
> > >
> > > **Reference:**
> > >
> > > [1] A. Narang, E. Faulkner, D. Drusvyatskiy, M. Fazel, and L. J. Ratliff. Multiplayer performative prediction: Learning in decision-dependent games. JMLR, 24(202):1–56, 2023.

---

> ### Comment · Reviewer_UCcE · 2025-10-31
> **Best response calculation**
>
> Thank you for this clarification, it is clearer now what is meant. I would avoid the use of "best response" when talking about calculating a Nash equilibrium. For example this sentence is very confusing:
>
> > Following this, each player i computes the best-response $x^t_i$ under the game $\bar{G}(X_{t−1})$ induced by $X_{t−1}$
>
> Instead I would just say each player plays their par of a Nash equilibrium.
>
> Can you also describe how you solve for this equilibrium in practice? The example above you mention that you are solving a system of equations but in the text you mention that because it is strongly monotone you can solve for easily by gradient descent for each player. In practice are you using repeated gradient descent and if so how you define the stopping condition?
>
> I see it is mentioned that the system of equations approach is used for the cournot game but not for the other games. In general this seems challenging if the loss functions are more complex?

---

> > ### Author Response · Authors · 2025-11-01
> > **Response to equilibrium computation**
> >
> > ---
> >
> > We thank the reviewer for these insightful suggestions regarding the equilibrium computation. We agree that the original phrasing could be confusing and have revised it.
> >
> > ---
> >
> > We are pleased to clarify the equilibrium-solving procedures in our experiments.
> >
> > - For Section 5.1, the game is separable, and the equilibrium can be achieved by minimizing each player's loss function in each iteration. We use 'sklearn.linear\_model.Ridge' to solve the regression problems in each iteration.
> >
> > - For Section 5.2, the solution for this non-separable game is detailed in Appendix J.5, where we solve the linear equation system.
> >
> > - For Section 5.3, the game is separable. We use the closed-form solution $x_i=\frac{z_i}{\alpha}$ to minimize the loss functions in each iteration.
> >
> > We clarify that we do not apply gradient descent in our experiment, since the games in our experiments are tractable via the above approach, making the gradient-based optimization approach unnecessary.
> >
> > ---
> >
> > We clarify that our framework is based on a repeated retraining procedure, which differs from repeated gradient descent. The difference is the timing of deploying the decisions:
> >
> > - **Repeated retraining**: In each iteration, players compute the Nash equilibrium and deploy the corresponding decisions under the current data distributions. Then, the data distributions shift due to the deployment of the Nash equilibrium decisions.
> >
> > - **Repeated gradient descent**: In each iteration, players compute a single gradient step from their previous decisions and deploy the updated decisions. Then, the data distribution shifts due to these partially updated decisions.
> >
> > ---
> >
> > We acknowledge that computing Nash equilibria for more complex, non-separable games can be challenging.
> >
> > ---
> >
> > **Revision:**
> >
> > - We have rephrased the sentence in Section 4 (second paragraph).
> >
> > - We have clarified the solution of the equilibrium in Sections 5.1 and 5.3.

---

### Review · Reviewer_ND5j · 2025-10-16

**Summary Of Contributions:**

This paper studies an extension of the performative prediction problem to multi-player settings, "decision-dependent games", in which multiple players make decisions to maximize a payoff, which influence some responsive distribution that future payoffs depend on (e.g., markets or user behavior). Prior work has studied algorithms that converge to "performatively stable equilibria" under a strong, unverifiable conditions of $\beta$-smoothness; this paper introduces an algorithm that relaxes this condition, and prove convergence even under finite samples from the responsive distribution. The authors also run numerical experiments and determine that the proposed algorithm performs very well in practice.

Strengths: The paper is clearly written, and has strong results which improve over prior work both theoretically and practically. In particular, the experimental performance is impressive.

Weaknesses:
(a) Theorem 3: First, in Theorem 3, is an equivalent statement $\forall t \geq N(...), P(\|X^t - X^{PS}) \geq f(m_t, ...)$ for some function $f$ of $m_t$ and some $N$ that's constant in $m_t$? (If so, I would recommend altering the phrasing, since the current presentation makes it seem like the claim is that $P(for all t, ....) \geq $). But my main concern out of this is how the lower bound for $t$ is independent of $m_t$ (and all previous $m_{t'}, t' < t$) -- if for all previous $t'$ you have $m_{t'} = 1$, how can you guarantee that $\|X^t - X^{PS}\}$ is small?

I suspect perhaps the solution is because $\sigma$ depends on the information learned in previous iterations, in which case this should be reflected in the notation. I also think it would be helpful to explain intuition for $\sigma$ and how large it should be expected to be, to make the results practically helpful. I do notice that $\sigma$ appears to be calculable from the sampled data, which is a strength.

(b) Algorithm: While the algorithm clearly has good experimental error minimizing / revenue maximizing behavior, does it actually converge to a stable equilibrium of the original game, or just of the regularized game? How unstable is it, if not? (Theorem 4 says it's the "least different" regularization, but this could still be quite far?)

**Audience:**

Yes

**Audience Explanation:**

Yes, this is an interesting result in performative prediction.

**Claims And Evidence:**

Yes

**Claims Explanation:**

The paper is clearly written, and the results appear sound, aside from the concerns above about Theorem 3; I will re-evaluate this after hearing from the authors about this.

**Requested Changes:**

Essential changes:
- Please answer my questions about Theorem 3 in the weaknesses section above
- It would be very helpful to gain some intuition for how large $m$ has to be in Theorem 3, especially since $\sigma$ is somewhat uninterpretable.

Non-essential changes / suggestions:
- If you have insight into my question (b) that would be great to see; also if the algorithm does not converge to a PSE of the original game and actually has a fairly unstable solution, it would be good to state.
- In section 1.1, $\alpha$ is mentioned in bullet point 1 without any previous definition; it would be helpful to add some context.
- Is Equation 23 missing a comma in $\theta_{i \quad X}$?
- The text mentions that $\alpha$-strong monotonicity is sufficient to guarantee _existence_ of a unique Nash equilibrium, but it would be good to cite results showing that strong monotonicity also implies that learning will converge to this equilibrium. It's also ill-defined exactly what process is occurring during Step 6 in Algorithm 1 -- what is $x_{-i}$? Are all players doing a no-regret learning subprocess?
- In Table 2, it's difficult to compare the numbers and intuit the width of the confidence intervals. It would be helpful to report these results in scientific notation.

---

> ### Author Response · Authors · 2025-10-30
> **Response 1**
>
> We sincerely appreciate the reviewer's valuable time spent in carefully reviewing our manuscript, and the recognition of the clear structure, theoretical and practical contributions, and extensive experimental validation.
>
> We also greatly appreciate the constructive feedback and detailed suggestions expertly provided in the theoretical analysis and algorithm design. Below, we respond to the reviewer's valuable feedback. All revisions made in the manuscript are highlighted in blue for easy identification.
>
> ---
>
> - Regarding the probabilistic guarantee in Theorem 3, we thank the reviewer for the insightful comment and for providing the opportunity to clarify this point.
>
>   The reviewer is correct that our Theorem 3 claims $P(\forall t\ge N, ||X^t-X^{PS}||<\delta)>f(m_t,\cdots)$, where $N$ is independent to sample size $m_t$. We acknowledge the lack of rigor in the probability in the original formulation. The probability should account for all previous time steps, since $||X^t-X^{PS}||<\delta$ is guaranteed when Eq. (22) is satisfied at every time step. Thus, the correct probabilistic guarantee in Theorem 3 is $\prod _{\mu=1}^t(F _{\chi^2(d)}(\frac{m _\mu \sum _{i = 1}^n\hat{\varepsilon} _i^2\delta^2}{\sigma _\mu}))$, which incorporates all historical sample sizes $m _{t'}$. The result ensures that, with this probability, $X^t$ will enter and stay in a $\delta$-ball around $X^{PS}$ for $t\ge N$.
>
>   We agree to denote $\sigma$ as $\sigma_t$. We will present the value of $\sigma_t$ in our experiments in the next response. We clarify that $\sigma_t$ depends only on data at step $t$, not on previous iterations, which is considered in the corrected probability.
>
>   **Revision:** We have revised probability in Section 3.3 (Theorem 3 and proof) and Appendix G.
>
> ---
>
> - Regarding the sample size $m_t$ for convergence, we appreciate the reviewer for the suggestion. The probabilistic guarantee in Theorem 3 depends on parameters $\sigma_t,\hat{\varepsilon}_i$, which can be estimated from our code. Based on our experimental observation that SIR$^2$ converges around 10 time steps, we calculate the required sample size $m_t$ for convergence with time step $t=10$ under at least 95\% probability (requiring $F _{\chi^2(d)}(\cdot)\ge0.995$ for each time step). In our experiment, we use the same sample size for all time steps, i.e., $m_t = m,\forall t$.
>
>   - **Section 5.1**: With distribution map parameter $\sigma_{a_i}^2=2.5$, the number of players $n=2$, and the data dimension $d=10$, we have the result $\sigma_t\approx2.5 \times 10^{-8},\sum _{i=1}^n\hat{\varepsilon} _i^2\approx100$ from our code. To achieve a convergence radius of $\delta=10^{-4}$, the condition $F _{\chi^2(10)}(40 m_t)\ge0.995$ must be satisfied. The critical value of the $\chi^2$ distribution with 10 degrees of freedom for a cumulative probability of 0.995 is 25.188. Therefore, we require $40m_t\ge25.188$, which yields $m_t\ge0.6297$.
>
>   - **Section 5.2**: With distribution map parameter $\mu_A=0.25$, the number of players $n=28$, and the data dimension $d=1$, we have the result $\sigma_t\approx0.003,\sum _{i=1}^n\hat{\varepsilon} _i^2\approx1.25\times 10^{-11}$ from our code. We set the convergence radius of $\delta= 2000\times 28 = 5.6\times 10^{4}$, meaning that the average gap between $X^t$ and $X^{PS}$ is less than 2000 barrels of crude oil for each country. Then, the condition $F _{\chi^2(1)}(13.06m_t)\ge0.995$ must be satisfied. The critical value of the $\chi^2$ distribution with 1 degree of freedom for a cumulative probability of 0.995 is 7.879. Therefore, we require $13.06m_t\ge7.879$, which yields $m_t\ge0.6033$.
>
>   - **Section 5.3**: With distribution map parameter $\mu_A=0.25$, the number of players $n=2$, and the data dimension $d=11$, we have the result $\sigma\approx2\times 10^{-4},\sum _{i=1}^n\hat{\varepsilon}_i^2\approx2500$ from our code. To achieve a convergence radius of $\delta=0.01$, the condition $F _{\chi^2(11)}(1250m_t)\ge0.995$ must be satisfied. The critical value of the $\chi^2$ distribution with 11 degrees of freedom for a cumulative probability of 0.995 is 26.757. Therefore, we require $1250m_t\ge26.757$, which yields $m_t\ge0.0215$.
>
>   Thus, $m_t=1$ is sufficient for at least 95\% convergence probability in our experiments.
>
> ---

---

> ### Author Response · Authors · 2025-10-30
> **Response 2**
>
> ---
>
> - Regarding the details of our algorithm, we thank the reviewer for the suggestion.
>
>   We confirm that Algorithm 1 converges to a regularized game, since the performatively stable equilibrium convergence may fail for the original games with a small monotonicity parameter $\psi$.
>
>   The baseline RR method applies repeated retraining directly to the original game. As shown in experiments, it fails to converge to a performatively stable equilibrium. Consequently, its performance drops after a distribution shift.
>
>   The reviewer is correct that the gap between the original and the regularized equilibrium can be large. For small $\psi$ and highly sensitive distribution map (large $\hat{\varepsilon}_i$), we have to add a regularizer with large $\gamma$ to guarantee the convergence. However, we provide Theorem 4 to illustrate that the regularizer $R(x_i)=\frac{\gamma}{2}||x_i||_2^2$ used in our algorithm minimizes the upper bound of this gap.
>
>   **Revision:** We have clarified the convergence for the regularized game in Section 4 (last paragraph), and provided an example of non-convergence in repeated retraining for the original game in Appendix H.
>
> ---
>
> - Regarding the explanation of $\alpha$, we thank the reviewer for pointing out this point. $\alpha$ is the strong monotonicity parameter of a game, which can be tuned by a regularization approach.
>
>   **Revision:** We have explained $\alpha$ in Section 1.1 (second paragraph).
>
> ---
>
> - Regarding Eq. (23) (Eq. (24) in the revised manuscript), we appreciate the reviewer for the observation. The notation is intentional. In this equation, $\theta_{i\ X}$ denotes the shifted sample after joint decision $X$ is deployed, and $\theta_i$ denotes the original sample.
>
> ---
>
> - Regarding the existence of a Nash equilibrium, Step 6 in the algorithm, and the game setting, we appreciate the reviewer for the comments.
>
>   - We agree with the reviewer that citing the result for the existence of a Nash equilibrium [1] provides a rigorous presentation.
>
>   - We clarify that Step 6 in Algorithm 1 computes the best response for each player, which solves the Nash equilibrium in each iteration. Here, $x_{-i}$ represents the joint decision of all players except for player $i$, which is defined in Eq. (2).
>
>   - We clarify that our work assumes a complete information game, where all players are fully rational and make decisions by considering the optimal decisions of other players.
>
>   **Revision:** We have added the citation [1] in Section 3.2 (second paragraph), explained the equilibrium calculation in Section 4 (second paragraph), and clarified the game setting in Section 2 (second paragraph).
>
> ---
>
> - Regarding the presentation of Table 2, we thank the reviewer for the suggestion. We agree with the reviewer that it improves the clarity of the experimental results.
>
>   **Revision:** We have revised Table 2 to use scientific notation.
>
> ---
>
> **Reference:**
>
> [1] J. B. Rosen. Existence and uniqueness of equilibrium points for concave n-person games. Econometrica: Journal of the Econometric Society, pp. 520–534, 1965.

---

### Comment · Action_Editor_45em · 2025-09-16
**Delay in reviewing**

Dear authors, reviewers, editors:

I had a very difficult time finding qualified reviewers for this paper. Several reviewers were unavailable and I received four declines from custom invitations (outside the usual TMLR reviewer pool).

In the end, I have found three who are very well-suited to review it. However, due to time constraints and schedules, we will be unable to receive reviews within the usual time frame. Given timelines, I am postponing the due date of the reviews to October 14th.

I apologize for the inconvenience, but unfortunately this is an exceptional circumstance. I believe this is better than the alternative of recruiting different reviewers who would be not be matched well to the parrticular topic and content.

Thanks,
Your Action Editor

---

> ### Author Response · Authors · 2025-10-30
>
> We sincerely thank the Action Editor for the considerable effort in finding qualified reviewers for our manuscript. We are grateful for the detailed explanation of the situation and fully understand the challenges. We also appreciate the commitment to matching our work with suitable experts, whose insightful suggestions are invaluable.
>
> ---
>
> In response to the reviewers, we have carefully addressed all reviewers' comments and suggestions. Specifically, we have:
>
> - enhanced related work discussion by adding proper citations in Section 1 (third paragraph), Section 3.2 (second paragraph), and clarifying the connections and distinctions in Section 3.1 (second paragraph), Section 3.2 (last paragraph), and Section 5.2 (first paragraph).
>
> - improved clarity of notations and problem definition in Section 1.1 (second paragraph), Section 2 (second paragraph).
>
> - strengthened theoretical results by providing additional lemmas and theorems for finite sample analysis in Appendices E, F, and G.2, updating the probabilistic guarantee, motivation, and presentation in Section 3.3 (Theorem 3), Section 4 (last paragraph and Theorem 4), and Appendices C and G.
>
> - improved methodological detail in Section 3.3 (last paragraph), Section 4 (second and third paragraphs).
>
> - enhanced experimental presentation in Section 5.1.1 (first paragraph), Section 5.2.1 (first paragraph), Section 5.3.1 (first paragraph), and Appendix J.2.
>
> ---
>
> A detailed point-by-point response to each reviewer's comments is provided, explaining how their feedback has been incorporated. All revisions made in the manuscript are highlighted in blue for easy reference.

---

### Decision · Action_Editor_45em · 2025-12-02

**Recommendation:** Reject

**Additional Comments:**

With the volume of edits and lack of confidence from all reviewers, this paper need another round of new reviews. Specifically, it needs to be reviewed by an expert in this area and more confident recommendations.

It is unclear that TMLR is the right choice of audience and I would encourage the authors to reconsider carefully the choice of venue for this paper.

Some other minor comments made by the reviewers in their final recommendations:

> I encourage the authors to include the argument that the
 required will be small, and to emphasize that while the algorithm uses the best regularizer, the equilibrium it converges to could potentially be quite different from the true equilibrium

> The convergence of Algorithm 1 (SIR^2) is now explained clearly, although it is somewhat disappointing that it only converges to the performative stable equilibrium of the regularized game rather than that of the original game

**Audience:**

Yes

**Audience Explanation:**

I am choosing "Yes" in this category because all three reviewers chose "Yes" in their official recommendations.

However, this was a surprise to me because I feel that ICLR is not a good venue for this paper. There is a tenuous connection to machine learning because the algorithm is gradient based and the prior work in builds on Narang 2023 which was published in JMLR, but there are better choices more focused on game theory and optimization. In terms of conferences: EC, WINE, SAGT, SODA, and AAMAS are all better fits. In terms of journals: this topic may be a better fit to Games & Economic Behavior, SIAM Journal on Control and Optimization.

I had a particularly difficult time finding qualified reviewers for this paper because of this audience/expertise mismatch (in fact, I had to recruit some externally). I am not convinced that this paper would be well-received at TMLR. As stated above, I suggest the authors try different venues and if they choose a machine learning venue, maybe JMLR is a better fit since matches the venue of Narang 2023.

**Claims And Evidence:**

No

**Claims Explanation:**

The authors present a densely technical paper, and many technical questions came up during review.

Several clarifications were made by authors and some of the reviewers are happy with the changes but have a low confidence in their (weak) recommendations.

Another reviewer is still negative citing the sheer number of clarifications that were needed leading to a very low confidence in the correctness of the result.

---

> ### Author Response · Authors · 2025-12-04
>
> We sincerely thank the Action Editor for the efforts in finding qualified reviewers for our work and for the decision comments. The reviewers' comments are insightful, and addressing their concerns strengthens our paper.
>
> The suggestions regarding alternative venues are appreciated and will be considered before our next submission.
>
> Thank you again for your support and professionalism throughout the process.

---

> ### Comment · Action_Editor_45em · 2025-12-04
> **A few more thoughts**
>
> Dear authors,
>
> Thank you for the response. Actually, I have been thinking about this paper since yesterday, and I would to add a few more thoughts.
>
> First, I'd like to encourage you to resubmit this paper -- many changes were made to accommodate the reviewers' comments and I do feel it improved significantly as a result of the review process. Two of the three reviewers were positive in their final recommendations; the main problem was outstanding uncertainty around correctness and nobody had a high enough confidence.
>
> Where to resubmit it is the real question. I stand by the suggestions for better audience fits, but I would like to say say one thing in favor of resubmission to TMLR in case you ultimately decide on resubmitting to TMLR. After having gone through the paper in more detail and upon reflection, I have a much better feel for the subject content and I'm confident that the reviewer matching could be easier on a resubmission (even if I'm not the assigned Action Editor, I could help the assigned Action Editor with reviewer matching).

---

> > ### Author Response · Authors · 2025-12-04
> >
> > Dear Action Editor,
> >
> > Thank you for your thoughtful effort on our work, even after the decision has been made, which has given us more confidence to resubmit it.
> >
> > After careful consideration, we decide to resubmit this work to TMLR.
> >
> > According to the message sent via OpenReview, our understanding is that we need to create a new submission to TMLR; and we are required to provide the link to this submission as well as a description of the changes made.
> >
> > We would like to seek advice on the following issues:
> >
> > 1. Is our understanding correct?
> >
> > 2. If so, what changes should we describe? Should the changes be compared with the original submission in August 29 or the latest revised version with blue highlights in November 18?
> >
> > 3. Do we need to keep the blue-highlighted revisions in the new submission?
> >
> > 4. In addition to the changes that have already been made, can we make further changes on the resubmission? E.g., we may move the proof sketch of some theorems to the Appendix to ensure the main paper is still within 12 pages.
> >
> > Thank you again for your professionalism and encouragement.

---

> ### Comment · Action_Editor_45em · 2025-12-04
> **Response**
>
> Dear authors,
>
> 1. Confirmed: correct, you need to create a new submission and you're required to link to this one.
>
> 2. Both. But also, the most negative reviewer was still mostly unsure about the stochastic results. If there are any outstanding clarifications to make there, please do.
>
> 3. No, please remove the blue highlights, because it's a completely new submission. This is the reason to ask for a description of the changes separately.
>
> 4. Yes. That is what separates a "Reject + Re-submit" from "Accept with Minor Revisions". Please absolutely, if you intend to keep it in the same category (short paper) then it is required to stay under 12 pages.
>
> Hope this helps.

---

> > ### Author Response · Authors · 2025-12-05
> >
> > Dear Action Editor,
> >
> > Thank you for the reply. We will consider your advice and revise our work accordingly, then proceed to submit the revised manuscript.